

# Need for split: integrative taxonomy reveals unnoticed diversity in the subaquatic species of *Pseudohygrohypnum* (Pylaisiaceae, Bryophyta)

Vladimir E. Fedosov[1,2], Anna V. Shkurko[3], Alina V. Fedorova[3], Elena A. Ignatova[2], Evgeniya N. Solovyeva[4], John C. Brinda[5], Michael S. Ignatov[2,3] and Jan Kučera[6]

[1] Botanical Garden-Institute, FEB RAS, Vladivostok, Russia
[2] Biological Faculty, Lomonosov Moscow State University, Moscow, Russia
[3] Tsitsin Main Botanical Garden of RAS, Moscow, Russia
[4] Zoological Museum of Lomonosov Moscow State University, Moscow, Russia
[5] Missouri Botanical Garden, St. Louis, MO, USA
[6] Faculty of Science, Department of Botany, University of South Bohemia, České Budějovice, Czech Republic

Corresponding author
Vladimir E. Fedosov,
fedosov_v@mail.ru

## ABSTRACT

We present an integrative molecular and morphological study of subaquatic representatives of the genus *Pseudohygrohypnum* (Pylaisiaceae, Bryophyta), supplemented by distribution modelling of the revealed phylogenetic lineages. Phylogenetic analyses of nuclear and plastid datasets combined with the assemble species by automatic partitioning (ASAP) algorithm revealed eight distinct species within the traditionally circumscribed *P. eugyrium* and *P. subeugyrium*. These species are therefore yet another example of seemingly widely distributed taxa that harbour molecularly well-differentiated lineages with narrower distribution ranges. Studied accessions that were previously assigned to *P. eugyrium* form three clearly allopatric lineages, associated with temperate regions of Europe, eastern North America and eastern Asia. Remarkably, accessions falling under the current morphological concept of *P. subeugyrium* were shown to be even more diverse, containing five phylogenetic lineages. Three of these lineages occur under harsh Asian continental climates from cool-temperate to Arctic regions, while the remaining two, referred to *P. subeugyrium* s.str. and *P. purpurascens*, have more oceanic North Atlantic and East Asian distributions. Niche identity and similarity tests suggested no similarity in the distributions of the phylogenetically related lineages but revealed the identity of two East Asian species and the similarity of two pairs of unrelated species. A morphological survey confirmed the distinctness of all eight phylogenetic lineages, requiring the description of five new species. *Pseudohygrohypnum appalachianum* and *P. orientale* are described for North American and East Asian plants of *P. eugyrium* s.l., while *P. sibiricum*, *P. subarcticum* and *P. neglectum* are described for the three continental, predominantly Asian lineages of *P. subeugyrium* s.l. Our results highlight the importance of nontropical Asia as a center of bryophyte diversity. Phylogenic dating suggests that the diversification of subaquatic *Pseudohygrohypnum* lineages appeared in late Miocene, while mesophilous species of the genus split before Miocene cooling, in climatic conditions close to those where the ancestor of *Pseudohygrohypnum* appeared. We speculate that radiation of the *P. subeugyrium*

complex in temperate Asia might have been driven by progressive cooling, aridification, and increases in seasonality, temperature and humidity gradients. Our results parallel those of several integrative taxonomic studies of North Asian mosses, which have resulted in a number of newly revealed species. These include various endemics from continental areas of Asia suggesting that the so-called Rapoport's rule of low diversity and wide distribution range in subpolar regions might not be applicable to bryophytes. Rather, the strong climatic oscillations in these regions may have served as a driving force of speciation and niche divergence.

## INTRODUCTION

The genus *Pseudohygrohypnum* was introduced by *Kanda (1976)* to accommodate two autoicous species of subaquatic pleurocarpous mosses, previously placed in the genus *Hygrohypnum*, that have serrulate leaf apices and well-differentiated, coloured and often excavate alar cells. However, this novelty was not broadly accepted until molecular phylogenetic data for *Hygrohypnum/Pseudohygrohypnum eugyrium* first appeared. According to the phylogenetic reconstruction by *Gardiner et al. (2005)*, the generitype of *Hygrohypnum*, *H. luridum* (Hedw.) Jenn. clearly belongs to Amblystegiaceae, while *H. eugyrium* appears in a clade assigned to Pylaisiaceae. A similar result was published by *Oliván, Hedenäs & Newton (2007)*, that also showed the close relationship between *H. eugyrium* and *H. subeugyrium*. Based on molecular data later published by *Gardiner et al. (2005)*, *Ignatov & Ignatova (2004)* accepted the genus *Pseudohygrohypnum* and transferred *H. subeugyrium* Renauld & Cardot to it. *Kučera et al. (2019)* in their morpho-molecular revision of the genus *Hypnum* suggested broadening of the concept of *Pseudohygrohypnum* by placing a suite of non-aquatic species from sect. *Fertilia* (*H. calcicola, H. fauriei, H. fertile*, etc.) within it. Currently, according to the Tropicos online database (http://www.tropicos.org/name/35001387, accessed 28 Oct 2021) and The Bryophyte Nomenclator (https://www.bryonames.org/nomenclator?group=Pseudohygrohypnum, *Brinda & Atwood, 2021*), the genus includes ten accepted species and one species in synonymy.

Although the diversity of *Pseudohygrohypnum* in its current delimitation is largely concentrated in temperate East Asia, subaquatic or hygrophytic species of the genus in the region remain insufficiently known and have generally been treated as synonymous with the species from Europe and Eastern North America, *i.e., P. eugyrium* and *P. subeugyrium* (*Jamieson, 1976*; *Jamieson, 2014*; *Czernyadjeva, 2003*; *Hu et al., 2008*; *Blockeel, Kiebacher & Long, 2019*). Japanese authors have also accepted two additional taxa for eastern Asia, *Hygrohypnum purpurascens* Broth. (*Kanda, 1976*; *Iwatsuki, 1991*; *Noguchi, 1991*) and *H. subeugyrium* var. *japonicum* Cardot (*Iwatsuki, 1991*; *Noguchi, 1991*), but these have not become more widely accepted.

Potentially unnoticed diversity was suspected due to conflicts between the descriptions and illustrations of *P. eugyrium* and *P. subeugyrium* based on plants from different areas of their distribution. While *Lüth (2019)* and *Blockeel, Kiebacher & Long (2019)* illustrated European *P. subeugyrium* as lacking a central strand, and the same is true for the illustration of North Siberian plants by *Czernyadjeva (2003)*, *Ignatov & Ignatova (2004)* depicted a well-developed central strand based on a specimen from Bashkiria (south Ural Mountains). In addition, North Siberian plants are substantially smaller, with leaves shorter than one mm and denser branching, reminiscent of non-aquatic species of the genus. Similar discrepancies exist in descriptions of *P. eugyrium*: while *Jamieson (1976)* emphasized coloration of the alar cells as quite distinctive for the species, *Kanda (1976)* did not mention this for Japanese plants, which agrees with plants from the Russian Far East that have hyaline or greenish, rarely partly brownish but never reddish alar groups. In addition to differences in the morphological descriptions of the plants, the unusual distribution pattern of *P. subeugyrium* s.l. also raised doubts about its integrity; in Europe and North America, it is associated with areas of rather mild oceanic and suboceanic climates, very different from the climatic conditions under which the species occurs in north Siberia (*Czernyadjeva, 2003*; *Fedosov & Ignatova, 2005*; *Ivanov et al., 2017*) or Transbaikalia (*Afonina, 2009*; *Ivanov et al., 2017*). Considering recent treatments of species with seemingly disjunct distributions (*e.g.*, *Vigalondo et al., 2019*), we decided to check the identity of Asian subaquatic *Pseudohygrohypnum* species using molecular data, which were so far only available for plants from subatlantic populations. We also decided to verify the somewhat arguable monophyly of *Pseudohygrohypnum* as defined by *Kučera et al. (2019)* with respect to the different topologies obtained by *Câmara et al. (2018)* and *Schlesak et al. (2018)* using additional plastid markers. To account for differences in the ecology of subaquatic *Pseudohygrohypnum* species in different areas of their distribution, we attempted to employ species distribution modelling within an integrative taxonomic framework.

## MATERIALS & METHODS

### Molecular phylogenetic methods

The study is based on the analysis of plants currently referrable to *Pseudohygrohypnum eugyrium* and *P. subeugyrium*, for which 59 accessions were included that cover their known distribution area. To place our data in a broader phylogenetic context, we added a selection of non-aquatic representatives of the genus and outgroup Pylaisiaceae taxa identified using the dataset of *Kučera et al. (2019)*. In total, 104 accessions were included in our study, of which 70 represent seven currently recognized species of the genus *Pseudohygrohypnum*: *P. eugyrium* (16), *P. subeugyrium* (43), *P. calcicola* (2), *P. densirameum* (2), *P. fauriei* (3), *P. fertile* (2), and *P. skottsbergii* (2). Molecular sampling included the nuclear internal transcribed spacers 1, 2 and 5.8S rRNA gene as well as the plastid *trnL–trnF* region and *atpB–rbcL* spacer that were used by *Kučera et al. (2019*; see also *Stech & Quandt, 2010* for details). In addition, we added newly sampled sequences for the *rps4–trnS* region, *rbcL* gene and *trnK–psbA* region; the two latter regions were obtained only for selected accessions

from each lineage that were later identified as species. Specimen vouchers and GenBank accession numbers are provided in the Supplemental Information 1.

The laboratory protocols followed our previous studies (*Gardiner et al., 2005*; *Fedosov et al., 2016b*; *Kučera et al., 2019*). For the *rbcL* gene and *trnK–psbA* region, we designed new primers based on published sequences for other hypnalean chloroplast genomes (see Supplemental Information 2). The length of the *trnK–psbA* region (ca. 2,600 bp) necessitated amplification and sequencing in four parts; similarly, the *rbcL* gene (1,428 bp) was amplified and sequenced in two parts. Sequences were aligned using the online interface of MAFFT v.7.487 (*Katoh & Standley, 2013*), applying the E-INS-i strategy and otherwise default settings before manual fine-tuning in BioEdit (*Hall, 1999*). Indels were scored using the simple indel coding approach (*Simmons & Ochoterena, 2000*) in SeqState 1.4.1 (*Müller, 2005*). One of two alternative states of a highly homoplastic inversion in the *trnL–trnF* spacer, missing in subaquatic species of *Pseudohygrohypnum* and within the genus present only in *P. fauriei* was inserted as a reverse complement and coded as a deletion that enables considering this state as binary code by SeqState and keeping further changes in the alternative state available for analysis. In the single gene analyses, the ITS matrix (113 terminals, 930 positions, 161 indels) was divided into three partitions, which corresponded to ITS1, 5.8S rRNA gene and ITS2; plastid markers were divided into three partitions, one for cds, one for coding tRNAs, and one for non-coding parts (spacers and introns), except for *atpB–rbcL* which was coded as a single partition. In the combined cp dataset plastid data (*atpB–rbcL*, *trnL–trnF*, *rps4*, *trnK–psbA*, *rbcL*, named cpATRKR dataset hereafter, with 112 terminals, 5,995 positions, 112 indels), the matrix was divided into two partitions, one for *trnK–psbA* plus *rbcL* where a significant part of data was absent, and one for *atpB–rbcL* plus *trnL-trnF* plus *rps4 trnS*, as suggested by Partitionfinder 2.1.1 (*Lanfear et al., 2017*). Phylogenetic analyses were performed using Bayesian Inference (BI) estimated by MrBayes 3.2.7a (*Ronquist et al., 2012*) and Maximum Likelihood (ML) analysis calculated in RAxML 8.2.12 (*Stamatakis, 2014*). BI was run in two parallel analyses, each consisting of six Markov chains, 10 000 000 generations with the default number of swaps and a sampling frequency of one tree each 2000 generations, the chain temperature was set at 0.02. The GTR+G+I substitution model was selected for nuclear and combined plastid dataset based on the Akaike information criterion assessed by MegaX (*Kumar et al., 2018*). The convergence between runs assessed as an average split deviation frequency lower than 0.01 was reached after 0.5–1 million generations. Additionally, PSRF values were checked to be close to 1.000 and ESS values were checked to be higher than 200 using Tracer v.1.7.2 (*Rambaut et al., 2018*). Consensus trees were calculated after omitting the first 25% trees as burn-in. In ML analysis, robustness of the nodes was assessed using the thorough bootstrapping algorithm (*Felsenstein, 1985*) with bootstopping based on the extended majority rule consensus tree criterion. Analyses were performed on the Cipres Science Gateway (http://www.phylo.org/portal2) on XSEDE (*Miller, Pfeiffer & Schwartz, 2010*) and MetaCentrum VO computing infrastructure (https://metavo.metacentrum.cz/). Trees were rooted by two accessions of *Stereodon*, based on the topology published by *Kučera et al. (2019)*, and visualized using FigTree 1.4.3 (*Rambaut, 2009*).

Since the ingroup topologies inferred from the nuclear and plastid markers differed in their topology at multiple well-supported nodes, the combined nr & cp dataset was not analysed. As the incongruence between nr and cp-based topologies might imply reticulate evolution or incomplete lineage sorting, we used the Neighbour Net method implemented in SplitsTree v.4.16.2 (*Huson & Bryant, 2006*) to visualize affinities between lineages within the crown Pylaisiaceae clade (*i.e.*, the clade remaining after the *Aquilonium* clade split). The split trees were built based on nr ITS and cp *atpB–rbcL*, *trnL–trnF*, *rps4* (cp ATR hereafter) datasets with 1,000 bootstrap iterations. Testing for best schemes of assembling species by automatic partitioning of lineages to account for lumping *vs.* splitting hypotheses in *Pseudohygrohypnum* was performed for 64 *Pseudohygrohypnum* accessions for which all-four markers were available with unscored indels using the online interface of assemble species by automatic partitioning (ASAP) (https://bioinfo.mnhn.fr/abi/public/asap/, *Puillandre, Brouillet & Achaz, 2020*) with the default parameters. Statistics for ITS and cp ATR *Pseudohygrohypnum* alignments were as follows: alignment length 930 and 1,835 bp respectively; 92 and 114 sites were variable, 79 and 108 were parsimony informative. Since the cp and ITS datasets resulted in different topologies, we ran the ASAP test for the nuclear and cp data separately, searching for a scheme of dividing the dataset into species which (1) would not challenge accepted species concepts outside the two subaquatic *Pseudohygrohypnum* species, under consideration in the present study; (2) could be assigned to both the plastid and nuclear datasets; (3) would not contradict the topologies of the phylogenetic trees (since ASAP uses similarity of pairwise genetic distances rather than phylogenetic inferences), (4) would have threshold distance values of at least 0.005 for the ITS dataset (minimal barcode gap considered as default by ASAP) and 0.001 for the less variable cpATR dataset; (5) would have a high probability rank and all the suggested species also with a high probability rank, and (6) would have the lowest summarized ASAP score among all proposed schemes which meet criteria 1–5. In addition, within and between group nucleotide uncorrected p-distances in the ITS and cp ATR datasets were estimated within the genus *Pseudohygrohypnum* using Mega X (*Kumar et al., 2018*). Analyses were conducted using the Maximum Composite Likelihood model (*Tamura, Nei & Kumar, 2004*) for the 64 *Pseudohygrohypnum* accessions in both the nr ITS and cp ATR datasets.

Divergence time estimates were calculated in BEAST 1.10.4 (*Suchard et al., 2018*) based on a specially designed exATRKRN dataset (73 accessions, 6,819 sites without indel coding), which added the mitochondrial *nad5* gene intron to the ATRKR dataset. In this dataset, Pylaisiaceae was represented by 29 accessions (with a single accession of Pylaisiaceae per species/molecular lineage) and were supplemented by representatives of Leskeaceae, Amblystegiaceae, Brachytheciaceae, and other lineages of crown Hypnales for which fossil records were available (*Laenen et al., 2014*). Accessions of *Enrothia polyclada* (Müll. Hal.) Ignatov & Fedosov, *Homalia trichomanoides* (Hedw.) Brid. and *Leptodon smithii* (Dicks. ex Hedw.) F. Weber & D. Mohr were included as outgroup, used for rooting. The analysis was run for 100 million generations and the Yule model was set as the tree prior. The uncorrelated lognormal relaxed clocks were set for all partitions individually. Calibration points of five fossil records from Baltic and Dominican amber are shown in Supplemental Materials 3 (See *Huttunen et al., 2008* for discussion). We checked the convergence of the

runs and that the ESS values were all above 200 by exploring the likelihood plots using Tracer v1.5 (*Rambaut et al., 2018*). The initial 10% of trees were discarded as burn-in. A lineage through time plot was generated in Tracer using log and tree outputs from BEAST.

## Morphological methods

The molecular phylogenetic study was complemented by a morphological revision of subaquatic *Pseudohygrohypnum* species in MW, MHA, LE, NSK, CBFS, and MO. In addition to standard microscopic observations, peristomes of selected specimens were studied by the Scanning Electron Microscope Jeol 6380, coated by gold without any additional preparation. Light microscope observations were made under a Zeiss Axioplan Imaging microscope, images were prepared using the 3D digital microscope Hirox RH-2000.

## Distribution modelling methods

To account for similarities/differences in species distribution in terms of their ecology and formalize it within the integrative taxonomic framework, we employed species distribution modelling using the MaxEnt 3.4.1 software package (*Phillips, Anderson & Schapire, 2006*; *Phillips & Dudík 2008*), followed by niche identity and similarity/divergence tests.

Revised georeferenced specimens from MW, MHA, LE, NSK, and MO supplemented by occurrences with coordinates with precision better than 0.01° downloaded from *GBIF.org (2021)*, the Moss Flora of Russia (*Ivanov et al., 2017*) database and literature data were used as an input for distribution modelling. Species distribution models (SDMs) were obtained for each lineage, revealed by phylogenetic analyses with more than 15 available occurrences. Based on the revealed distribution pattern of the molecularly defined lineages, GBIF occurrences were used only for three molecular lineages, "*eugyrium* 1", "*eugyrium* 2" and "*subeugyrium* 1" (see Results section), which are well differentiated from other lineages geographically, for their European and East North American localities, since occurrences from other areas were assumed to represent different lineages. In this way, accessions of *P. eugyrium* from Europe were automatically referred to the lineage A, while those from North America were referred to lineage B. Theoretically, misidentifications between *P. subeugyrium* s.str. and *P. eugyrium* lineage A in Europe and lineage B in North America may influence the results of modelling, but their confusion is not very common except for older American specimens of *P. subeugyrium*, which are often kept under the name *P. eugyrium*. To account for this possibility, we excluded *P. eugyrium* specimens collected before 1970.

Seven non-correlated bioclimatic layers were chosen as variables for modelling (Supplemental Materials 4). The input bioclimatic layers were downscaled to 2.5 arc second resolution and restricted to North America and Eurasia in QGIS 3.16.1 (*QGIS.org, 2021*) and ArcGIS 10.3 (ESRI Inc.) software. Since herbarium data are often spatially biased, we created a bias correcting file using the biased prior method to estimate the relative search effort through modelling in MaxEnt of all moss herbarium occurrences from the GBIF and Moss Flora of Russia databases as a single target group (*Phillips et al., 2009*; *Merow, Smith & Silander, 2013*; *Fourcade et al., 2014*; *El-Gabbas & Dormann, 2018a*; *El-Gabbas &*

*Dormann, 2018b*). The resulting output layer can be considered to reflect sampling effort and was used for non-uniform weighting of background points during modelling. The spatial k-fold cross-validation approach (*Shcheglovitova & Anderson, 2013*; *Radosavljevic & Anderson, 2014*) implemented in SDMtoolbox 2.4 (*Brown, Bennett & French, 2017*) for ArcGis was used to choose feature classes and B-regularization. It allowed us to choose 10 models from among the 30 with the lowest omission error rate and with maximum AUC. The final models were selected from this set by both Akaike and Bayesian information criteria (*Warren & Seifert, 2011*; *Warren et al., 2014*) implemented in ENMTools 1.4.4 (*Warren, Glor & Turelli, 2010*).

To compare the environmental niches of the revealed *Pseudohygrohypnum* lineages we used the identity and similarity tests based on Schoener's D and the Hellinger's I metrics (*Warren, Glor & Turelli, 2008*) in ENMTools 1.4.4. At first, we tested pairs of revealed lineages for niche identity. This test is used to check whether the models generated from two or more species are more different than would be expected if they were drawn from the same underlying distribution (*Warren, Glor & Turelli, 2008*). For pairs with non-identical niches, we performed the niche similarity background test to determine whether the models drawn from populations with partially or entirely non-overlapping distributions are more different from one another than expected by random chance (*Warren, Glor & Turelli, 2008*). During both tests, 100 iterations of random points (pseudoreplicates) were assigned in two directions for each of the species pairs.

To visualize the environmental niche similarity/difference, box-plots of nine selected bioclimatic variables, along which the divergence was most remarkable, were drawn in GraphPad Prism 8.4.3 (GraphPad software, LLC). Distributions of the eight revealed lineages along these variables were checked for significance of the median difference using the Mann–Whitney U Test in Past v4 (*Hammer, Harper & Ryan, 2001*). Values of seven bioclimatic variables used for modelling in observed localities assigned to the revealed phylogenetic lineages of subaquatic *Pseudohygrohypnum* were then used as an input for PCA in the pca3d package for R (*R Core Team, 2021*).

The electronic version of this article in Portable Document Format (PDF) will represent a published work according to the International Code of Nomenclature for algae, fungi, and plants (ICN), and hence the new names contained in the electronic version are effectively published under that Code from the electronic edition alone. The online version of this work is archived and available from the following digital repositories: PeerJ, PubMed Central SCIE, and CLOCKSS.

# RESULTS

## Molecular phylogenetic analyses

The topology obtained from the concatenated plastid data matrix differs at several nodes from the topology inferred from nuclear ITS data (Fig. 1). Plastid data support the monophyly of *Pseudohygrohypnum* as defined by *Kučera et al. (2019)* (PP1/0.99, BS73/58, see caption to Fig. 1), while ITS data render *Pseudohygrohypnum* paraphyletic, forming a grade of lineages basal to the crown Pylaisiaceae, containing accessions of

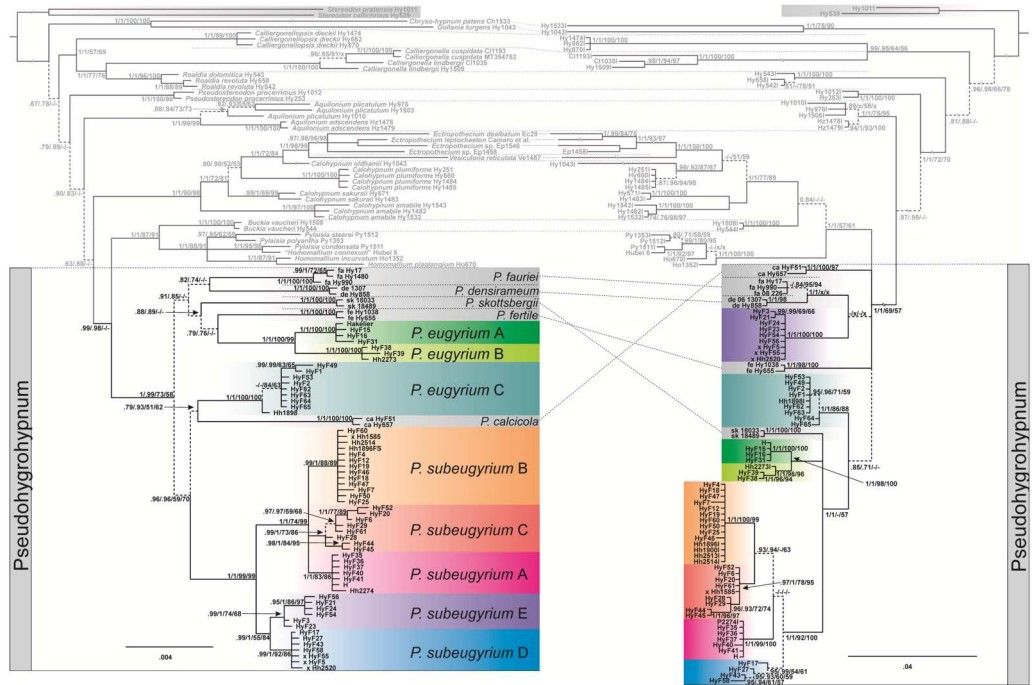

**Figure 1  Bayesian trees inferred from cp ATRKR (left side) and nr ITS (right side) datasets.** Statistical support indicating Bayesian posterior probabilities (PP) and maximum-likelihood bootstrap support (BS) inferred from matrices without (1) and with (2) indel coding is provided at the branches in the order PP1/PP2/BS1/BS2; a dash (-) indicates no support (PP < 0.7 and BP < 50). Dashed line indicates not or weakly supported clades.

*Pylaisia, Homomallium, Buckia, Calohypnum, Ectropothecium,* and *Vesicularia.* Subaquatic representatives of *Pseudohygrohypnum* appear in eight lineages interspersed among non-aquatic species; the relationships among these lineages also differ between the plastid and nuclear datasets. Plants with a differentiated stem epidermis (hitherto named *P. eugyrium*) appear in three fully supported lineages, with the European (lineage A) and eastern North American (lineage B) lineages forming together a fully supported monophyletic clade of amphi-Atlantic plants, while the Far Eastern lineage C occurs well outside of this clade and instead among other clades composed of subaquatic *Pseudohygrohypnum* species. Plants without a differentiated epidermis (hitherto named *P. subeugyrium* and *P. purpurascens* if recognized from the former) form a fully supported monophyletic clade based on plastid data with five recognizable internal lineages which mostly also receive full support, however, based on ITS data, one of these internal lineages (lineage E) appears outside of the clade containing the rest of the *P. subeugyrium* s.lat. lineages.

Four *P. subeugyrium* s.l. accessions have different clade assignments in the nr- and cp-based topologies (marked x in the tree in Fig. 1). One accession of *P. subeugyrium* clade C in the ITS tree appears in clade B of the cp-based tree, and three accessions of clade E based on ITS data occur in clade D of the plastid tree.

NN split networks (Fig. 2) reveal the same entities within the subaquatic *Pseudohygrohypnum* species. Although they are less divergent than the currently accepted mesophytic *Pseudohygrohypnum* species included in the analysis, these entities are nearly identical in composition in the networks constructed from nuclear and plastid markers (excepting the four aforementioned specimens), and have reliable support inferred from both the plastid and nuclear datasets or only from the nuclear dataset. Mean nucleotide p-distances between identified lineages in most cases are much higher than the mean within-group p-distances. The latter are mostly below 0.001 in the cp dataset and 0.005 in the ITS dataset, while between-group distances exceed 0.003 in the the cp dataset and 0.010 in the ITS dataset (Supplemental Materials 5). The smallest between-group distances are observed between *P. eugyrium* lineages A and B; and *P. subeugyrium* lineages A, B, C, and in the cp dataset also D and E. Although the mean distances between *P. eugyrium* lineages A and B and *P subeugyrium* lineages D and E are low according to cp data, they are rather high in the ITS dataset.

Among the 10 best schemes proposed by ASAP for dividing the plastid dataset into species, the 13 species scheme which recognizes all letter-named lineages as separate species in addition to the already existing ones (Supporting Figs. 1, 2) has the second-best score after an unrealistically oversplitting 22-species scheme. Two other schemes, which agree with most criteria specified in the Methods section, a 12-species scheme (with *subeugyrium* lineage D and E merged) and an 8-species scheme (*subeugyrium* lineage A-E merged) have worse scores and contradict the topology of the ITS based phylogenetic trees, where *subeugyrium* lineage E is found well outside of the *P. subeugyrium* clade. Among the 10 best schemes proposed for the ITS dataset, only the 13-species scheme was shared with the best-scoring schemes based on cp data. Although the 13-species scheme was not proposed as an optimal one for the ITS dataset, the similar 12 species scheme with *P. subeugyrium* C and D merged has a highest score among schemes proposed for ITS. However, the topology of the phylogenetic trees precludes further consideration of this 12-species scheme. Thus, a 13-species scheme is the only one, which meets the criteria specified in the methods section and is justified as a working hypothesis supporting specific status for *P. purpurascens* and further splitting of the traditional concepts of *P. eugyrium* and *P. subeugyrium* into 7 species based on molecular data.

The topology of the tree, inferred from the expanded dataset, designed for dating *Pseudohygrohypnum* diversification, is well resolved and statistically supported excepting a few deep nodes (Fig. 3). Topology within the Pylaisiaceae clade is congruent with the one, obtained from the plastid dataset (Fig. 1, left side), but demonstrates higher support for the clades within the genus *Pseudohygrohypnum*. The *Pseudohygrohypnum* clade splits into two weakly supported clades, first of which accommodates *P. fauriei*, *P. fertile*, *P. skottsbergii*, *P. eugyrium* A and B while the second one comprises *P. eugyrium* C, *P. calcicola*, and *P. subeugyrium* s.l. According to the divergence dates, inferred based on the expanded dataset, the split of two major clades of *Pseudohygrohypnum* is estimated at 28.96 (23.93–34.44) Ma. Splitting of the *P. eugyrium* C lineage from its closest relative, *P. calcicola* is estimated at 21.26 (15.58–26.8) Ma. Average estimates for splits of the mesophilous *Pseudohygrohypnum* lineages vary and are 26.78 Ma for *P. fauriei* (the rather closely related *P. densirameum* was

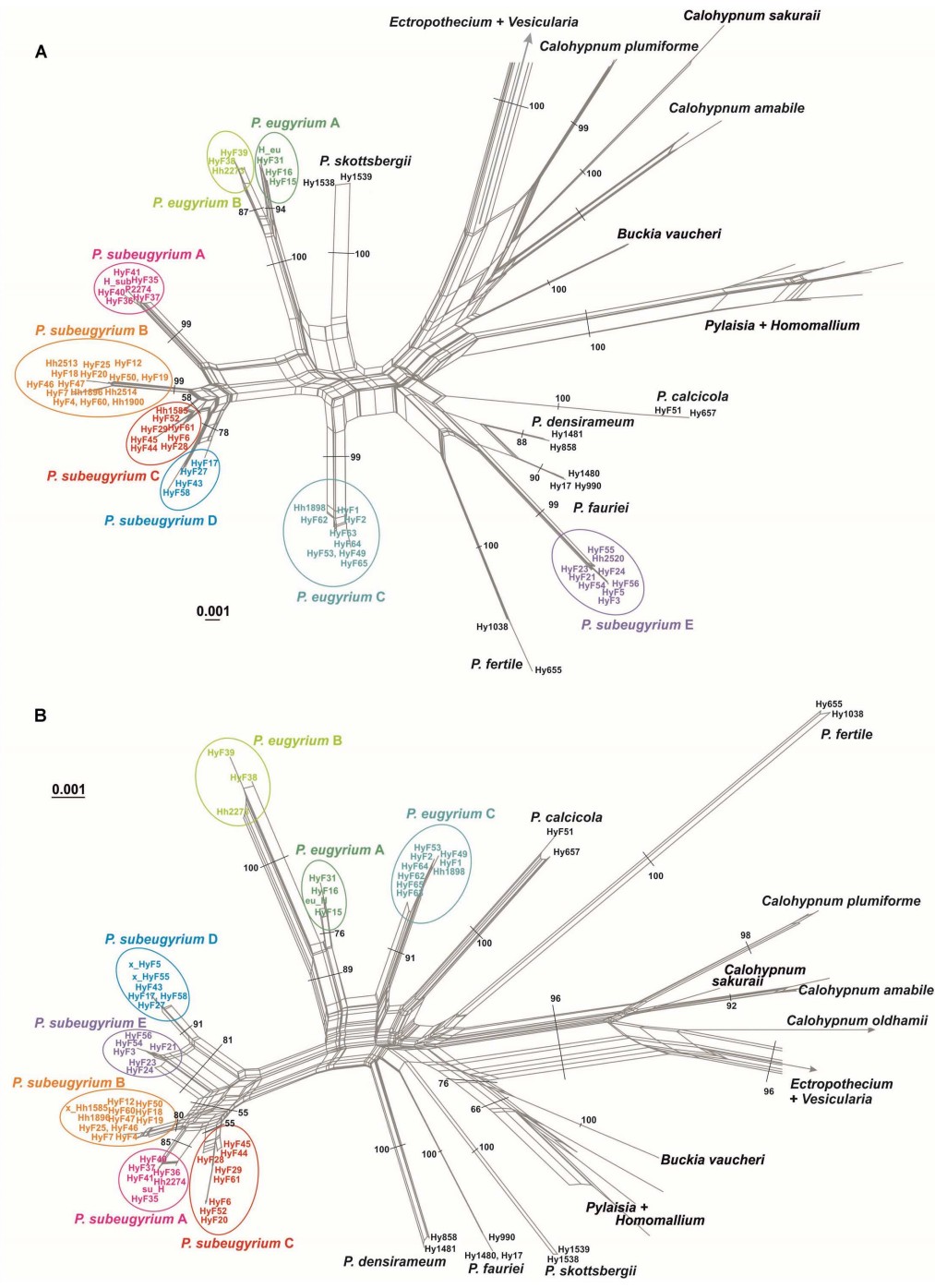

**Figure 2** **Split trees of the crown clade of Pylaisiaceae, originated from the nr ITS (A) and cp ATR (B) datasets.** Colors indicate revealed lineages of subaquatic *Pseudohygrohypnum*. Bootstrap values are indicated at branches corresponding to genus- and/ or species level. The bootstrap values appeared from 1,000 iterations of the bootstrap analysis in Splitstree 4.

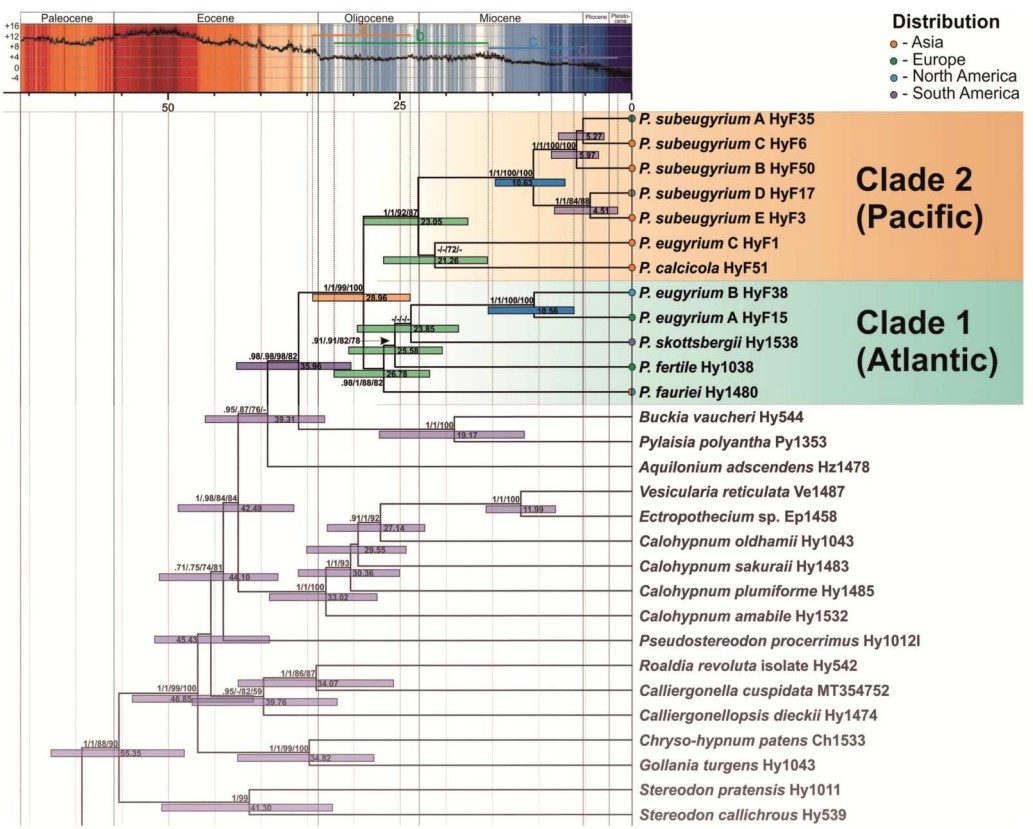

**Figure 3** **Crown part of bayesian tree inferred from expanded cp & mt dataset for Pylaisiaceae with nodes dated by Beast.** Temperature course is indicated by the black curve and also by colour-filling above according to *Westerhold et al. (2020)*. Coloured bars in the temperature curve indicates estimated periods of basal radiation of the genus *Pseudohygrohypnum* (A), origin of mesophilous species of the genus (B), basal radiation of subaquatic lineages of the genus (C), further radiation of *P. subeugyrium* complex (D) in relation to temperature.

not included in this dataset), 25.58 for *P. fertile* and 23.85 Ma for *P. skottsbergii*. Splitting of the two Atlantic lineages of *P. eugyrium* is estimated to have occurred 10.56 (6.25–15.5) Ma, and the basal radiation of *P. subeugyrium* at 10.63 (7.22–14.75) Ma.

## Morphological analyses

The overall morphology and leaf anatomy of subaquatic mosses is often strongly affected by the hydrological regime of their habitat and this is especially true for *Pseudohygrohypnum*. However, our morphological survey has shown that the eight above-described molecularly defined lineages are also reasonably distinct morphologically (see Table 1 and Figs. 4–7) and therefore ought to be recognized as distinct species. In addition to diagnostic characters, plants previously referred to *P. subeugyrium* differ in their habit (Fig. 4). *Pseudohygrohypnum subeugyrium* lineages A and C also differ in plant coloration: lineage A (*P. subeugyrium* s.str.) is typically represented by olivaceus to brownish plants, while plants of lineage C are typically purplish to vinaceous. With its typically complanate shoots and weakly falcate, nearly plane and acute leaves, plants from lineage B resemble *Hygrohypnum*
**Table 1  Morphological differentiation of the revealed subaquatic *Pseudohygrohypnum* lineages.**

| Trait | Eugyrium A (*P. eugyrium* s.str.) | Eugyrium B (*P. appalachianum*) | Eugyrium C (*P. orientale*) | Subeugyrium A (*P. subeugyrium* s.str.) | Subeugyrium B (*P. sibiricum*) | Subeugyrium C (*P. purpurascens*) | Subeugyrium D (*P. neglectum*) | Subeugyrium E (*P. subarcticum*) |
|---|---|---|---|---|---|---|---|---|
| Coloration | Bright green, pinkish to reddish or blackish | Bright green | Light green | Olivaceous green, brownish rarely reddish | Green, dirty yellow to brownish or reddish | Variegated, green/purple to deep vinaceous | Variegated, green/pinkish to purple | Bronze with golden sheen |
| Stem central strand | present | present | present | absent | present | absent | absent or indistinct | absent |
| Stem hyalodermis | Differentiated (nearly) throughout | Differentiated throughout or in patches | Differentiated in patches to not differentiated | Not differentiated | Not differentiated, rarely differentiated in patches | Not differentiated | Not differentiated | Not differentiated |
| Leaf shape | Ovate to ovate-lanceolate, acute to apiculate | Ovate to ovate-lanceolate, apiculate | Ovate to ovate-lanceolate, apiculate | Ovate-lanceolate, acute rarely blunt | Ovate to ovate-lanceolate, blunt, acute to short acuminate | Ovate-lanceolate, rarely ovate, blunt, rarely acute | Ovate-lanceolate, acute to short acuminate | Lanceolate, rarely ovate-lanceolate, blunt |
| Leaf falcateness | Not or slightly falcate | Not or slightly falcate | Distinctly falcate | Slightly to strongly falcate | Not or slightly falcate | Slightly to strongly falcate | Distinctly falcate | Strongly falcate |
| Leaf length (mm) | 1.2–1.6(−1.8) | (1.3-)1.5–1.8 | (0.9)1–1.6(−2) | 1.2–1.75 | (1.2-)1.3–1.6 (−1.8) | 1.2-1.8(−2) | (1.3-)1.4–1.8(2.0) | (0.65-)0.7–0.9(−1) |
| Leaves widest at | 1/5–1/2 | 1/5-1/2 | (1/5-)1/3-1/2 | 1/7–1/5(−1/3) | 1/5–1/3 | 1/7–1/5(−1/3) | 1/7–1/3 | 1/9–1/5 |
| Leaf concavity | Strongly concave | Strongly concave | Strongly concave | Concave, canaliculate distally | Weakly concave | Concave, canaliculate distally | Slightly to moderately concave | Concave, canaliculate distally |
| Costa extension | 1/4–1/2 | 1/4—1/2 | 1/4—1/2 | 1/5 | 1/4—1/2 | 1/5 | 1/7–1/5 (-1/3) | 1/10–1/7 |
| Margin serrulation | At tip | Nearly entire or at tip | At tip to upper 1/3 | At tip to upper 1/3 | At tip to upper 1/2 | At tip to upper 1/3 | 1/2 | Nearly to the base |
| Leaf cells | 30–62 μm flexuose | 45–75 μm flexuose | 30–50(-65) μm, not flexuose | 40–105 μm flexuose | 35–70 μm, slightly flexuose | 40–105 flexuose | 40–75 flexuose | 50–80 not flexuose |
| Alar group differentiation | Strongly differentiated, Rounded, or transverse elongate, usually not reaching costa, inflated | Strongly differentiated, Rounded, not reaching costa, strongly inflated | Moderately differentiated, rounded, or transverse elongate, usually not reaching costa, weakly inflated, | Moderately to strongly differentiated, transverse elongate, weakly delimited proximally, usually not reaching costa, inflated | Moderately differentiated, rounded, or transverse elongate, usually not reaching costa, inflated or not | Moderately to strongly differentiated, transverse elongate, weakly delimited proximally, reaching costa, inflated | Strongly differentiated, rounded, not reaching costa, inflated | Small, weakly delimited, not inflated |
| Alar cells shape | Large, thick-walled brown, rarely hyaline | Large, thin-walled, hyaline to slightly colored | Large, thin-walled, hyaline to thick walled, brownish | Large, thick-walled with brown walls | Large, Thin-walled, hyaline to thick walled, brownish | Large, thick-walled with brown walls or hyaline | Large, thin-walled, hyaline to slightly colored | Small, quadrate, thick walled, brownish |
| Operculum | conic | conic | conic | conic | conic | conic | conic | conic-rostrate |
| Spore size (μm) | 12–20 | 14–23 | 17–25 | 12–20 | 13–20 | 15–30 | 12–18 | 15–19 |
| Distribution | Europe, Caucasus? | East North America | East Asia | North Europe, East North America | South Ural, Central & East Asia | East Asia | NE Asia & E North America | North Siberia |

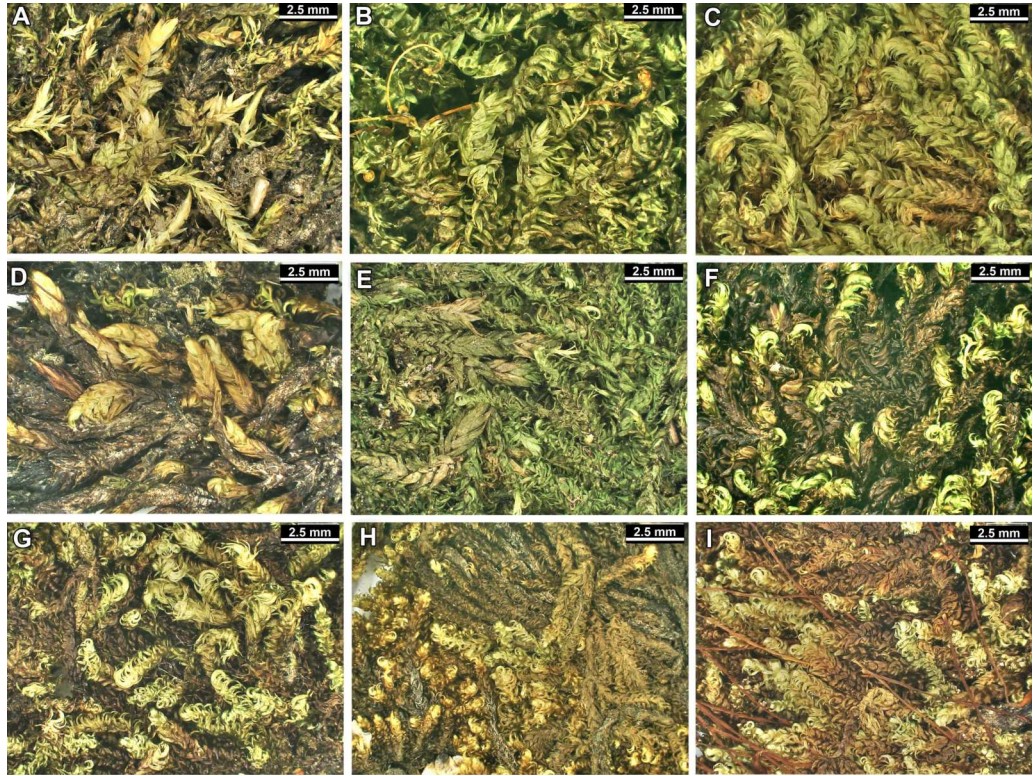

**Figure 4** **Habit of plants from the revealed lineages of subaquatic *Pseudohygrohypnum*.** (A–C) *P. eugyrium* lineages A–C correspondingly, (D–G) *P. subeugyrium* lineages A-D correspondingly, (H) recombinant plant *P. subeugyrium* D ×*P. subeugyrium* E, (I) *P. subeugyrium* lineage E.

*luridum* (Hedw.) Jenn., Asiatic morphotypes of *Calliergonella lindbergii* (Mitt.) Hedenäs, or *Stereodon pratensis* (W.D.J. Koch ex Spruce) Warnst. At the same time, specimens of the lineage B from Primorsky Territory resemble plants from the lineages C and D in habit, although differing in having a well-developed central strand and different alar group shape. Plants of the molecularly closely related lineages D and E are quite dissimilar morphologically. While lineage E resembles *Campylium bambergeri* (Schimp.) Hedenäs, Schlesak & D. Quandt in habit having narrow and strongly falcate leaves and differing from it and all other *Pseudohygrohypnum* species in its much smaller size, plants of lineage D are rather large and resemble *Calliergonellopsis dieckii* (Renauld & Cardot) Jan Kučera & Ignatov. In most cases, microscopic examination allows identification of the newly recognized species, although we observed problematic specimens which we discuss in the Taxonomy section below.

## Distribution modelling analyses

Phylogenetic lineages of subaquatic *Pseudohygrohypnum* as revealed by our analyses are quite distinct geographically (Fig. 8). Results of the pairwise comparison using the Mann–Whitney U Test (Supplemental Materials 7) suggest significant differences in their distribution along the nine analysed bioclimatic variables. Only for one pair of lineages

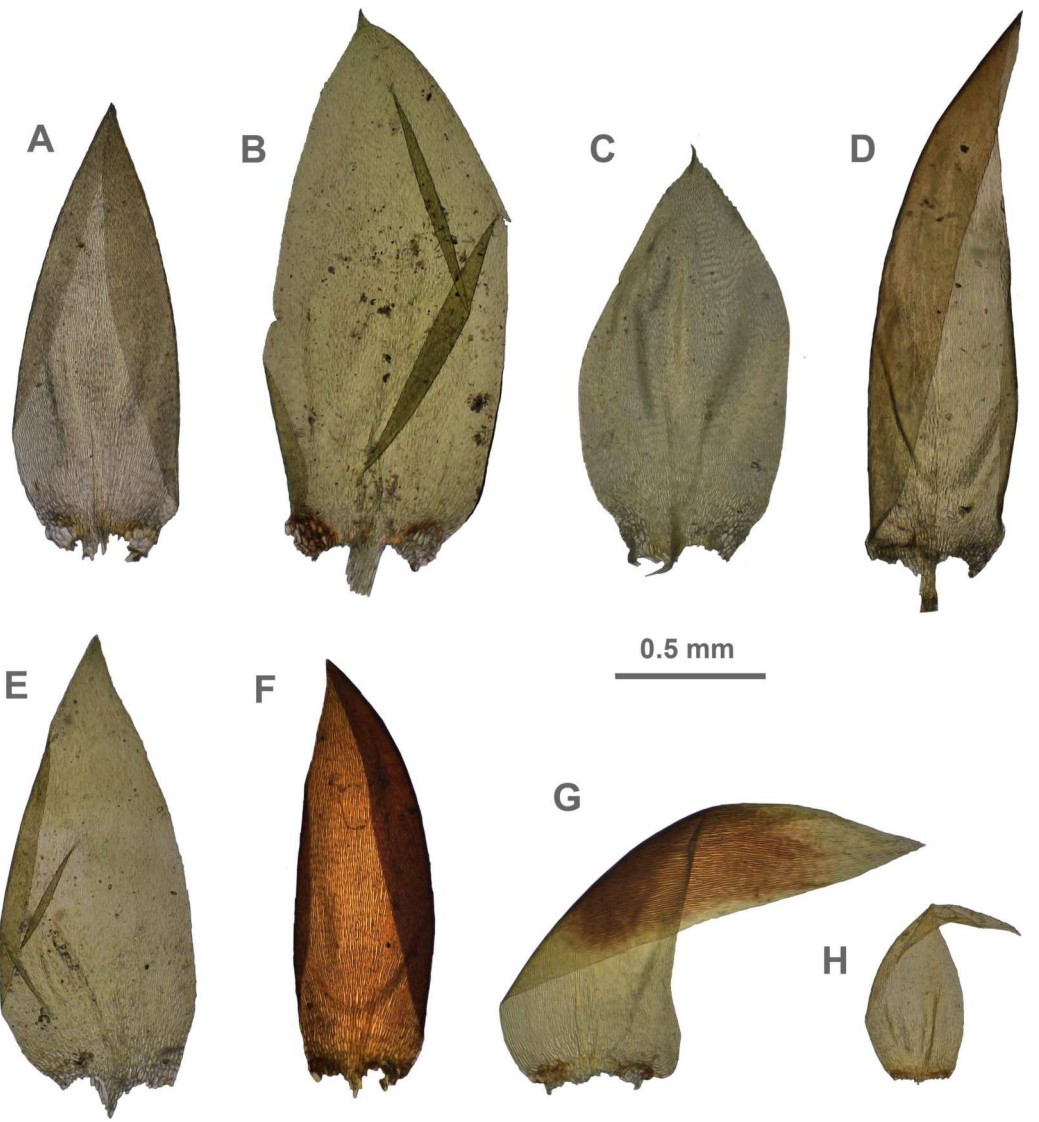

**Figure 5** **Stem leaves in plants from the revealed lineages of subaquatic *Pseudohygrohypnum*.** (A–C) *P. eugyrium* lineages A–C correspondingly, (D–H) *P. subeugyrium* lineages A–E correspondingly.

(*eugyrium* C and *subeugyrium* C) no significant difference of medians was revealed, while in all other pairs significant median difference was suggested for at least four variables. All three lineages previously assigned to *P. eugyrium* occur exclusively in mild climatic conditions influenced by oceanic air masses. Observed occurrences of these species originate exclusively from the areas with BiO1 above 0 °C (Fig. 9). The same is true for two species of the *P. subeugyrium* complex, so far referred to as lineages A and C. Contrary to these five, three lineages recognized within *P. subeugyrium* s.l. generally occur in colder environments, largely associated with a mean annual temperature below 0 °C. Plants combining ITS and cp loci characteristic for different lineages of *P. subeugyrium* largely occur in areas with conditions suitable for both lineages. BiO2 can be considered as a proxy for air moisture;

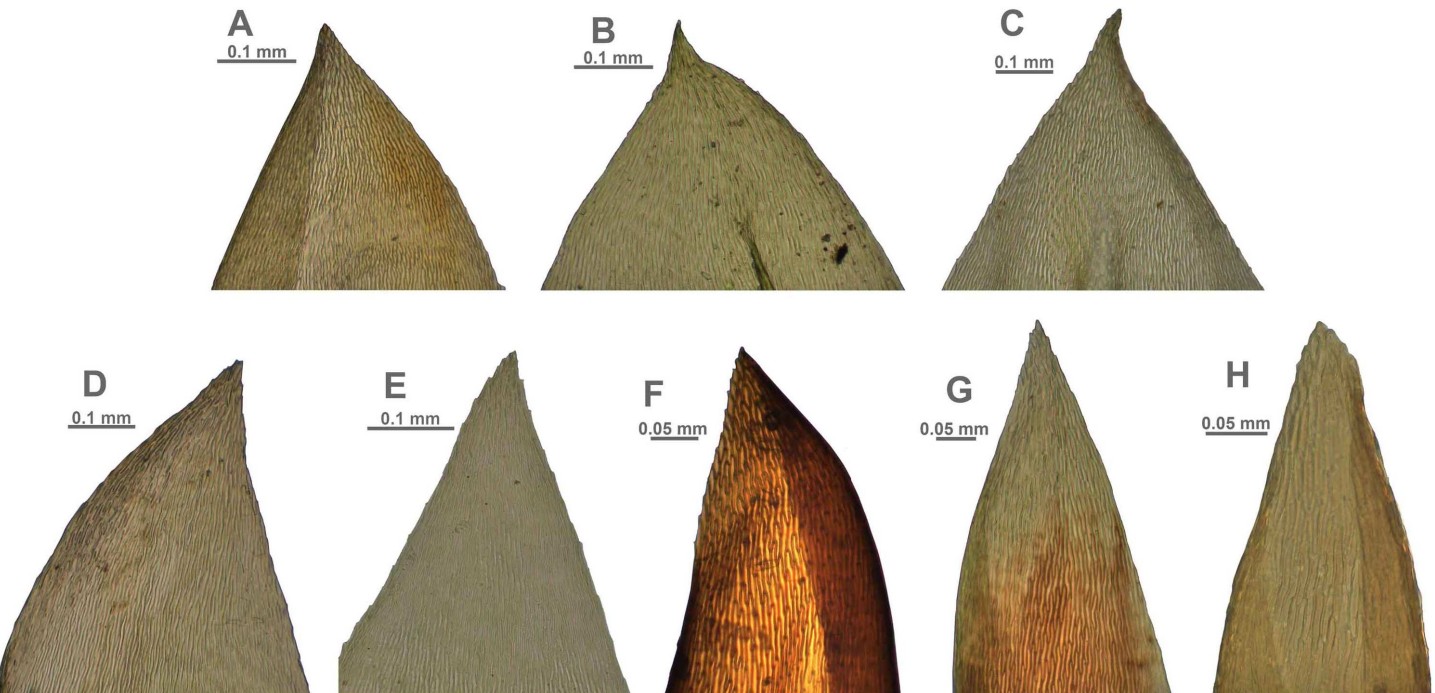

**Figure 6  Stem leaf tips in plants from the revealed lineages of subaquatic *Pseudohygrohypnum*.** (A–C) *P. eugyrium* lineages A–C correspondingly, (D–H) *P. subeugyrium* lineages A–E correspondingly.

occurrences of "continental" lineages B and D of *P. subeugyrium* generally are associated with higher BiO2 values, but it is rather low in continental lineage E and high in *P. eugyrium* lineage B. Likewise, BiO7 indicates temperature seasonality; occurrences of *P. subeugyrium* lineages B, D and E are confined to areas with rather high temperature seasonality, while *Pseudohygrohypnum eugyrium* as well as *P. subeugyrium* lineages A and C inhabit areas with lower BiO7 values; European and American lineages of *P. eugyrium* are noticeably different in their relation to BiO7.

The same two groups, "suboceanic" (*eugyrium* A-C, *subeugyrium* A, C) and "continental" (*subeugyrium* B, D, E) appeared in relating the 8 subaquatic *Pseudohygrohypnum* lineages to the annual rate of precipitation (BiO12), but the distribution of *P. subeugyrium* lineage D along this gradient overlaps with *P. subeugyrium* lineage A, rather than with lineage E. Two East Asian lineages, *P. eugyrium* C and *P. subeugyrium* C occur in areas with higher precipitation rates than the Atlantic suboceanic lineages (Fig. 9). The three Atlantic lineages are confined to areas with low seasonality of precipitations (BiO15); north Siberian and two East Asian lineages occur under much higher precipitation seasonality, and *P. subeugyrium* clades B and D, which occur in inland boreal areas of Asia, grow under even higher BiO15 values. Precipitation of the driest quarter (BiO17) in inland north Asian areas inhabited by *P. subeugyrium* B, D and E is significantly lower than in the areas where other lineages of subaquatic *Pseudohygrohypnum* occur. At the same time, precipitation of the warmest quarter (BiO18) in the ranges of East Asian species *P. eugyrium* C and *P. subeugyrium* C exceeds areas where the other lineages occur while

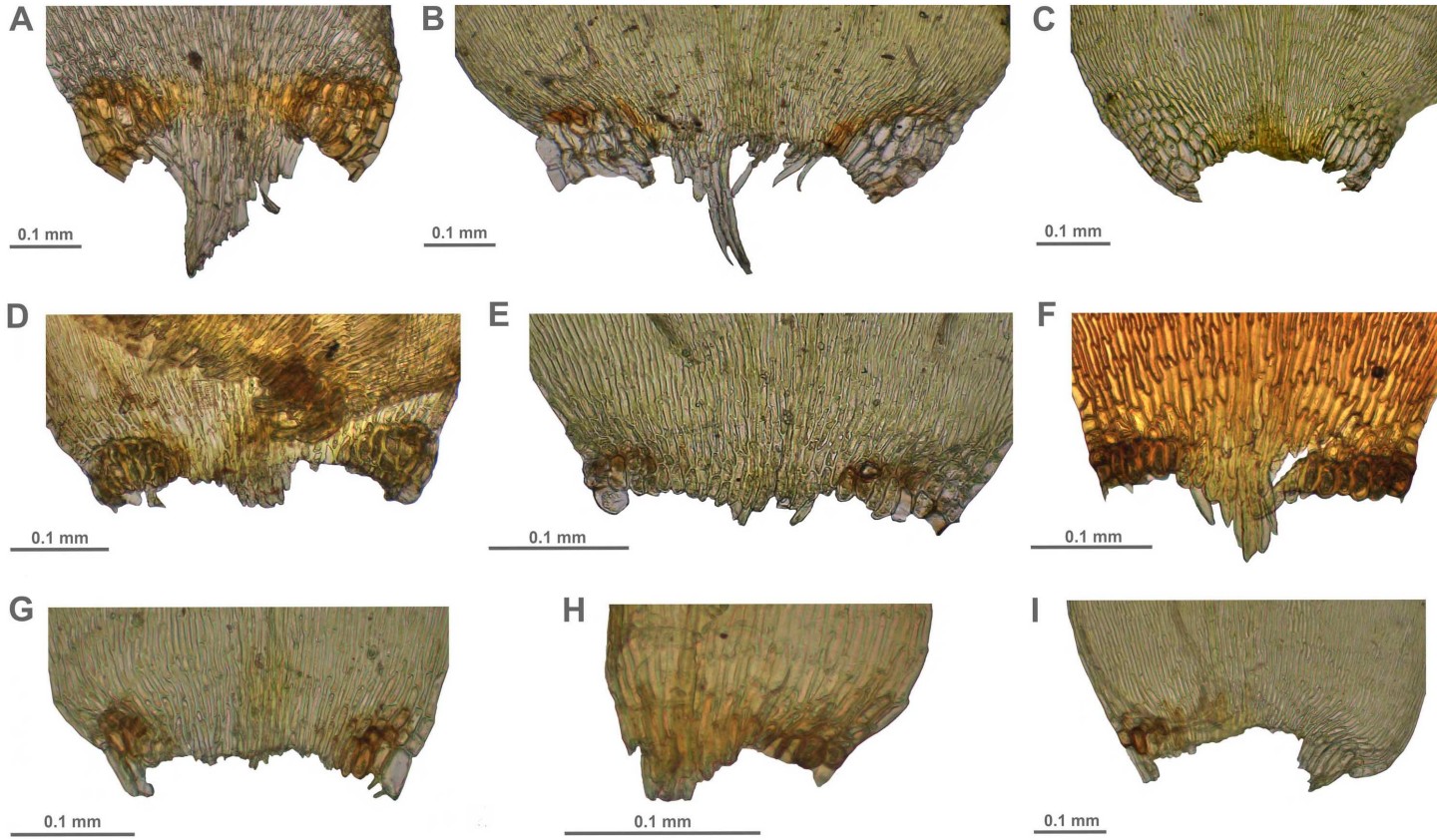

**Figure 7** **Stem leaf base in plants from the revealed lineages of subaquatic *Pseudohygrohypnum*.** (A–C) *P. eugyrium* lineages A–C correspondingly, (D–G) *P. subeugyrium* lineages A–D correspondingly, (H) recombinant plant *P. subeugyrium* D ×*P. subeugyrium* E, (I) *P. subeugyrium* lineage E.

subarctic *P. subeugyrium* lineage E survives in the driest summer conditions. Mean monthly precipitation in the coldest quarter (BiO19) is rather high in the areas where suboceanic *P. eugyrium* A-C and *P. subeugyrium* A, C lineages occur, and much lower within the ranges of continental *P. subeugyrium* lineages B, D and E. Noteworthy, according to an aridity index, lineage B is associated with even drier areas than subarctic lineage E, but at the same time may occur in the same areas where two East Asian subaquatic *Pseudohygrohypnum* species occur (Fig. 9).

Statistics of SDMs obtained for seven of the eight phylogenetic lineages of *Pseudohygrohypnum* are summarized in Supplemental Materials 6. Models obtained for these seven lineages have reliable quality and largely meet the criteria for bronze-silver standards, introduced by *Araújo et al. (2019)*. The final SDMs for these species demonstrate a "good" quality with AUC > 0.960 (only for *P. subeugyrium* lineage B model AUC = 0.915) and with a training omission rate under maximum sensitivity and specificity thresholds from 0.00 to 0.056. Although identical sets of predictors were used for modelling, SDMs of the individual lineages differ in the contributions of these predictors. Both metrics (*i.e.,* contribution and permutation importance) indicate that BiO7 (temperature annual range)
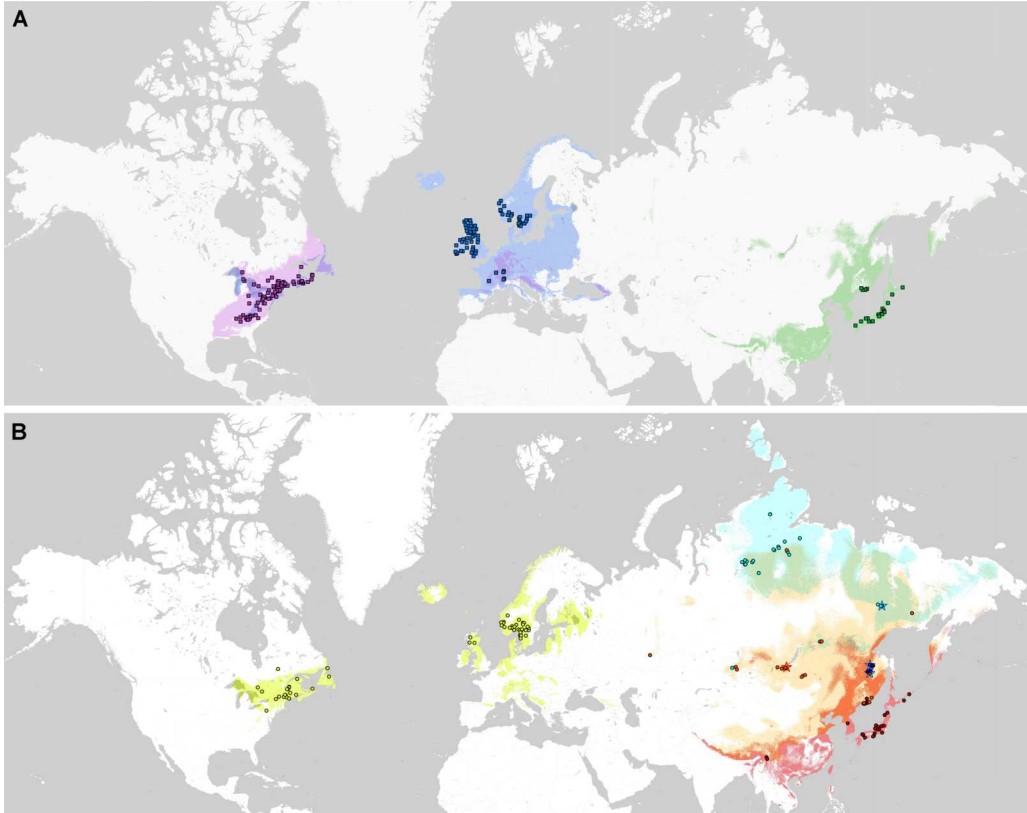

**Figure 8  Distribution of the revealed lineages of subaquatic *Pseudohygrohypnum*.**  upper map –*P. eugyrium* A (blue squares), B (violet squares) and C (green squares); lower map –*P. subeugyrium*  A (lemon circles), B (orange circles), C (red circles), D (dark blue circles), E (pale blue circles) and recombinant specimens *P. subeugyrium* B × C (orange stars) and D × E (pale blue stars). Colour shades indicate distribution of the lineage with concolorous symbols obtained from SDMs.

is the most important for distribution of European *P. eugyrium* (lineage A) and its values are least in this lineage among all subaquatic *Pseudohygrohypnum* species. At the same time, SDM of North American *P. eugyrium* (lineage B) is affected by precipitation seasonality and it is associated with the areas where this predictor has lower values compared to all other lineages. The second-best predictor for this lineage, BiO17 (precipitation in the driest quarter) also has somewhat higher values than in other lineages. SDMs obtained for East Asian lineages of *P. eugyrium* (C) and *P. subeugyrium* (C) are largely controlled by BiO18, which is higher in areas inhabited by these lineages, than in areas where other lineages of subaquatic *Pseudohygrohypnum* occur. SDMs obtained for *P. subeugyrium* lineages A and B do not reveal clearly dominant climatic variables, but in the case of lineage B, the significant variables are those associated with precipitation. Finally, the SDM obtained for *P. subeugyrium* lineage E indicates BiO1 as the most important variable, and its value is much lower in areas inhabited by this lineage compared with other lineages of subaquatic *Pseudohygrohypnum*.

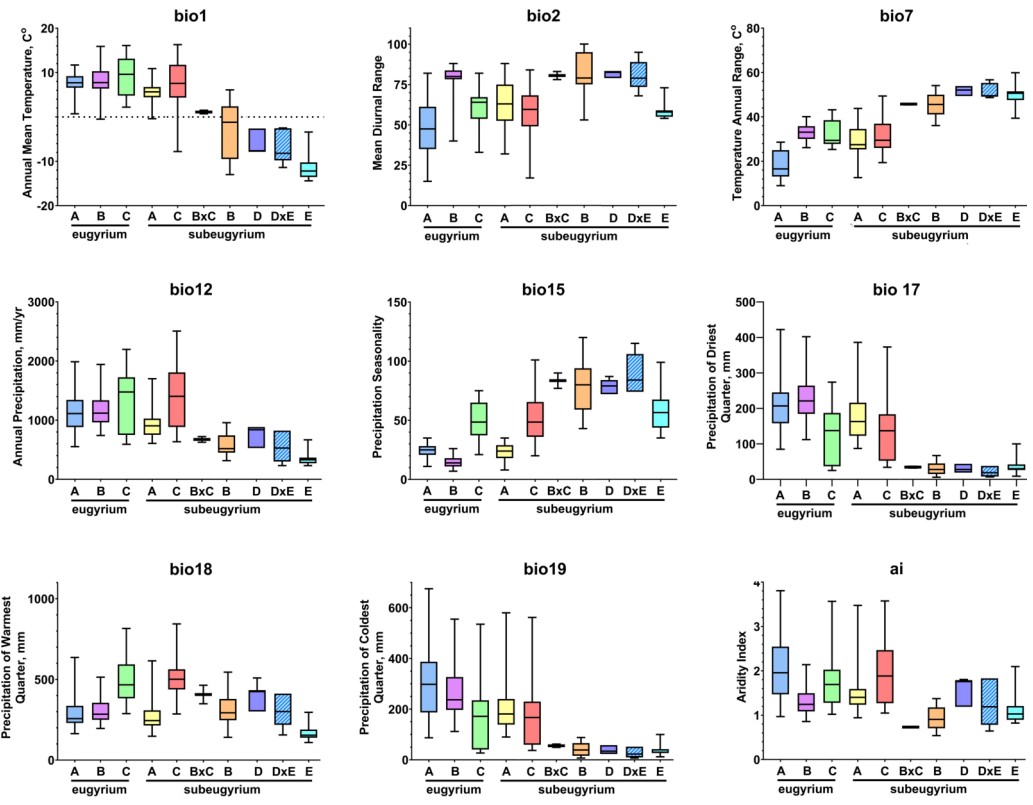

**Figure 9** **Distribution of the revealed lineages of subaquatic *Pseudohygrohypnum* along the nine selected CHELSA bioclimatic variables based on observed occurrences.** Colour legend for the lineages under consideration follows the one in Fig. 8.

PCA suggested clear delimitation of environments where the three lineages so far assigned to *P. eugyrium* occur (Fig. 10). For the five lineages of *P. subeugyrium*, distributions of suboceanic lineages (A and C) and continental lineages (B, D, E) partly overlap. According to results of identity tests *via* I and D metrics, nearly all species pairs have non-identical niches ($p < 0.01$), excluding the *P. eugyrium* lineage C—*P. subeugyrium* lineage C pair; both tests found their distributions identical (p below 0.01).

Niche similarity tests (Table 2) revealed significant similarity (see MM section for details) for the species pairs *P. eugyrium* B and *P. eugyrium* C ($I = 0.54$, $D = 0.25$), *P. eugyrium* B and *P. subeugyrium* C ($I = 0.52$, $D = 0.24$), *P. eugyrium* B and *P. subeugyrium* A ($I = 0.72$, $D = 0.46$), *P. eugyrium* C and *P. subeugyrium* C ($I = 0.52$, $D = 0.24$). Contradicting results of "direct" and "reverse" pairwise comparisons were obtained for species pairs of *P. eugyrium* A vs. *P. subeugyrium* A, *P. eugyrium* C vs. *P. subeugyrium* A, *P. eugyrium* C vs. *P. subeugyrium* B, P. subeugyrium A vs. C, *P. subeugyrium* C vs. E which might be assessed as rather similar. For the rest (more than half) of the lineage pairs, significant niche divergence was suggested.

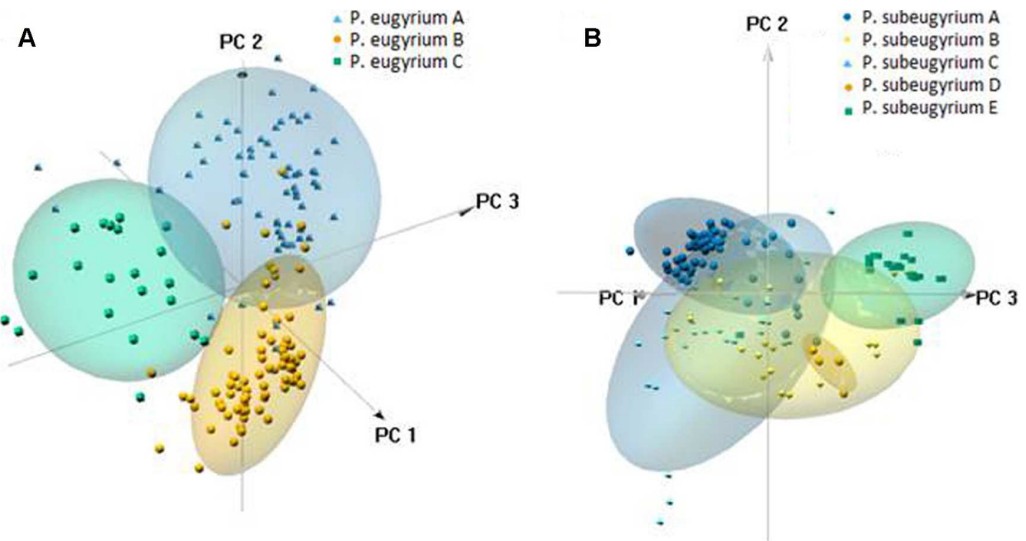

**Figure 10** 3-D PCA scatterplots of the observed occurrences (small symbols) and those predicted by model under 80% threshold (clouds) based on their distribution along the CHELSA bioclimatic variables used for modelling for *P. eugyrium* A–C (A) and P. subeugyrium A-E (B).

## DISCUSSION

### Taxonomic implications of the molecular phylogenetic results

Topologies of the phylogenetic trees, obtained during the present study generally do not contradict previously published large-scale phylogenies (*Gardiner et al., 2005*; *Cox et al., 2010*; *Huttunen et al., 2012*; *Câmara et al., 2018*; *Schlesak et al., 2018*; *Kučera et al., 2019*). As an exception, in the trees by *Gardiner et al. (2005)* inferred from combined ITS and trnL-trnF sequences, *Buckia vaucheri* (Lesq.) D. Ríos, M.T. Gallego & J. Guerra groups with *Pseudohygrohypnum eugyrium* not with *Pylaisia polyantha* (Hedw.) Schimp., although with a low support. The highly supported grouping of *Roaldia revoluta* (Mitt.) P.E.A.S. Câmara & Carv.-Silva and *Pseudostereodon procerrimus* (Molendo) M. Fleisch. in the same tree likely originates from misidentification.

Molecular data retrieved from specimens traditionally assigned to *Pseudohygrohypnum eugyrium* and *P. subeugyrium* as reported above were found to be congruent with the morphological heterogeneity observed in our initial morphological studies. The three molecularly suported lineages of *P. eugyrium* and five lineages of *P. subeugyrium*, recognized also by means of automatic partitioning based on genetic distances, can be characterized morphologically and their distribution patterns were found to be distinct in occupying mostly non-overlapping climatically defined niches. We therefore propose recognizing these lineages as eight species, five of which are newly described in the Taxonomy section. The need for description of a new species for the East Asian *P. eugyrium* lineage C is obvious on both molecular and morphological grounds, while the degree of niche overlap and metrics of niche similarity are nearly the same between the morphologically and molecularly closer *P. eugyrium* lineages A and B, as between the less similar and

**Table 2  Results of niche similarity tests.** Black values - similarity lower than random red values - similarity higher than random orange values - in one direction similarity is higher than random, while in second lower. "D" and "I" columns indicate Schoener's D and Hellinger's I metrics.

| Species pair | I | D |
|---|---|---|
| EUGYRIUM GROUP | | |
| *eu A–eu B* | 0.51 | 0.18 |
| *eu B–eu C* | 0.54 | 0.25 |
| *eu A–eu C* | 0.30 | 0.08 |
| SUBEUGYRIUM GROUP | | |
| *seu A–seu B* | 0.05 | <0.01 |
| *seu A–seu C* | 0.47 | 0.22 |
| *seu A–seu E* | <0.01 | <0.01 |
| *seu B–seu C* | 0.68 | 0.40 |
| *seu B–seu E* | 0.66 | 0.35 |
| *seu C–seu E* | 0.35* | 0.12 |
| EUGYRIUM –SUBEUGYRUM | | |
| *eu A–seu C* | 0.34 | 0.09 |
| *eu A–seu B* | <0.01 | <0.01 |
| *eu A–seu E* | <0.01 | <0.01 |
| *eu A–seu A* | 0.70 | 0.36 |
| *eu B–seu C* | 0.52 | 0.24 |
| *eu B–seu B* | 0.10 | <0.01 |
| *eu B–seu E* | 0.04 | <0.01 |
| *eu B–seu A* | 0.72 | 0.46 |
| *eu C–seu C* | **0.98** | **0.83** |
| *eu C–seu B* | 0.60 | 0.34 |
| *eu C–seu E* | 0.26 | 0.07 |
| *eu C–seu A* | 0.46 | 0.21 |

phylogenetically rather distant lineages A *vs.* C, and B *vs.* C. On the other hand, it would be theoretically possible to merge the European and North American lineages of *P. eugyrium*, which form a well-supported monophyletic unit, and likewise the topology inferred from the plastid dataset would allow for preserving *P. subeugyrium* s.l. including *P. purpurascens*, *i.e., P. subeugyrium* lineages A-E. However, *P. eugyrium* lineages A and B differ in both morphology and occupied niches, and molecular data suggest a sufficient divergence of these lineages, corroborated by the results from the ASAP analysis. Similarly, a broad circumscription of *P. subeugyrium* as including lineages A-E would become vague due to high variability of diagnostic traits, strongly disjunctive range and obvious niche divergence among them. Since European and North American populations of *P. subeugyrium* s.str. (lineage A) are identical morphologically and have identical sequences they must be maintained as a single species, however the four other lineages are better recognized as separate species. In particular, the presence of a central strand, short leaves, thin-walled alar cells and growth under continental climatic conditions suggest a rather clear exclusion of lineage B from the broadly circumscribed *P. subeugyrium.* Similarly, lineage C, despite sharing with lineage A the lack of a central strand, canaliculate leaves, cell areolation in the

basal leaf portion, and occupying somewhat similar climatic regions, their ranges have been isolated for a long time and tests of niche identity suggest their divergence. Plants of these lineages differ in habit, especially in their coloration. Although *P. subeugyrium* lineages D and E are closely related according to cp data and recombinant plants between these two occur, the ITS-based topology, clear-cut difference in size, coloration and structure of alar group and allopatric distributions support considering them as separate species.

The name *Pseudohygrohypnum eugyrium* is based on a type from Central Europe (northern Black Forest, Germany) where no other lineage of subaquatic *P. eugyrium* except lineage A occurs. The name *P. subeugyrium* is based on a type from North America (Newfoundland, Canada), which we verified as corresponding to our *P. subeugyrium* lineage A (see Taxonomy Section). The type specimen of *P. purpurascens* illustrated by *Kanda (1976)* perfectly corresponds to several of our specimens assignable to *P. subeugyrium* clade C, and plants from Japan are (nearly) identical with several specimens used for our molecular study. We are therefore reasonably certain of the identification of this lineage with the type of *P. purpurascens*.

The type of another name, *Hygrohypnum poecilophyllum* Dixon, was synonymized with *P. purpurascens* by *Kanda (1976)*. According to the protologue, it differs from the latter in the more slender plants which excludes identity with *P. subeugyrium* lineage D, which consists of rather robust plants in combination with purplish coloration. These plants also do not possess the strongly concave ovate broadly acute to subobtuse leaves that are mentioned in the protologue of *H. poecilophyllum* (*Dixon, 1934*). The other recognized lineages of *P. subeugyrium* occurring in East Asia even more strongly conflict with the protologue of this species. Therefore, this name most likely applies to the continental morphotype of *P. purpurascens* (see the Taxonomy section).

According to the description of *Hygrohypnum subeugyrium* var. *japonicum* Cardot by *Noguchi (1991)*, this variety resembles *P. eugyrium* lineage C, in occasionally lacking a hyalodermis, but the reported shorter leaf cells and smaller spores seem to preclude the identity of *P. eugyrium* lineage C with *P. subeugyrium* var. *japonicum*. The epithet ''*japonicum*'' has never been applied to a taxon at the rank of species within *Pseudohygrohypnum*. However, a combination with this epithet already exists in the genus *Hygrohypnum* (*H. molle* var. *japonicum* Sakurai), where *Pseudohygrohypnum* species were considered for a long time. For these reasons we prefer not to apply this epithet to our *H. eugyrium* lineage C despite the possibility of their identity.

*Hygrohypnum mackayi* (Schimp.) Loeske, described from Ireland, was suggested by *Jamieson (1976)*, who studied the type specimens of this taxon, to be identical with *P. eugyrium*, and we have no reason to doubt this opinion. Presumeably, the type would belong to our lineage A as no other lineage of *P. eugyrium* s.lat. is known to occur on the British Isles. Our specimen HyF16 from Scotland fully agrees *P. eugyrium* var. *mackayi* morphologically. In sum, *P. eugyrium* lineages B and C as well as *P. subeugyrium* lineages B, D and E do not have any available names and consequently are described as new species in the Taxonomy section.

Since molecular phylogenetic studies became a common tool within the framework of integrative taxonomy, a large number of taxonomic concepts that were previously

accepted have been challenged. In particular, *Patiño & Vanderpoorten (2018)* suggested that "apparently widely distributed bryophyte species actually correspond to complexes of multiple species with much narrower distribution ranges" and indeed, an abrupt increase in narrowly circumscribed species with overlapping morphological characters can be observed in the last decade. The specific concepts have been often challenged in taxa with the disjunct distributions, where isolation and different climatic conditions in different parts of the species range could trigger allopatric speciation. In such cases, acceptance of the geographically circumscribed, genetically distinct species is a common practice (for discussion see *Heinrichs et al., 2009*) even if they are hardly distinguishable morphologically. For instance, *Rhynchostegium shawii* Hutsemékers & Vanderp. was described for a distinct lineage within *R. riparioides* (Hedw.) Cardot s.l., which however hardly differs morphologically from the European plants considered as *R. riparioides* s.str. (*Hutsemékers et al., 2012*). Similarly, allopatric and phylogenetically distinct populations of *Bryoxiphium*, despite the lack of morphological differentiation between some of them, are still considered separate species (*Patiño et al., 2016*; *Fedosov et al., 2016a*). On the contrary, disjunct genetically distinctive European and North American populations of *Claopodium whippleanum* (Sull.) Renauld & Cardot and *Dicranoweisia cirrata* (Hedw.) Lindb. ex Milde were considered conspecific by *Shaw, Werner & Ros (2003)*, as no morphological differentiation between them could be discerned.

Our results which quadrupled the number of accepted semiaquatic species of *Pseudohygrohypnum* compared to the earlier monograph of *Jamieson (1976)* and further treatments based on this study, has underlined the importance of cool-temperate and boreal Asia as one of the overlooked centers of bryophyte diversification. For a long time, diversity in this region has been underestimated due to poor bryological exploration and in general, northern Asia has been assumed to house a limited number of endemic mosses. Rapoport's rule (*Rapoport, 1982*; *Stevens, 1989*) enhanced by their rather slow evolution and high migration rate was thought to lead to large ranges for Holarctic mosses (cf. *Patiño & Vanderpoorten, 2018*) until molecular phylogenetic studies started to suggest differently. Recent studies of both mosses and liverworts based on the integrative taxonomic approach (*Sofronova, Mamontov & Potemkin, 2013*; *Vilnet, Borovichev & Bakalin, 2014*; *Bakalin & Vilnet, 2014*; *Bakalin & Vilnet, 2018*; *Ignatova et al., 2016*; *Ignatova et al., 2017*; *Ignatova et al., 2019*; *Fedosov et al., 2017*; *Ignatova, Kuznetsova & Ignatov, 2017*; *Bakalin et al., 2020*; *Hedenäs, Kuznetsova & Ignatov, 2020*; *Ignatov et al., 2020*) have revealed a number of overlooked or previously unknown species that are endemic to the Asian sector of the Holarctic. In many cases these have for a long time been considered conspecific with European or North American species but are in fact remarkably distinct. These studies focused on northern Asia thus corroborate patterns seen in other recent studies on bryophyte species with disjunct distributions (*Medina et al., 2013*; *Draper et al., 2015*; *Ignatova et al., 2015*; *Patiño et al., 2016*; *Vigalondo et al., 2019*, *etc.*).

## Species distribution modelling, the missing piece of integrative taxonomy

Our results indicate remarkable niche divergence between two related lineages of *P. eugyrium* (lineages A and B) and four related lineages of *P. subeugyrium* (lineages A, B, C and E). On the other hand, we confirmed niche identity for a pair of species that are not related to each other but are distributed in the same area (*P. eugyrium* C and *P. subeugyrium* C), although the equivalency tests tend to reject the null hypothesis of excessive niche identity (*Peterson, 2011*; *Broennimann et al., 2012*). Likewise, a niche similarity higher than random was found for two pairs of unrelated suboceanic lineages, (1) American *P. eugyrium* (B) and amphiatlantic *P. subeugyrium* s.str. (A), as well as (2) American *P. eugyrium* (B) and *P. purpurascens* (*P. subeugyrium* C), which are clearly allopatric. At the same time, the niches of (1) two suboceanic lineages of *P. eugyrium* (A, C), (2) two suboceanic lineages of *P. subeugyrium* (A, C), and (3) two continental lineages of *P. subeugyrium* (B, E) are less similar. This is especially interesting since no evidences of niche divergence have been obtained in bryophytes by environmental niche modelling (for discussion see *Collart et al., 2021*).

Although the integrative taxonomy approach implies combining different data sources helpful in species delimitation, most studies which claim to employ this approach are based on only two sources: a molecular phylogenetic study and a morphological examination of the revealed entities. The ecological and distributional data rarely enjoy thorough consideration, and formal methodologies such as SDMs, which have undergone an explosive development recently, are not typically used to explore the niche divergence among the studied lineages. This is unfortunate since niche divergence and/or allopatric distributions might present the final pieces of evidence supporting the taxonomic conclusions (*Van Valen, 1976*; *Mayr, 1982*). Moreover, niche differentiation likely reflects a primary physiological differentiation (in our case probably increasing tolerance to desiccation in continental lineages), which is as significant as a morphological differentiation, although harder to assess.

The few exceptional studies (*Gama, Aguirre-Gutiérrez & Stech, 2017*; *Moroni, O'Leary & Sassone, 2019*; *Nie et al., 2020*; *Collart et al., 2021*, *etc.*) usually consider only limited areas and therefore do not deal with intercontinental disjunctions. With their effective dispersal (*Shaw, 2001*; *Huttunen et al., 2008*; *Patiño & Vanderpoorten, 2018*), bryophytes are suitable model organisms for considering niche similarity or divergence hypotheses on wider areas, covering whole species distribution ranges.

The congruence of the molecular results, morphological evidence and niche divergence of individual lineages, all obtained from independent lines of evidence, justifies the need to split the two traditionally recognized subaquatic *Pseudohygrohypnum* species into eight species as proposed in the Taxonomy Section. We believe that complementing the standard morpho-molecular toolkit of integrative taxonomic studies with species distribution modelling should become recommended common practice.

## Origin and early radiation of *Pseudohygrohypnum*

According to our topology, inferred from plastid markers (Figs. 1, 2, 3), the genus *Pseudohygrohypnum* represents a well-established and remarkably diverse lineage, distributed mostly in the Holarctic, with its centre of species diversity occurring in temperate East Asia. An East Asian centre of distribution is characteristic of earlier diverging species in both major clades identified within *Pseudohygrohypnum* (Fig. 3), *P. fauriei* (which also occurs in Eastern North America), *P. densirameum*, *P. calcicola* and *P. eugyrium* C, and three poorly known species assigned to the genus by *Kučera et al. (2019)*, *P. perspicuum* (Mitt.) Jan Kučera & Ignatov, *P. submolluscum* (Besch.) Jan Kučera & Ignatov and *P. emodifertile* (Broth. ex Ando) Jan Kučera & Ignatov, have a Sino-Himalayan distribution. East Asian representatives also prevail among the closely related genera of Pylaisiaceae, *Calohypnum*, *Pylaisia* and *Homomallium*. In the case of *Pylaisia*, the East Asian representatives also prevail among the early diverging species according to the topology of the ITS-IGS-*trnL* tree published by *Ignatova et al. (2020)*. This strengthens the hypothesis that the radiation of core Pylaisiaceae including the basal radiation of the genus *Pseudohygrohypnum* occurred in East Asia while the Atlantic region was colonized later and independently by several lineages of the family. According to our estimate (Fig. 3), the genus *Pseudohygrohypnum* originated in the Oligocene; and the common ancestor of current *Pseudohygrohypnum* species likely was an element of the Arcto-Tertiary flora. The East Asian–East North American disjunction, shown by the earliest diverging extant *Pseudohygrohypnum* species, *P. fauriei*, is believed to be typical of Arcto-Tertiary species (*Schofield & Crum, 1973*; *Schofield, 1988*).

The two main lineages of the genus (Fig. 3) are somewhat distinct morphologically, since within clade 1 most of the species have a well-developed hyalodermis (weakly developed in *P. fertile* and *P. skottsbergii*), while species of clade 2 have no hyalodermis except for *P. eugyrium* C where it is irregularly differentiated. The phylogenetic reconstructions suggest that the mesophilous species of *Pseudohygrohypnum* originated in the late Oligocene and warmer early Miocene before the cooling of the climate; the same is true for *P. eugyrium* C, which remains associated with temperate climates in Asia where it mostly occurs south of the 43rd parallel (*Ivanov et al., 2017*).

## Diversification of the "atlantic" *Pseudohygrohypnum* clade

Diversification of the *P. eugyrium* A plus B and *P. subeugyrium* A–E lineages occurred much later, after Miocene cooling (14 Ma), and apparently nearly simultaneously in different areas. Since the earliest diverging lineage of *Pseudohygrohypnum* clade 1 (*P. fauriei*, cf. Fig. 3) is distributed in both East Asia and eastern North America and the remaining species of this clade (except *P. densirameum*, which likely originated from an Asian population of *P. fauriei*) occur either in the amphiatlantic region (*P. eugyrium* A & B, *P. fertile*), or in South America (*P. skottsbergii*), we assume that diversification of the *P. eugyrium* A plus B clade was associated with the Atlantic region, and presents an example of allopatric speciation. Although our sampling is sparse, we discovered a notable divergence within North American populations of *P. eugyrium* B, as compared to the low nucleotide diversity in European *P. eugyrium* A, despite sampled plants originating from quite distant areas

(Alps, Scandinavia, UK). *Schofield (1988)* proposed pre-Pliocene isolation of relict groups of southern amphiatlantic bryophytes, to which he assigned *P. eugyrium* and indeed, it seems plausible that the amphiatlantic *P. eugyrium* (A+B) lineage originated in North America and colonized Europe as a result of a single dispersal event or several such events within a short period of time. Our dating provides a median age of 10.56 Ma for the *P. eugyrium* A/B node. This was a period of an abrupt late Miocene cooling and drastic changes in atmospheric circulation, which resulted in the formation of modern biomes (*Pound et al., 2012*). Nearly at the same time (8–10 Ma), migration of several tree species across the north Atlantic land bridge occurred (*Grímsson & Denk, 2005*; *Denk, Grímsson & Zetter, 2010*). For other groups of plants with amphiatlantic disjunctions, more recent divergence times between European and North American taxa have been suggested (*Milne, 2004*). The phylogeographic study of the moss genus *Homalothecium* by *Huttunen et al. (2008)* provides an estimate of the divergence between the American and Eurasiatic clades at 5.69 Ma and this estimate might rather refer to the origin of the European–western North American disjunction of the genus *Homalothecium*, while the amphiatlantic disjunction of *H. sericeum* (Hedw.) Schimp. has originated later, likely as a result of recent colonization of eastern North America from Europe (see reconstruction by *Huttunen et al., 2008*). Likewise, the molecular phylogenetic study of the genus *Antitrichia* Brid., showed no differentiation between European and eastern North American populations of *A. curtipendula* (Hedw.) Brid. This also is the case of *Pseudohygrohypnum subeugyrium* s.str., for which no substantial molecular differentiation of European and eastern North American populations was found, but not the case of *P. eugyrium* A & B, which are remarkably distinct phylogenetically. The diversification of these lineages might thus represent the first evidence of the pre-quaternary isolation of eastern North American and European populations of an amphiatlantic lineage in mosses.

Heat transfer associated with the Gulf Stream which supports the mild climatic conditions of the northern amphiatlantic region likely contributes to the present distribution of *P. eugyrium* A in Europe and *P. eugyrium* B in North America. Moreover, the stability of these climatic conditions since the Pliocene (see *Denk, Grímsson & Zetter, 2010*; *Denk et al., 2013*; *Utescher, Bondarenko & Mosbrugger, 2015*) may be responsible for the lack of further diversification within the European *P. eugyrium* clade, as contrasted with *P. subeugyrium* s.lat.

### Diversification of *P. subeugyrium* s.l.

Along with the two early diverging East Asian species, *P. eugyrium* C and *P. calcicola*, the second major clade of *Pseudohygrohypnum* (Fig. 3) includes the remarkably diverse *P. subeugyrium* complex. In contrast to the Atlantic clade of the genus (cf. Fig. 3), diversification of this clade apparently was associated with temperate East Asia, which has experienced cooler climatic conditions than temperate Europe since the mid-Oligocene (*Utescher, Bondarenko & Mosbrugger, 2015*). The *P. subeugyrium* complex underwent a remarkable shift in ecology and spread across the continental areas of North Asia. According to obtained estimates, the two clades, containing the "continental" lineages D & E and B split 10.63 and 5.97 Ma, in the period that likely was associated with two stages of late Miocene

cooling, at 14 and 7 Ma. Along with the decreasing annual temperatures, this period was characterised by a dramatic aridification of continental Asia, increased seasonality in temperature and precipitation, and a stronger latitudinal temperature gradient. Xeric environments appeared in the continental part of North Asia due to intensification of the Siberian zone of high pressure and the establishment of the East Asian summer monsoon pattern (*Liu, Eronen & Fortelius, 2009*; *Pound et al., 2012*; *Utescher, Bondarenko & Mosbrugger, 2015*; *Holbourn et al., 2018*). This resulted in the formation of modern biomes (*Herbert et al., 2016*) and no significant glacial events followed in continental Asia during the Pleistocene (*Sheinkman et al., 2017*). The ancestor of the *P. subeugyrium* complex, which apparently faced these changes in East Asia, radiated and formed several lineages adapted to different climatic conditions (monsoon/oceanic *versus* continental). In the latter case, the adaptation may have included increasing tolerance to desiccation, since small watercourses, where such mosses typically grow in continental North Asia, usually dry out in mid-summer. Thus, the monsoon-related west to east moisture gradient may have been a driver of speciation in both *P. subeugyrium* subclades. Further climate oscillations during the Pliocene might have resulted in expansion northwards in warmer and moister periods and the subsequent isolation of northern populations in cooler and drier periods, causing bottlenecks and subsequent gene drift. Such effects might be inferred from the remarkable divergence of the continental *P. subeugyrium* lineages B, D and E and their low within group genetic diversity as compared to the "oceanic" lineage C. According to results of our dating, the youngest continental Pseudohygrohypnum species are contemporaries of mammuth; however, unlike the latter they have survived during Holocene climate change.

One of the two major lineages of *P. subeugyrium* s.l. spread northwards and differentiated into the *P. subeugyrium* lineage D which is locally common in a restricted area with a cool-temperate climate, and the subarctic *P. subeugyrium* lineage E. Lineage D also colonized North America, where it occurs sympatrically with *P. eugyrium* B and *P. subeugyrium* A, while lineage E has adapted to harsh subarctic environments that was associated by undergoing remarkable miniaturization. Interestingly, traces of gene flow between lineages D and E persist as recombinant plants occurring sympatrically with plants representing lineage D throughout its range in Asia. All small plants from southern Siberia, Yakutia and Khabarovsk Territory tested using molecular markers appeared recombinant and thus the identity of a specimen from Altai, referred to lineage E on morphological grounds, remains unclear. Apparently, the origin of these recombinant haplotypes is associated with milder climatic periods when the distributions of lineages D and E may have overlapped.

*Pseudohygrohypnum subeugyrium* lineage B, which occupies continental areas with a dry climate and strong temperature seasonality, has spread throughout southern Siberia and reached the Ural Mountains. Although the range of this species appears spotty, this might be the result of poor bryological exploration of southern Siberia.

The clade sister to *P. subeugyrium* lineage B (Fig. 3) is formed by two morphologically similar allopatrically distributed species which have remarkably diverged from each other. Unlike the quite homogeneous Atlantic *P. eugyrium,* the East Asian *P. purpurascens* (*P. subeugyrium* C) is represented by several separate lineages which might even be worthy of recognition as separate taxa. This diversity may have originated from the strong

climate gradients between insular East Asia, the continental Russian Far East and the Sino-Himalayan region, where different lineages of *P. subeugyrium* C occur, in contrast the range of *P. subeugyrium* A is rather uniform climatically. On the other hand, *P. subeugyrium* A may have originated from a long-distance dispersal event or its haplotype structure may be the result of bottleneck effects associated with survival during glaciation in the Atlantic sector. Traces of the wider distribution of *P. purpurascens* persist in recombinant plants from the Khamar-Daban Range (Baikal Region, south Siberia), suggesting past expansions of its range during milder climatic periods. Several disjunct species/populations of vascular plants considered as tertiary relicts with their closest relatives in East Asia have survived in this mountain range due to locally mild climatic conditions resulting from the influence of Lake Baikal (*Chepinoga, Protopopova & Pavlichenko, 2017*). It is noteworthy that our SDM obtained for *P. purpurascens* predicts its presence exactly in this isolated area.

There are numerous examples of species with East Asian–eastern North American disjunct ranges, including those in the genus *Pseudohygrohypnum*. In this case, we suggest that the ancestor of *P. subeugyrium* lineage A first inhabited North America and after diverging from lineage C it then colonized Europe, repeating the path travelled by the "Atlantic" clade. Higher haplotype diversity of *P. subeugyrium* lineage A in North America, according to both plastid and ITS data, with plesiomorphic character states revealed in the rps4 sequence obtained from specimen P2274, supports this suggestion.

**Has late Cenozoic climate cooling increased bryophyte diversity?**

Milankovitch climatic oscillations, which are considered to be responsible for low diversity and larger species ranges in higher latitudes (*Dynesius & Jansson, 2000*; *Dynesius & Jansson, 2002*), have apparently not affected the overall diversity of bryophytes since the latitudinal diversity gradient is weakly expressed in this group (*Hillebrand & Azovsky, 2001*; *Dynesius & Jansson, 2002*; *Shaw, Szövényi & Shaw, 2011*; *Mateo et al., 2016*). Because of this, bryophyte diversification has been proposed to have been associated with cooler climatic conditions (*Romdal, Araújo & Rahbek, 2012*). Massive extinctions suggested to have taken place in bryophytes (*Laenen et al., 2014*) likely have been balanced by the diversification of new lineages, such as those described above for *Pseudohygrohypnum*, or previously for other genera such as *Hedwigia* (*Ignatova et al., 2016*), *Orthotrichum* (*Fedosov, Fedorova & Ignatova, 2017*), *Orthothecium* (*Ignatov et al., 2020*), *Schistidium* (*Ignatova et al., 2010*), *Tomentypnum* (*Hedenäs, Kuznetsova & Ignatov, 2020*), and many other groups of bryophytes common at middle and high latitudes in Asia. Indeed, the diversification rate within the *P. subeugyrium* clade (0.38–0.7 new lineages per million years) is comparable with the one found by *Huttunen et al. (2008)* for *Homalothecium* and considered by those authors as fairly high. At the same time, within the amphiatlantic *P. eugyrium* clade, which arose around the same time, no significant radiation has followed.

According to *Jonsgard & Birks (1996)*, *Ellis & Tallis (2000)* and *Patiño & Vanderpoorten (2018)*, ecotopes appearing due to the change of climatic phase likely have been colonized by bryophytes as soon as they developed and became available. However, we suggest that such changes affecting simultaneously huge areas of North Asia, have resulted in a high local proportion of open ecological niches (so called "dark diversity", see *Pärtel, Szava-Kovats &*
*Zobel, 2011*) even for easily dispersable bryophytes. For instance, *Carter (2021)* concluded that substantial areas of climate suitable for North American endemic mosses exist (and likely existed before) outside of North America. Weakly saturated local species pools with the constant presence of open niches have favoured niche divergence in populations which have survived these changes *in situ* and thus may have been a driver of speciation followed by divergence along climatic gradients and molecular divergence. Such mechanisms might have contributed significantly to the diversification burst proposed by *Laenen et al. (2014)* and led to the formation of the inverse latitudinal gradient of species diversity, revealed in Europe by *Mateo et al. (2016)*.

## CONCLUSIONS

Our integrative taxonomic study, which combined molecular evidence, morphological revision of specimens and species distribution modelling revealed eight lineages within two traditionally circumscribed subaquatic species of the genus *Pseudohygrohypnum*. These lineages were found in three clades, the subatlantic *P. eugyrium* clade (with two lineages), *P. subeugyrium* clade (with five lineages), and the clade containing East Asian specimens, previously assigned to *P. eugyrium*. The former two clades originated rather recently, following the 14 Ma Miocene cooling, while the mesophilous species of the genus originated earlier, during a warmer climatic period. Within the main *P. eugyrium* clade, two species diverged allopatrically in mild suboceanic areas of Europe and Eastern North America. The distributions of several lineages within the *P. subeugyrium* clade partly overlap. At the same time, species distribution modelling and consequent niche similarity tests showed that closely related species occupy distinct niches. Within the *P. subeugyrium* clade where diversification was largely associated with Asia, three lineages underwent significant shifts in ecology that allowed them to colonize continental areas of northern Asia. Moreover, we propose that Milankovitch climatic oscillations may have fasten divergence and speciation, induced by the cooling and aridification of climates in the late Miocene and Pliocene.

## TAXONOMIC TREATMENT

Key to identification of subaquatic species of *Pseudohygrohypnum*

1. Stems with distinct central strand; leaves ovate, rarely ovate-lanceolate, with length/width ratio 1.5–2.5, straight to distinctly falcate, acuminate; costae extending to $\frac{1}{4}-\frac{1}{2}$ of leaf length; reddish coloration rare 2

- Stems without or with weak central strand; leaves ovate-lanceolate to lanceolate, with length/width ratio (2–)2.5–4.5, acuminate or attenuate at apex, distinctly to strongly falcate, rarely leaves shorter, ovate, but then plants have distinct red coloration; costae typically shorter than $\frac{1}{4}$ of leaf length 5

2.  Stem hyalodermis not differentiated or differentiated in several patches; alar cells not sharply delimited; Asia and Ural Mountains 3

-   Stem hyalodermis differentiated throughout the cross section; alar cells sharply delimited; Western Europe, Caucasus and Eastern North America 4

3.  Leaves strongly concave, with involute margins, apiculate distally; median leaf cells elongate to short linear, 30–50(–65) μm long *P. orientale* (*P. eugyrium* C)

-   Leaves weakly concave, with plane margins. acuminate; median leaf cells linear or vermiculate, 35–70(–85) μm long *P. sibiricum* (*P. subeugyrium* B)

4.  Leaves 1.2–1.6(−1.8) mm long; alar group composed of thick-, rarely thin- walled, red- or brownish coloured cells; upper leaf cells 30–62 μm long; endostome with three, free, typically short cilia, ca. 1/2–2/3 of segment length; Europe *P. eugyrium* (*P. eugyrium* A)

-   Leaves 1.6–1.8 mm long; alar group with thin-walled, hyaline cells or firm-walled hyaline cells forming the outer parts and reddish-colored, somewhat thicker-walled cells forming the inner parts; upper leaf cells 45–75 μm long; endostome with cilia adherent to each other, typically longer than or equal to segments; North America *P. appalachianum* (*P. eugyrium* B)

5.  Plants small; stems usually densely pinnately branched; leaves 0.6–1(−1.2) mm long, lanceolate to linear-lanceolate; alar cells not inflated; North Siberia 6

-   Plants medium-sized; stems irregularly branched; leaves usually longer than 1.2 mm, ovate-lanceolate to lanceolate; alar cells distinctly differentiated; cool temperate, humid areas of Holarctic 7

6.  Plants with remarkable golden sheen; alar groups indistinct, composed of few small, quadrate, thick-walled cells *P. subarcticum* (*P. subeugyrium* E)

-   Plants purplish or brownish; alar group distinct, composed of subquadrate, thick-walled cells with reddish to brownish cell walls *P. neglectum* × *P. subarcticum*

7.  Leaves slightly concave; alar groups round, inflated, composed of thin-walled cells; leaf base from alar region to costae occupied by not excavated, concolorous cells with thick, strongly porose walls *P. neglectum* (*P. subeugyrium* D)

-   Leaves concave, in upper part canaliculate; alar groups transversely elongate, composed of thin- or moderately thick-walled cells; leaf base from alar region to costae occupied by one row of excavated, brownish cells with thick, not or slightly porose walls 8

8.  Plants with reddish coloration, typically deeply wine red, rarely variegate to green; spores 15–30 μm; East Asia *P. purpurascens* (*P. subeugyrium* C)

-   Plants bronze to brownish, typically without or with slight reddish coloration; spores 12–20 μm; north Europe and Atlantic North America *P. subeugyrium* (*P. subeugyrium* A)

*Pseudohygrohypnum appalachianum* Brinda, Fedosov & Ignatova, sp. nov. (Figs. 4–7, 11, 12)

**Type:** USA, North Carolina, McDowell County, Pisgah National Forest, along Newberry Creek and Forest Road 482A, 35.6977N, 82.2397W, 872 m. alt., eastern deciduous forest with *Rhododendron*, on rock in stream; 28.V.2018, Brinda 11916 (Holotype MO-6967804!, Isotype MW!).

**Diagnosis.** Similar to *P. eugyrium* in its stem with central strand and weak hyalodermis, and leaves with well-marked, usually excavate alar regions of inflated, rectangular cells, but differing from that species in: (1) somewhat longer laminal cells, 45–75 µm *vs.* 30–62 µm; (2) alar groups composed of thin-walled, hyaline cells excepting the innermost rows with slightly thicker walls *vs.* uniformly thick-walled, reddish to brownish cells in *P. eugyrium*; (3) more closely set, often adherent endostome cilia *vs.* free ones in *P. eugyrium*; and (4) spores, which are on average larger, (14–)15–20(–23) µm *vs.* 13–20 µm in *P. eugyrium*.

**Etymology.** The species is named after the Appalachian Mountain Range, which spans the length of eastern North America where the species is especially frequent.

**Description.** Plants medium-sized, yellowish-green to reddish-brown, generally with a satin-like sheen. Stems prostrate or ascending, irregularly branched, to 4 cm long, yellow, becoming reddish-brown with age; epidermal cells somewhat larger, with thin outer walls, forming a weak hyalodermis; sclerodermis composed of 3–5 layers of smaller, thick-walled cells; central strand distinct. Stem leaves 1.3–1.8 × 0.6–0.9 mm, straight to, more commonly, falcate-secund, ovate, oblong-ovate or ovate-lanceolate, concave, narrowed to the insertion, widest at ca. 1/3 of leaf length, apex acuminate or more commonly with apiculus formed due to incurved upper leaf margins; costa usually short and double but sometimes reaching mid-leaf, or rarely single and/or spurred; margins erect and entire below, becoming increasingly denticulate and broadly incurved above, especially in the apiculus where they are consistently serrulate or eroded; upper laminal cells firm-walled, fusiform to linear, flexuose, (32–)45–75(–80) × 4–5(–6) µm, somewhat shorter toward the apices; basal juxtacostal cells concolorous or reddish, thicker-walled, frequently porose; alar group clearly delimited, enlarged to inflated, usually excavate, composed of several thin-walled, hyaline cells at margins which are surrounded by more firm-walled, usually reddish-colored cells in the interior but not reaching the costa; short-rectangular supra-alar cells usually present in one to several rows. Autoicous. Perichaetial leaves lanceolate, acute or acuminate, plicate, up to 3.5 mm long. Setae yellowish-brown to reddish-brown, 1.5–2.5 cm, smooth. Capsules horizontal, 1–2 mm long, oblong-cylindrical, strongly curved, contracted below mouth, yellowish-brown, mostly smooth. Exothecial cells at the capsule mouth small, oblate or rounded in 1–3 rows, the rest elongate, firm-walled. Opercula conic-apiculate. Annuli well differentiated, composed of 2–4 rows of inflated cells. Exostome teeth up to 500 µm, yellowish-brown to orange-brown, cross-striolate proximally, papillose distally; endostome nearly of the same length, weakly papillose, basal membrane ca. 200 µm; segments keeled, narrowly perforate along the median line; cilia 2–3, closely set or occasionally fused for part of their length, filiform, nodose. Spores (14–)15–20(–23) µm, smooth to finely papillose.

**Variation.** Coloration of the plants naturally varies from yellowish-green to reddish-brown with age and exposure. The alar cells also become better differentiated with age, so observations are best made along mature stems. The longer laminal cells of North

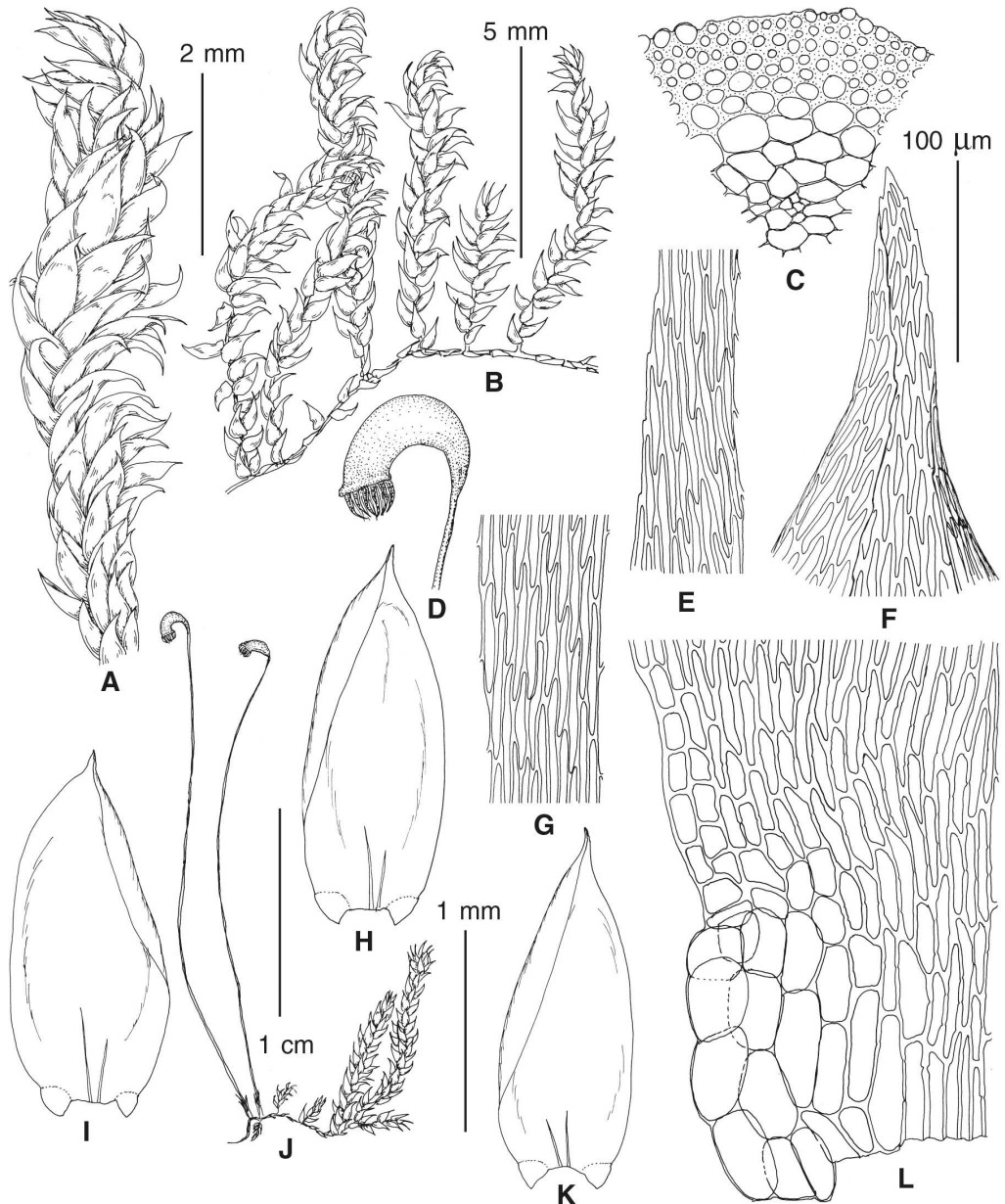

**Figure 11** ***Pseudohygrohypnum appalachianum* (from holotype).** (A, B, J) Habit. (C) Stem transverse section. (D) Capsule. (E, G) Mid-leaf cells. (F) Upper leaf cells. (H, I, K) Leaves. (L) Basal leaf cells. Scale bars: 5 mm for B; 2 mm for A, D; 1 mm for H, I, K; 100 μ m for C, E-G, L.

American plants as compared to *P. eugyrium* s.str. were already pointed by *Jamieson (1976)* in his revision of *Hygrohypnum*. Therefore, the given ranges for laminal cell size follow that treatment.

**Differentiation.** The unique structure of the alar groups of *P. appalachianum* is fairly diagnostic: they are composed of thin-walled, hyaline cells ringed by firm-walled, reddish-colored cells that abruptly transition into the thick-walled, and often porose

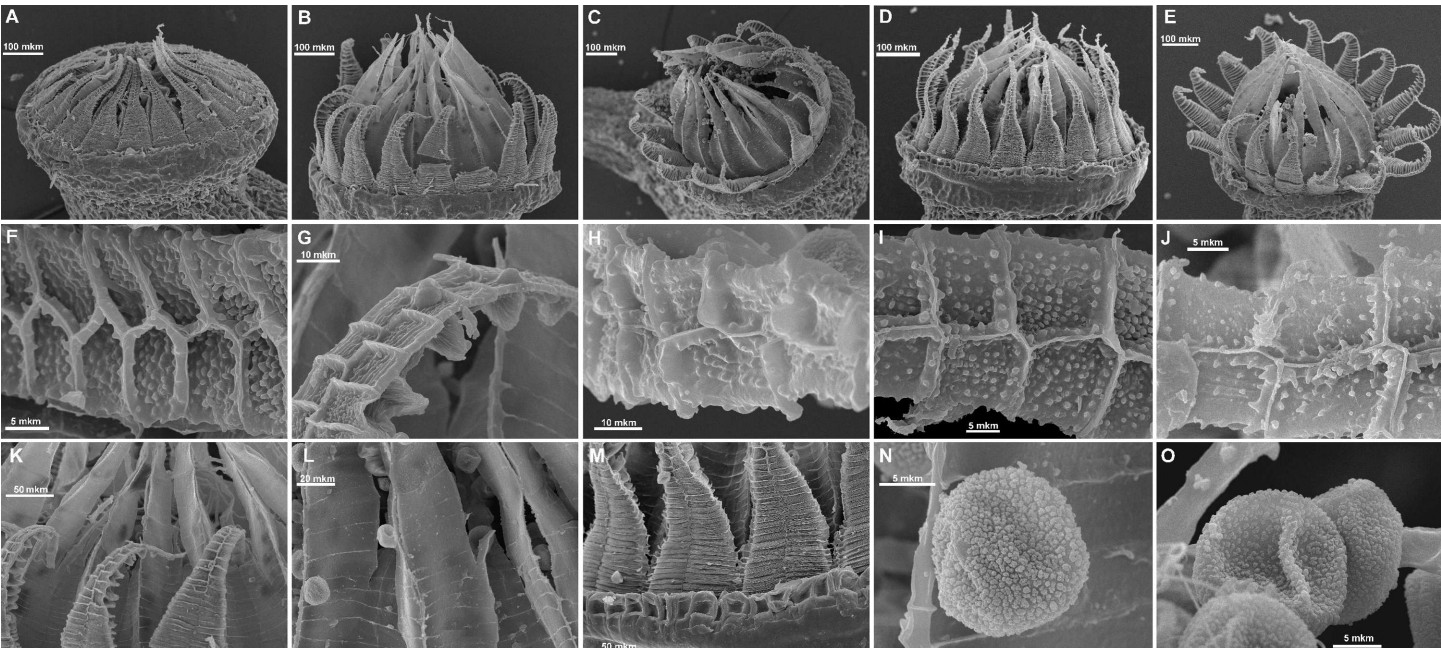

**Figure 12** **SEM images of peristomes and spores of selected subaquatic species of the genus *Pseudohygrohypnum* (all images from Holotypes except for I).** Peristome (A) *Pseudohygrohypnum appalachianum*. (B) *P. orientale*. (C) *P. sibiricum*. (D) *P. subarcticum*. (E) *P. neglectum*. Distal portion of tooth from the outside (F) *Pseudohygrohypnum appalachianum*. (G) *P. orientale*. (H) *P. sibiricum*. (I) *P. purpurascens* (from Ignatov & Ignatova 13-1388, HyF52). (J) *P. neglectum*. Fragments of peristome with outer endostome surface (K) *P. orientale*. (L) *P. sibiricum*. (M) *P. subarcticum*. Spore (N) *P. sibiricum*. (O) *P. neglectum*.

juxtacostal basal cells. In both *P. eugyrium* s.str. and *P. subeugyrium,* the firm-walled cells generally extend all the way to the leaf margin. In addition, the cilia of the endostome of *P. appalachianum* are longer than in *P. eugyrium,* the same length as the segments *vs.* half of the segment length, more closely set and partially joined (free in *P. eugyrium*); finally, spores of *P. appalachianum* are somewhat larger (rarely less than 15 μm in *P. appalachianum* but frequently less than 15 μm in *P. eugyrium*). Differentiation from *P. neglectum* is perhaps more difficult, since both species have at least partly thin-walled, hyaline alar cells. However, the distinct stem central strand, hyalodermis, and strongly concave leaves of *P. appalachianum* with coloured interior basal cells should serve to separate it from that species.

**Ecology.** The species grows on acidic rocks both in and along watercourses, usually in the mountains. In the southern Appalachian Mountains, the species reaches more southerly latitudes than are generally experienced in this group. Therefore, it seems reasonable to hypothesize that these habitats could have served as refugia for this species through past climatic oscillations as was considered for other species (*Billings & Anderson, 1966*; *Anderson & Zander 1973*).

**Distribution.** *Pseudohygrohypnum appalachianum* is apparently a common member of the genus from the southern Appalachian Mountains of North Carolina, Georgia, and Tennessee (*Crum & Anderson, 1981*) north to Maine (*Allen, 2014*) and eastern Canada

(*Jamieson, 1976*; *Jamieson, 2014*). In the north-eastern portion of its range, it may be confused with *P. subeugyrium, P. neglectum*, or *P. eugyrium* s.str., and a revision of herbarium material is needed to outline the true limits of its distribution.

**Other specimens examined (paratypes):** Canada: Nova Scotia, Kings County, creek in mixed forest, 16.VII.1974, R.R. Ireland 17459 (LE); Quebec, Terrebonne County, On wet rocks at edge of brook, 17.VII.1959, H. Crum & H. Williams 10333 (LE); USA: Maine, Oxford County, Batchelders Grant, White Mountain National Forest, along Route 113, Stony Brook, on boulder in stream, 17.VI.2007, B. Allen 28268 (MO); Knox County, Camden Township, Mount Megunticook, Maiden Cliff trail, Camden State Park, on boulders in stream, 25.VII.1997, B. Allen 20038 (MO); the same place, 2.VII.1993, B. Allen 14642 (DUKE); North Carolina, Kanati Fork, Great Smoky Mts. Natl. Park, Swain County, wet boulders of stream, 11.VII.1959, W.B. Schofield 10332 (LE); the same area, over boulders in streamlet, 13.VII.1959, W.B. Schofield 10075 (LE, MO); the same area, wet rocks in stream, hemlock cove, 22.X.1969, L.E. Anderson 20994 (LE); Clay County, rocks at the edge of stream, 13.VIII.1948, L.E. Anderson 8004 (LE); McDowell County, locality same as the type (both collected during the 2018 Crum Workshop), on streamside rock; 28.V.2018, Brinda 11911 (MO, MW); Pennsylvania, Centre County, ca. 2.5 mi. ESE of Millheim, 5.VI.1975, R.A. Pursell 10322 (MO); Tennessee, Rainbow Falls, Great Smoky Mts. Natl. Park, wet cliff slope near waterfall, 28.VII.1959, W.B. Schofield 10656 (LE).

**Affinity.** *P. appalachianum* is closely related to European *P. eugyrium*, but since their divergence, has gained both molecular synapomorphies and morphological differences, which substantiate recognition of *P. appalachianum* at the species level.

***Pseudohygrohypnum eugyrium*** (Schimp.) Kanda, J. Sci. Hiroshima Univ., Ser. B, Div. 2, Bot. 16: 106. 1976[1977]. –*Limnobium eugyrium* Schimp., Bryol. Eur. 6: 73. pl. 579. 1855. –*Hypnum eugyrium* (Schimp.) Sull., Manual (ed. 2) 671. 1856. –*Amblystegium eugyrium* (Schimp.) Lindb., Musci Scand. 33. 1879. –*Calliergon eugyrium* (Schimp.) Kindb., Eur. N. Amer. Bryin. 1: 83. 1897. –*Hygrohypnum eugyrium* (Schimp.) Loeske, Verh. Bot. Vereins Prov. Brandenburg 46: 198. 1905. = *Hypnum eugyrium* var. *mackayi* Schimp., Syn. Musc. Eur. (ed. 2) 782. 1876. (Figs. 4–7, 13).

**Description.** For species description see *Smith (2004)*.

**Differentiation.** For differentiation of *P. eugyrium* s.str. from North American *P. appalachianum* and from East Asian *P. orientale* see comments under those species.

**Distribution.** *P. eugyrium* has a subatlantic distribution, largely restricted to Europe. But since we have assigned nearly all studied North American specimens previously referred to *P. eugyrium* to the newly described *P. appalachianum* , its distribution outside Europe is poorly understood. A single specimen from Newfoundland, which however lacks sporophytes and for which we failed to obtain molecular data, in the structure of the alar group rather resembles European *P. eugyrium*. A revision of all American specimens previously referred to *P. eugyrium* is needed to clarify if this species occurs in North America. *Pseudohygrohypnum eugyrium* is reported from Georgia, Adzharia based on a single specimen (*Zündorf, 2011*), but we have not been able to examine those plants. According to the model produced based on European localities of *P. eugyrium*,

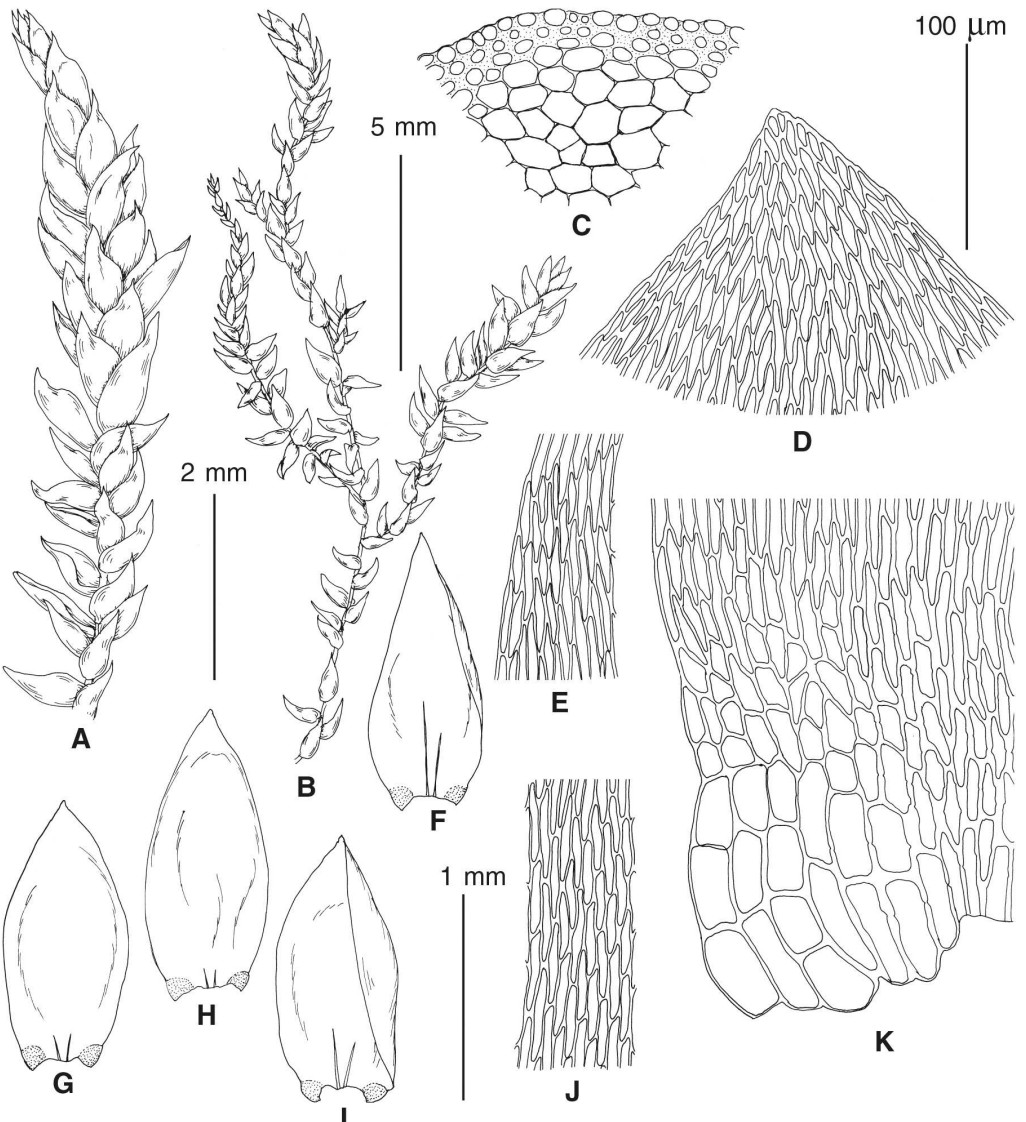

100 μm

5 mm

2 mm

1 mm

**Figure 13** ***Pseudohygrohypnum eugyrium* (from Austria, 16.10.2005. G. Schlüsslmayr, HyF15).** (A, B) Habit. (C) Stem transverse section. (D) Upper leaf cells. (E, G) Mid-leaf cells. (F–I) Leaves. (K) Basal leaf cells. Scale bars: 5 mm for B; 2 mm for A; 1 mm for F–I; 100 μ m for C–E, G–K.

this species is expected to occur both in humid coastal areas of the western Caucasus and in eastern North America, although its predicted distribution there is shifted northward, compared to the model based on north American occurrences, which largely represent *P. appalachianum*.

**Specimens examined:** Sweden, 28.VII.1902. T. Gustafson (LE); 19.VII.1896, A. Arven (LE); 24.VII.1879, O.L. Sillen (LE); 12.VI.1883, O.L. Sillen (LE); 20.VI.1916, P.A. Larsson (LE); Belgium, V.1921, Fr. Toussaint. Austria, 16.10.2005. G. Schlüsslmayr (CBFS, MW); Poland, 24.IV.1956, S. Lisowski (LE); Faroes, 8.V.1896, C. Jensen (LE); 23.V.1896, C. Jensen

(LE); UK, 7.VII.1919 (LE); Scotland, 6.7.2003, J. Kučera (CBFS). Canada, Newfoundland, Avalon Peninsula, 12.V.1981, G.R. Brassard 13562 (LE).

***Pseudohygrohypnum orientale*** Fedosov & Ignatova, sp. nov. (Figs. 4–7, 12, 14).

**Type**: Russia. Primorsky Territory, Partizansk Distr., Ol'khovaya Mt., Kamenystyj Creek valley, 43.33N, 133.65E, ca. 530 m alt., on boulder near the creek, 4.IX.2006, M.S. Ignatov 06-2399, MW9060444* (Holotype), MHA9046084, LE (Isotypes).

**Diagnosis**. This species is similar to *P. eugyrium* in presence of stem central strand, ovate, strongly concave, apiculate leaves with distally serrulate margins, and rather short leaf cells, but differs from it in (1) lack of dark coloration in older parts of shoots, while in *P. eugyrium* older parts of plants are typically brownish to blackish or reddish; (2) weakly developed hyalodermis, which is present only as fragments or may be absent; (3) weakly delimited alar groups; (4) larger spores, 19–23 μm *vs.* 13–20 μm in *P. eugyrium*, and (5) distribution in Eastern Asia, whereas *P. eugyrium* occurs mainly in Europe.

**Etymology.** The species name reflects its East Asian distribution.

**Description.** Plants medium-sized, light yellowish to bright green, pale brownish proximally, moderately turgid, not or weakly glossy. Stems prostrate, yellow, brownish or reddish, to five cm, irregularly branched; hyalodermis well differentiated in several areas of the cross section and not or weakly differentiated in other places, rarely weakly differentiated throughout cross section; sclerodermis 3–4 layered; central strand well differentiated. Stem leaves (0.9–)1–1.6(–2) × (0.3–)0.55–0.8(–0.85) mm, falcate-secund, ovate to ovate-lanceolate, acuminate, typically with slightly attenuate tips, strongly concave; margins broadly and gently incurved below apices; longer leaves canaliculate to tubulose distally to nearly cucullate, slightly and gently or abruptly narrowed toward insertion, widest at ca. (1/5–)1/3–1/2 of leaf length; costa double, rather strong, typically reaching 1/4–1/3(–1/2) of leaf length; margins serrulate distally; upper laminal cells 30–50(–65) ×3–5(–6) μm, linear, thin- to moderately thick-walled, not porose, near leaf tip shorter, elongate to rhomboid, with thicker walls; basal cells usually thick-walled, concolorous, often with porose walls; alar groups distinctly delimited, inflated, ovate to trapezoidal, rarely transversely elongate, composed of wide, inflated, hyaline or thick-walled, short rectangular cells, in the later case usually with brownish cell walls, not reaching costa, bordered by small subquadrate supra-alar cells. Cells between alar groups linear, very thick-walled, porose. Autoicous. Perichaetial leaves lanceolate, acuminate, plicate, up to 3.6 mm long. Setae red, 1.5–2.3 cm. Capsules 1.5–2 mm long, horizontal, curved, not contracted below mouth (but abruptly widened to operculum when young), pale-brown, smooth. Exothecial cells moderately thick-walled, elongate, below mouth in few rows rounded. Opercula conic. Annuli differentiated, composed of large, inflated cells. Exostome teeth up to 400 μm, reddish-brown, cross-striolate proximally, papillose distally; endostome longer, up to 550 μm, basal membrane ca. 250 μm, papillose on the inner surface; segments carinate, perforated along a median line, papillose on the inner surface; cilia 2, filiform, nodulose. Spores 19–23 μm. This species was also illustrated and described by *Czernyadjeva (2003)* based on the specimen from Primorsky Territory as *P. eugyrium*.

**Variation.** *Pseudohygrohypnum orientale* is rather variable morphologically and the spectrum of its morphotypes largely corresponds to that described by Jamieson for *P.*

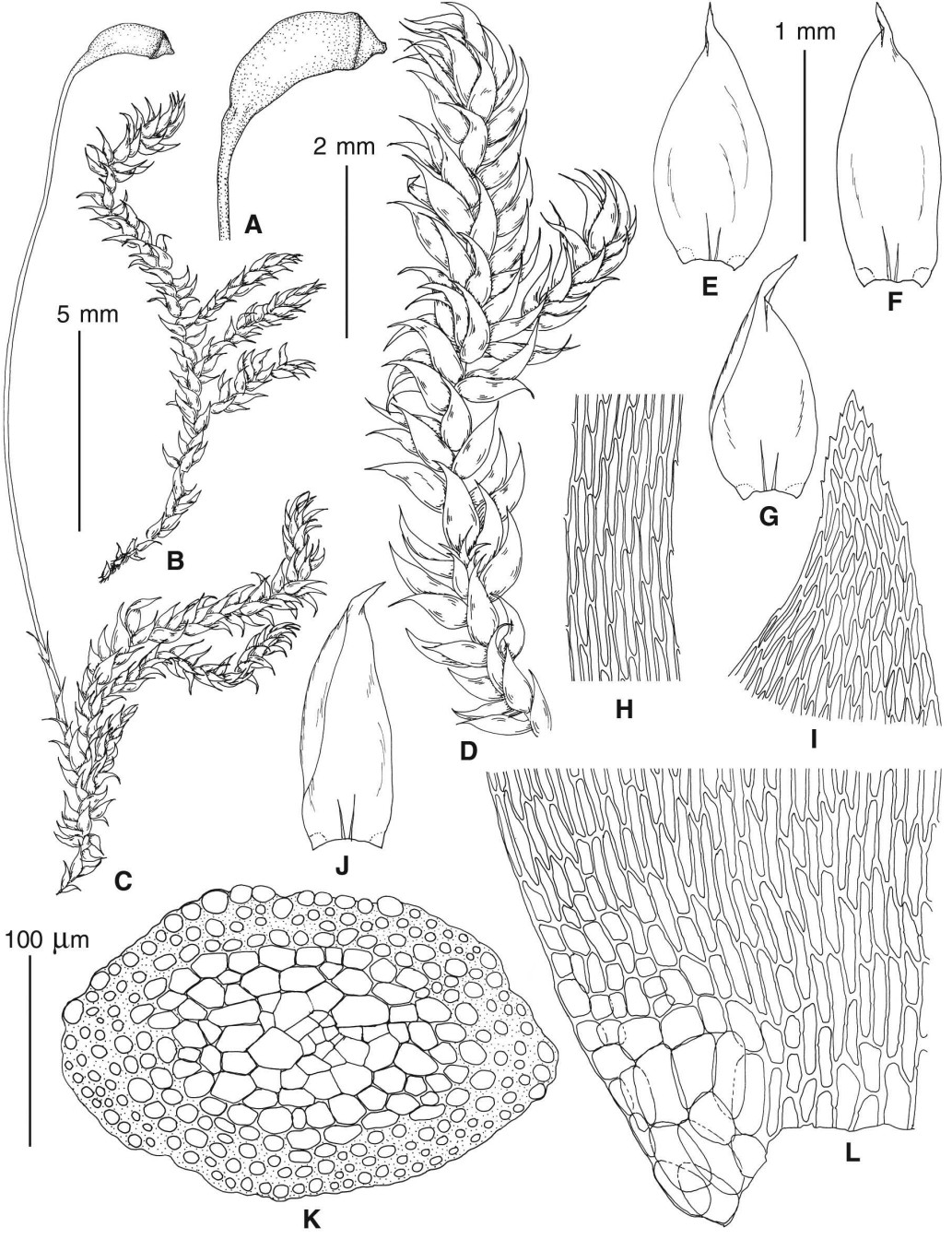

**Figure 14 *Pseudohygrohypnum orientale* (from holotype).** (A) Capsule. (B–D) Habit. (E–G, J) Leaves. (H) Mid-leaf cells. (I) Upper leaf cells. (K) Stem transverse section. (L) Basal leaf cells. Scale bars: 5 mm for B, C; 2 mm for A, D; 1 mm for E–G, J; 100 μ m for H, I, K, L.

*eugyrium.* Leaf length and shape are among the most variable traits; typically leaves in this species are ovate to short ovate (all Russian specimens and nearly half of the Japanese ones), but in several samples the leaves are longer and then resembling *P. purpurascens* in their canaliculate distal portion. The alar groups are also extremely variable in shape and areolation; typically, they are round or round-trapezoidal (*i.e.,* resembling those of *Calliergon giganteum* in shape) and stop far from reaching the costae, but in a few Japanese specimens they are transverse-elongate. In a few other specimens (mostly also from Japan) the alar groups are composed of thick-walled cells with brownish walls. Although based on stem cross sections such specimens can be assigned to *P. orientale,* they may actually represent a different haplotype than the continental plants. The hyalodermis in *P. orientale* is less developed than in *P. appalachianum* and *P. eugyrium*, and in several specimens it is nearly absent, but other morphological traits and molecular data suggest their assignment to *P. orientale*. Most likely *P. subeugyrium* var. *japonicum* Cardot was described based on such specimens, at least the description provided by *Noguchi (1991)* in most points corresponds to *P. orientale* except for the shorter laminal cells, 25–30 μm *vs.* 30–50 μm, and smaller spores, 12–20 μm *vs.* 19–23 μm. Thus, synonymization of *P. subeugyrium* var. *japonicum* with *P. orientale* at the moment seems premature, although it seems there is no other possibility of what *P. subeugyrium* var. *japonicum* might be. In general, insular specimens are more variable than those from Primorsky Territory, but this variability does not have an obvious geographical pattern; at least, the description provided by *Kanda (1976)* for Japanese *P. eugyrium* (which actually represents *P. orientale*) clearly fits plants from Primorsky Territory and Shikotan Island.

**Differentiation.** Although morphologically *P. orientale* is reminiscent of *P. eugyrium*, the origin of the material (East Asian *vs.* Atlantic) clearly indicates if the specimen represents one or the other. Differentiation of them is considered in the diagnosis; among other useful traits coloration is worth mentioning. According to Kanda and our observation, *P. orientale* never has dark, especially brownish or blackish coloration, and just a single specimen has a pinkish color. Typically, living plants are light to bright green throughout, while in *P. eugyrium* older parts of the plants are usually darker or have a red coloration. Two other species of *Pseudohygrohypnum*, *P. sibiricum* and *P. purpurascens*, occur sympatrically with *P. orientale* and may be similar morphologically. From *P. sibiricum* the latter species usually differs in its differentiated hyalodermis and strongly concave, apiculate leaves, that are canaliculate distally. In a few cases, the hyalodermis may be misleading since it may be absent in *P. orientale* and present in *P. sibiricum*. In such cases the shorter upper leaf cells, typically 30–50 μm *vs.* 35–70 μm in *P. sibiricum,* are helpful. Due to their canaliculate, falcate leaves, long leaved morphotypes of *P. orientale* (such collections occur in Japan) may resemble *P. purpurascens*, which is especially likely to be confused with *P. orientale*, since these two species occur in the same areas and habitats. Absence of the red coloration and the presence of a stem hyalodermis and central strand differentiate *P. orientale* from *P. purpurascens*. In addition, *P. orientale* has shorter laminal cells than most species of *P. subeugyrium* affinity, excepting one of the continental morphotypes of *P. purpurascens*, described under that species.

**Ecology.** In Primorsky Territory and Shikotan Island *P. orientale* grows on wet siliceous rocks in and near creeks and in the spray zone of waterfalls at lower elevations; according to our field observations and specimens in herbaria, near waterfalls this species may be very abundant. All studied collections are restricted to an elevation range of 0–100 m on Shikotan Island and 200–400 m in Primorsky Territory; however, according to *Kanda (1976)*, in Japan this species reaches an elevation of 1,400 m.

**Distribution.** Distribution of this species is restricted to temperate East Asia and remains insufficiently known. In the Russian Far East it occurs in the southern part of Primorsky Territory, southward to the 43rd parallel and on Shikotan Island between the 43rd and 44th parallels, but according to the SDM, it also may also occur in the middle and northern parts of the Sikhote-Alin Range (Primorsky Territory), which remain insufficiently studied for bryophytes. In addition, it also occurs nearly throughout Japan, but so far has not been found on Sakhalin and the Islands of Greater Kuril Ridge, even though at least Kunashir is well studied for mosses. Although we have not seen the specimens, we would propose that records of *P. eugyrium* from Korea (*Kim, Higuchi & Yamaguchi, 2020*) and in SE China and the Sino-Himalayan Region actually represent *P. orientale*, since our SDM predicts its presence there. According to the Maxent SDM, the species distribution is strongly dependant on precipitation of the Warmest quarter (BiO18), which should be above 300–350 mm, that likely limits the species distribution westward.

**Other specimens examined (paratypes)**: Russia. Primorsky Territory, Ussurijsky State Reserve, Suputinka River upper course, Mironov Klyuch, on basalt rock, 23.X.1934, A.S. Lazarenko (LE); Shkotovsky Distr., vicinity of Anisimovka village, 21.IX.1977, L.V. Bardunov (MW9060447, LE); Olsky Distr., waterfall on Milogradovka Creek, 43.458N, 134.322N, wet cliffs, 310-370 m alt., wet cliffs and temporary flooded cliffs, rocks near waterfall, 21.VIII.2007, M.S. Ignatov 07-316, 07-318, 07-332 (MHA9046081*, 9046082, 9046083, MW9060445, 9060446, 9060448*, LE); Lazovsky Distr., Elomovskyj Klyuch Creek on the path to Benevskie Waterfalls, 43.232N, 133.751E, 290 m. alt., on boulders near stream; 6.IX.2019, V.E. Fedosov 19-2-296 (MW9116124), the same place, Kučera (CBFS:21474*), the same place, 5.IX.2013, M.S. Ignatov, E.A. Ignatova & E. Malashkina, Pr-5-12-13 (MHA9046080*). Sakhalin Province, Shikotan Island, vicinity of Dimitrova Bay, 26.VIII.2021, V.E. Fedosov & A.V. Shkurko (MW9116127*, 9116128*, 9116130, 9116131); the same area, vicinity of Malokuril'skoe Settl., 8.VIII.2021, V.E. Fedosov & A.V. Shkurko (MW9116126*, 9116134*). Japan, Hokkaido, 21.VIII.1951, A. Noguchi (MHA9056558 ex NICH), Honshu, 14.VIII.1909, Ishiba (MHA9056557 ex NICH), 6.VI.1951 (MHA9056561 ex NICH), 3.VIII.1954, A. Noguchi (MHA9056559, 9056560 ex NICH), 16.VI.1959, K. Nosoi (MHA9056562 ex NICH), Kyushu, pref. Fukuoka, Mts. Sebui, 4.V.1968, J. Amakana 6110 (MHA9056556 ex NICH). China, Sichuan, Markham County. ca 20 km E of Markham along Dai Dou river, 2675-2760 m. alt., steep NE facing slope, 30.VIII.1988, Allen 6926 (MO-5129726).

**Affinities.** Despite the morphological similarity of *P. orientale* and *P. eugyrium*, molecular data suggest that these species are not particularly close phylogenetically; according to the reconstruction inferred from plastid data, among known *Pseudohygrohypnum* species *P. orientale* is rather close to *P. calcicola*, which has a similar structure of alar groups, strongly

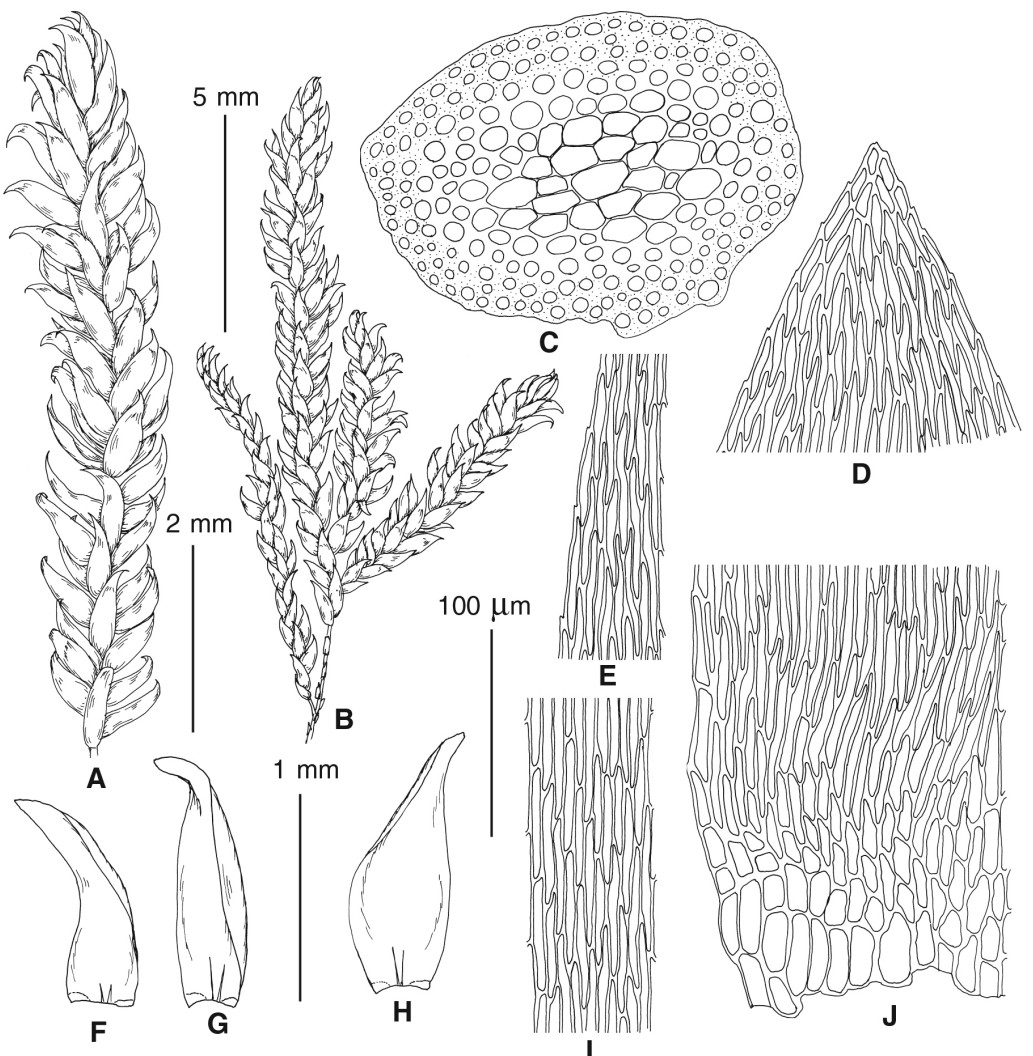

**Figure 15** *Pseudohygrohypnum subeugyrium* **(from UK, Scotland, Rothero, 2019001, dupla in MW, HyF35).** (A, B) Habit. (C) Stem transverse section. (D) Upper leaf cells. (E, I) Mid-leaf cells. (F–H) Leaves. (J) Basal leaf cells. Scale bars: 5 mm for B; 2 mm for A; 1 mm for F–H; 100 µ m for C–E, I, J.

concave leaves with rather short cells, and long perichaetial leaves. The similarity of *P. orientale* and *P. eugyrium* is likely caused by environmentally induced convergence.

*Pseudohygrohypnum subeugyrium* (Renauld & Cardot) Ignatov & Ignatova, Arctoa 11 (Supplement 2): 845. 2004. –*Hypnum subeugyrium* Renauld & Cardot, Bot. Gaz. 22(1): 52, pl. 5, f. B. 1896; Type: Newfoundland: Exploits (Rev. A. C. Waghorne, 1893), syntype: !MO-406297, http://legacy.tropicos.org/Specimen/90207675 –*Calliergon subeugyrium* (Renauld & Cardot) Kindb., Ottawa Naturalist 14: 80. 1900. –*Limnobium subeugyrium* (Renauld & Cardot) G. Roth., Eur. Laubm. 2: 648. 1905. *Hygrohypnum subeugyrium* (Renauld & Cardot) Broth, Nat. Pflanzenfam. 231[I,3]: 1039. 1908. (Figs. 4–7, 15).

**Description.** For species description see *Blockeel, Kiebacher & Long (2019)*.

**Comments on morphology.** For a long time, *P. subeugyrium* was considered to be a subspecies of *P. eugyrium* and therefore it is absent from key European checklists, manuals and floras published in 20th century (*Podpěra, 1954*; *Nyholm, 1965*; *Smith, 2004*; *etc.*). This species was revived in the monographic treatment of the genus *Hygrohypnum* by *Jamieson (1976)*, which was accepted by *Crum & Anderson (1981)* and *Corley & Crundwell (1991)*. The protologue of *Hypnum subeugyrium* is not very informative so its circumscription was further clarified by *Jamieson (1976)*. This treatment was subsequently followed by *Czernyadjeva (2003) Ignatov & Ignatova (2004)*, *Jamieson (2014)*, and finally *Blockeel, Kiebacher & Long (2019)*. However, Jamieson's description also included the taxon recognized below as *P. neglectum*. The descriptions by *Czernyadjeva (2003)* and *Ignatov & Ignatova (2004)* which are based on Russian specimens do not actually represent *P. subeugyrium* s.str. The description by Blockeel, Kiebacher & Long was based on plants from the United Kingdom, which were partly included in the present study. We are able to confirm the identity of type material of *P. subeugyrium* (duplicate at MO) with plants illustrated by *Blockeel, Kiebacher & Long (2019)*.

**Variation.** Unlike *P. eugyrium*, European and North American populations of *P. subeugyrium* have identical sequences for all studied markers excepting one american specimen, P2274, indicating either presence of gene flow between them or recent colonization of the European or North American range. The amount of studied material is not sufficient to fully assess its morphological variation; but at first glance, although the specimens studied by us originate from all parts of species range, they look quite uniform.

**Differentiation.** *P. subeugyrium* differs from related species in the combination of brownish coloration with weak purplish tint, lack of a differentiated stem central strand and hyalodermis, leaves canaliculate distally, which is especially evident in falcate leaves, and transversely elongate alar groups, composed of thick-walled cells and typically extending to the costa. In addition, this species occurs in northern Europe, where we failed to find any other *Pseudohygrohypnum* species except *P. eugyrium*, despite this area being likely the most thoroughly studied worldwide. In eastern North America the ranges of *P. subeugyrium* s.str. and *P. neglectum* (see description below) overlap, and these species may be confused. Small, round alar groups typically composed of thin-walled cells and less strongly incrassate sclerodermis are the best traits to differentiate *P. neglectum* from *P. subeugyrium*, since canaliculate leaves, typical for *P. subeugyrium*, occasionally occur in the former species. Differentiation of *P. subeugyrium* and *P. purpurascens* is considered under the latter species.

**Distribution.** *Pseudohygrohypnum subeugyrium* s.str. is an oceanic or even hyperoceanic north Atlantic species, which occurs in areas with a mild climate: SW Norway, S Sweden, UK (mostly in the NW), and eastern North America –Newfoundland (locus classicus), and the eastern states of Canada and the USA, from Quebec to Tennessee (*Jamieson, 2014*).

**Specimens examined:** Canada: Newfoundland, ca. 49.525N 55.070W, 14.XI.1893, A.C. Waghorne 26 (MO, syntype); Spruce Brook, George's Lake, St. George's, 8-9.VII.1949, R. Tuomikoski 2791 (LE, as *P. eugyrium*). USA, Maine, Waldo County, Lincolnville Town, Camden Hills State Park, 44.263N, 69.045W, 337 m alt., on rocks and roots in small stream, 4.VII.2002, Bruce Allen 24431 (MO*); New York, Essex County, Town of Keene,

Adirondack Park, cliffs and valley SW of Chapel Pond, 44.141N, 73.747W, 460 m alt., on boulder in stream, 19.IX.2004, Bruce Allen 27202 (MO*). Virginia, Nelson County. Shamokin Springs Nature Preserve, Large rock in stream, mostly above water. H. Hamilton 742, 13.8.2017, as *P. eugyrium* (DUKE*); Sweden, Närke, Tysslinge, 10.X.1992, N. Hakelier (MHA9056564* ex NICH). UK, Scotland, Allt Donachain Dalmally, 88 m alt., on rock in large burn, 25.XI.2018, G.P. Rothero 2018176 (MW* ex Herbarium GP Rothero); Corarsik Burn, Coval, 210 m alt., on rocks in burn, 20. I.2019, G.P. Rothero 2019001 (MW* ex Herbarium GP Rothero); Leacann, Water north of Furnace, 300 m alt., on rock in burn, 8.VI.2019, G.P. Rothero 2019029 (MW* ex Herbarium GP Rothero).

**Discussion.** Identity of *P. subeugyrium* var. *japonicum* Cardot is discussed under *P. orientale*.

***Pseudohygrohypnum purpurascens*** (Broth.) Kanda, (J. Sci. Hiroshima Univ., Ser. B, Div. 2, Bot. 6: 109. 1976[1977]. –*Hygrohypnum purpurascens* Broth., Öfvers. Finska Vetensk.-Soc. Förh. 62A(9): 36. 1921. = *Hygrohypnum poecilophyllum* Dixon, Rev. Bryol. Lichénol. 7: 113. 1934. ((Figs. 4–7, 12, 16).

**Description.** For description and illustration based on the holotype see *Kanda (1976)*.

**Differentiation.** *Pseudohygrohypnum purpurascens* differs from other species of the genus in having a combination of (1) purplish to deep vine-red color; (2) absence of stem central strand; (3) leaves canaliculate distally, and (4) transversely elongate alar groups (nearly) reaching the costa. Somewhat similar combinations occur in its two closest relatives, *P. subeugyrium* and *P. neglectum*. *P. subeugyrium* resembles *P. purpurascens* in all essential traits (*Jamieson, 1976*) like absence of stem central strand and hyalodermis, strongly incrassate sclerodermal cell walls, transversely elongate groups of elongate, thick-walled, brownish cells reaching the costae at leaf base, thick-walled upper leaf cells, and canaliculate upper leaf portion. The few differences between them include much weaker red coloration in *P. subeugyrium,* which is often absent, somewhat smaller spores, 12–20 μm *vs.* 15–30 μm, and amphiatlantic *vs.* East Asian distribution. The latter is the most obvious way to differentiate these species. Although *Jamieson (1976)* considered coloration to be an unreliable trait for maintaining *P. purpurascens* as a separate species, molecular data indicate their divergence and here we propose its use only for distinguishing these separate lineages. Although Jamieson noticed that similar coloration occurs in numerous specimens of North American and Scandinavian plants, this observation may be based on specimens of *P. neglectum*, which occur in North America; actually only one of eight *P. subeugyrium* s.str. specimens studied in the course of the present study had a slight pinkish coloration. At the same time, shape and denticulation of leaf apex, which was noticed by *Jamieson (1976)* as possibly a suitable trait for differentiating Atlantic (*i.e., P. subeugyrium*) and Japanese (*i.e., P. purpurascens*) plants, does not work for continental East Asian plants which often have serrulation descending up to 1/3 of the leaf length. *P. neglectum* differs from *P. purpurascens* (which often can be variegate) in somewhat larger plant size, weakly concave, not canaliculate or tubulose in upper part leaves, and round alar groups, restricted to lateral portion of the base and sharply delimited from sublinear thick-walled and porose cells of the middle basal portion. Recombinant *P. neglectum* × *P. subarcticum* specimens

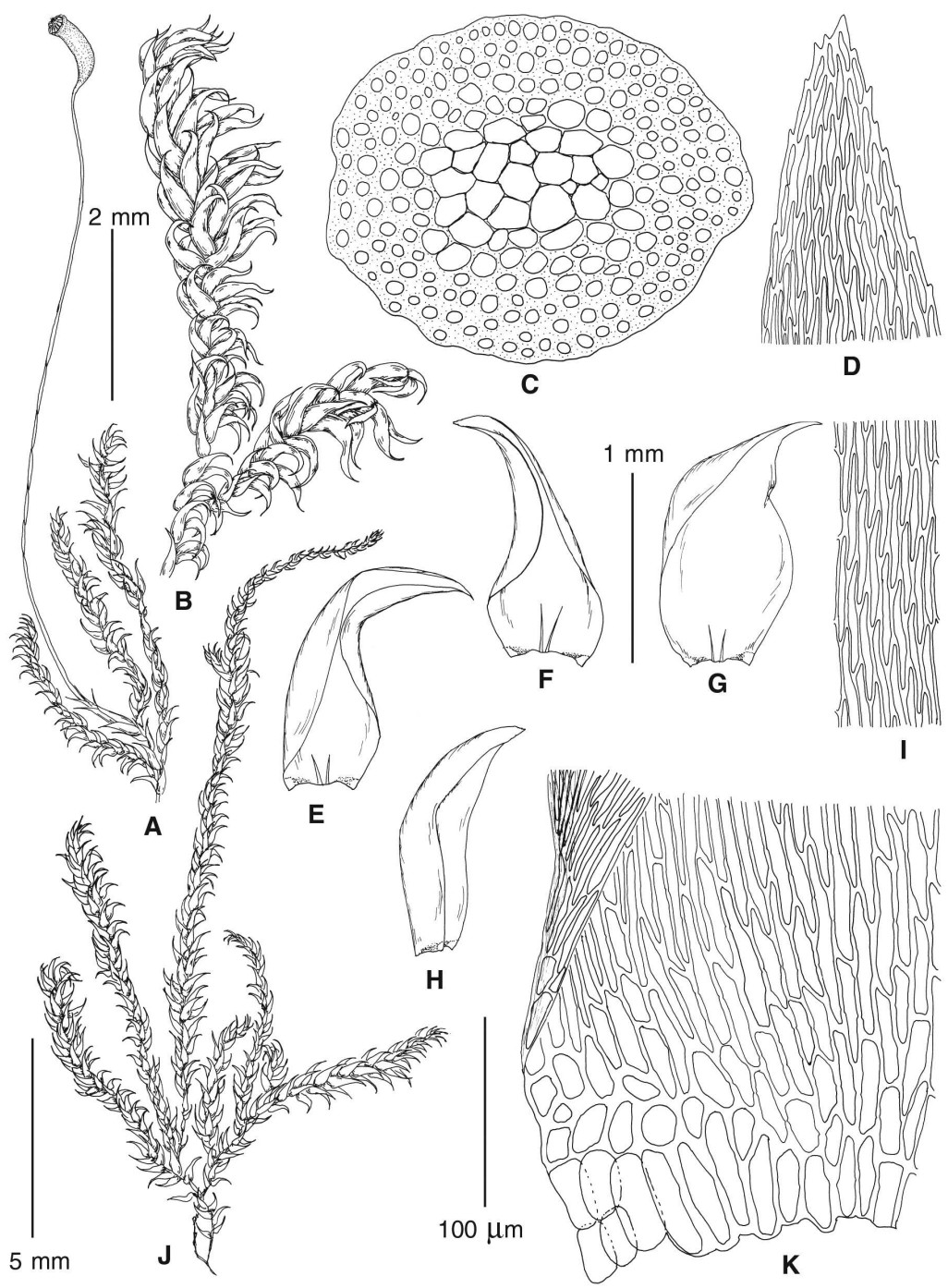

**Figure 16** *Pseudohygrohypnum purpurascens* **(from Primorsky Territory, Pidan Peak, Ignatov & Ignatova 06-2197 MW, HyF61).** (A, B, J) Habit. (C) Stem transverse section. (D) Upper leaf cells. (E–H) Leaves. (I) Mid-leaf cells. (K) Basal leaf cells. Scale bars: 5 mm for A, J; 2 mm for B; 1 mm for E–H; 100 µ m for C, D, I, K.

may look very similar to *P. purpurascens* in size, lack of central strand, leaf shape, coloration, and brown cells of alar groups. They differ in their smaller alar groups, not reaching costae, and the presence of well delimited, strongly porose cells located between the alar groups. Differentiation of *P. purpurascens* and *P. orientale* is considered under the latter species.

**Variation.** Remarkably, the description of the same species provided by *Noguchi (1991)* is quite different. According to Kanda's description, it has "median laminal cells 70–80× 3–4 μm, …alar cells usually very incrassate, sometimes arranged in a single row extending to costa in leaf base; the detailed description sporophytes is provided". This description does not contradict *Brotherus (1921)* and generally fits well for plants from our clade C, especially the specimen from Shikotan Island; however, according to our experience, *P. purpurascens* never has a central strand, while in the description provided by Kanda "central strand small" is written; since this trait is sometimes hard to define clearly, we do not consider this controversy as problematic. According to the description provided by *Noguchi (1991)*, it has "median laminal cells 85–100 × 3–4 μm, …alar cells bulging, thin-walled; sporophytes unknown".

Among eleven Japanese specimens available for our study nearly half have hyaline, thin-walled lateral alar cells, which correspond to the description of Noguchi and our plants from the southern Kuril Islands, while the remaining Japanese specimens have thick-walled, brownish alar cells that correspond to the description by Kanda and several plants from the continental part of the Russian Far East. In other respects these two groups are identical. In other words, insular plants of *P. purpurascens* often have alar groups composed of thin-walled, hyaline cells, while all continental specimens have alar groups composed of thick-walled cells. Two insular specimens with thin-walled hyaline alar cells do not form a group in the tree inferred from the cp markers, although they share specific ITS haplotype; at the moment we would prefer to consider this trait to be variable without further taxonomic recognition. Three continental specimens from Primorsky Territory have a very remarkable combination of thick-walled alar cells and wide and rather short median leaf cells, ca. 40–60 × 6–7 μm (*vs.* 60–85 × 3–5 μm in insular and other continental specimens), which are distinctly prorate ventrally. In addition, serrulation of the upper leaf portions descends in these plants well below the leaf apices, while, according to *Jamieson (1976)* and our own observation, the "insular" morphotype is typically characterized by only a few teeth near the extreme apex. These specimens have a nearly identical ITS haplotype (shared also by a "recombinant" specimen from Irkutsk Province), are basalmost within the *P. purpurascens* clade and form a strongly divergent clade on the tree inferred from cp markers. The revision of available specimens of *P. purpurascens* revealed no additional specimens from continental Asia, which share the aforementioned combination of traits and the remaining specimens from Primorsky Territory are nearly identical to insular plants, although they have thicker cell walls. In our opinion, the three specimens from Primorsky Territory discussed above may deserve taxonomic recognition at the infraspecific level. Comparison with the type of *P. poecilophyllum* is needed first, although its protologue does not agree with plants from the southern part of Primorsky Territory. Finally, two specimens from the Sino-Himalaya region are found in a distinct clade and share variegate to greenish coloration that differentiates them from all other examined

specimens of *P. purpurascens*. These plants rather resemble *P. subeugyrium* s.str., although molecular data and their distribution suggest placement in *P. purpurascens*.

**Ecology.** This species grows on rocks and boulders composed of acidic rocks in or near mountain creeks, often together with *P. orientale, Entodon luridus*, etc. According to our collections, this species is rather common in the southern part of the Sikhote-Alin Mountains in the forest belt, up to 1,100 m alt. However, places where it was collected (Elomovsky Klyuch Creek, Pidan Mountain, Ol'khovaya Mountain) are also the best studied places, since these areas are known to house the highest amount of temperate East Asian species, which apparently survived there due to the absence of massive fire events, and thus our impression of its frequency may be false. In the Kuril Islands this species occurs in the *Betula lanata* & *Sasa* or *Botryoides, Sasa, Spiraea* etc. dominated belt. In the southern part of its range this is a strictly montane species, which occurs at an altitudinal range of 2510–4090 m (*Blockeel, Kiebacher & Long, 2019*).

**Distribution.** *Pseudohygrohypnum purpurascens* occurs in East Asia: in Russia (southern Kuril Islands and the southern part of the Sikhote-Alin Mountains), throughout Japan, in South Korea, a few montane localities in China (Manchuria, Yunnan) and India (Sikkim) (*Kanda, 1976*; *Blockeel, Kiebacher & Long, 2019*; *Kim et al., 2020*) supplied by our data).

**Specimens examined:** Russia. Primorsky Territory, Lazovsky Distr., Elomovskyj Klyuch Creek on the path to Benevskie Waterfalls, 43.237N, 133.725E, 400 m. alt., on boulders near stream; 9.IX.2019, V.E. Fedosov 301 (MW9116122), the same area, 43.217N, 133.767E, ca. 250 m. alt., 6.IX.2013, E.V. Malashkina & O.V. Ivanov Pr-6-15-13 (MHA9046092); the same area, 43.245N, 133.719E, 670 m alt., rock outcrops near stream; 6.IX.2013, M.S. Ignatov & E.A. Ignatova 13-1388 (MHA9046091); the same area, Lazovsky State Reserve, middle course of Perekatnaya River, on boulders near stream, 21.IX.1974, L.V. Bardunov (LE); Shkotovsky Distr., Khalaza Mt., near stream, partly submerged, 18.X.1933, A.S. Lazarenko (LE); Livadijskaya Mt., 43.083N, 132.683E, ca. 700 m alt., on boulders near stream, 25.IX.2006, M.S. Ignatov & E.A.Ignatova 06-2197 (MW9060473*, MHA9046109, 9046110); Partizansk Distr., Ol'khovaya Mt., Kamenystyj Creek upper course, 43.345N, 133.675E, ca. 1,100 m alt., on boulder with oozing water, 13.IX.2014, V.E. Fedosov (MW9116129); Sakhalin Province, South Kuril Islands, Iturup Island, vicinity of Circ Bay, Circ Creek, 45.35N, 148.6E, ca. 50 m. alt, on boulder in stream bed; 10.IX.2015, V.E. Fedosov 15-2-047 (MW9060477*); Shikotan Island, Notoro Mt., 43.776N, 146.739E, ca. 250 m alt., wet bank of temporary stream, 25.VIII.2007, V.A. Bakalin K-42-44-07 (MW9060475*, MHA9036958). Japan. Honshu, V.1931, Y. Koyama (MHA9056574 ex NICH); 10.VII.1932 (MHA9056569 ex NICH); 18.V.1948, M. Mizutani 541 (MHA9056575, LE ex NICH); 3.VIII.1948, N. Takaki 5013 (MHA9056566 ex NICH); 16.X.1948, M. Mizutani 525 (MHA9056571, LE ex NICH); 28.VIII.1952, T. Nakajima 3057 (MHA9056572 ex NICH); 3.VIII.1954, A. Noguchi (MHA9056568 ex NICH); 22.VIII.1954, T. Nakajima 6120 (MHA9056570 ex NICH); 24.XIII.1976, Z. Iwatsukii 12052 (MHA9056567, LE ex NICH); Shikoku, 27.VII.1940, S. Hattori 124, 160 (MHA9056565, 9056573 ex NICH). China. Liaoning , Feng-cheng Co., Feng-huang Shan Mt., 8.IX.1988, X-Y Jia 686 (MO-5230506). Yunnan, Fu Gong County, Salween River watershed, 27.175N, 98.757E, 2675 m alt., on granitic boulders in river; 3.V.2004, J.R. Shevock 25337 (MO*); Gong Shan County, Cikai Xiang, QiQi He River,

27.713N, 98.502E, 2820 m alt., on granitic boulders seasonally submerged; 28.IX.2007, J.R. Shevock 30900 (MO*).

**Taxonomic comments.** Following *Jamieson*'s (*1976*) revision, *P. purpurascens* has largely been synonymised with *P. subeugyrium*, although in both the Japanese and Chinese floras (*Noguchi, 1991*; *Hu et al., 2008*) the name *P. purpurascens* is used along with or instead of *P. subeugyrium*. Actually, among the revealed species, *P. purpurascens* is most similar to *P. subeugyrium* and also the most closely related according to the cp data. However, our results indicate significant divergence of these two clearly allopatric lineages (for comments on *P. subeugyrium* var. *japonicum* see under *P. subeugyrium*).

Since our data suggest rather strong segregation of at least three molecular lineages within *P. purpurascens*, the name *P. poecilophyllum* with a Manchurian type, which was placed into synonymy with *P. purpurascens* by *Kanda (1976)*, could be used for one of the two continental lineages, but since ASAP has not suggested segregation of these lineages at the species level and the amount of material available for morphological and molecular study is quite limited, we would prefer keeping these lineage within one species, within which three varieties or subspecies might be delimited.

*Pseudohygrohypnum sibiricum* Fedosov & Ignatova, sp. nov. (Figs. 4–7, 12, 17).

**Type:** Kalarsky Distr., Kodar Mts, Srednij Sakukan River, 56.906N, 117.809E, ca. 1,060 m alt., 3.VII.2013, I.V. Czernyadjeva 13-13 (LE–Holotype; MW9077400*, MHA9046093– Isotypes);

**Diagnosis.** This species is similar to *P. orientale* in its well-developed stem central strand, ovate leaves and well differentiated, compact alar groups composed of thin to moderately thick-walled cells, but differs in its lack of a stem hyalodermis, less concave leaves that are not apiculate, longer leaf cells, and weakly inflated alar groups.

**Etymology.** The species epithet originates from the area where the range of the newly described species largely lies.

**Description.** Plants medium-sized, light- or yellowish-green to purplish-brown, not glossy or with bronze sheen, reminiscent of *Pohlia cruda*. Stems prostrate, green, brownish or reddish, to two cm, complanate, with leaves curved downward, with weakly differentiated epidermal cells (*i.e.*, with thinner walls than the inner sclerodermis cells), moderately thick, 3–4 layered sclerodermis and well differentiated central strand, irregularly to nearly pinnately branched, typically distinctly complanate. Leaves (1.2–)1.3–1.6(–1.8) × (0.5–)0.6–0.9 mm, straight or slightly falcate-secund, ovate to ovate-lanceolate, acuminate or apiculate, often blunt at tip, slightly concave to nearly plain, notably narrowed toward insertion, widest at ca. 1/5–1/3 of leaf length; costa double, rather strong, typically reaching 1/4–1/2 of leaf length, rarely (in narrower leaves) shorter; margins plane, serrulate in upper half or only near leaf apex; upper laminal cells 35–70(–85) × 4–5 μm, vermicular, thin- to moderately thick–walled, not porose, at leaf tip shorter, elongate to rhomboid; basal cells usually thick-walled, concolorous, along margin indistinctly bordered by a row of a few shorter and wider, lightly colored cells; alar groups well delimited, ovate to triangular, with base of triangle facing upward, not reaching costae, more or less inflated, composed of (1–)2–3 rows of inflated, more or less thin-walled, hyaline cells, more rarely alar

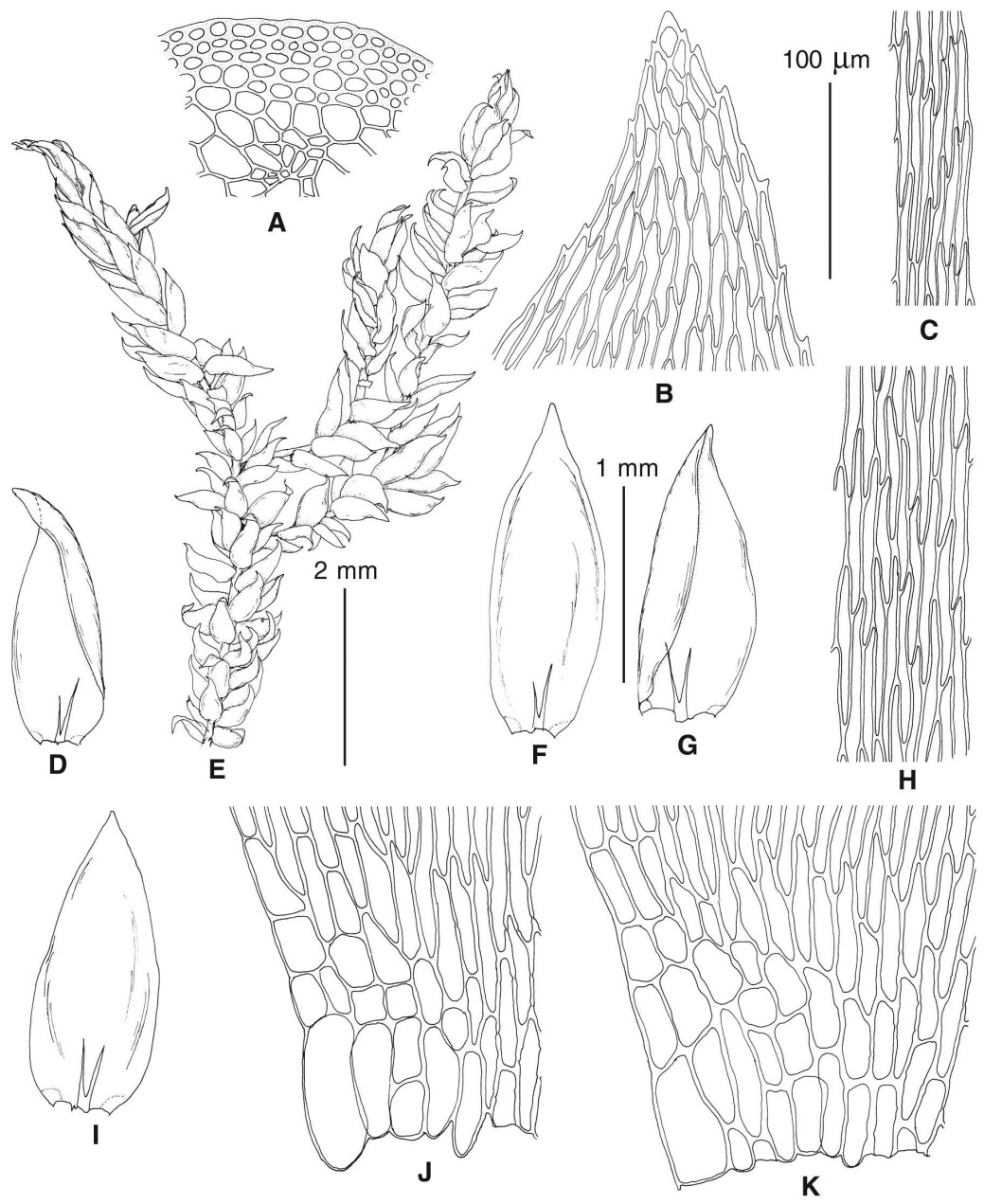

**Figure 17** *Pseudohygrohypnum sibiricum* **(from isotype MW9077400).** (A) Stem transverse section. (B) Upper leaf cells. (C, H) Mid-leaf cells. (D, F, G, I) Leaves. (E) Habit. (J, K) Basal leaf cells. Scale bars: 2 mm for E; 1 mm for D, F, G, I; 100 μ m for A–C, H, J, K.

groups composed of thick-walled cells with brown walls, or inflated, thin-walled cells may have slight brownish coloration; subquadrate supra-alar cells usually present. Autoicous. Perichaetial leaves lanceolate, acuminate, plicate. Setae reddish-brown, 1.5–2 cm. Capsules 1.5–2 mm long, inclined to horizontal, arch-like curved when young, contracted below mouth, brownish, usually longitudinally plicate. Exothecial cells moderately thick-walled,

elongate, below mouth in few rows round. Opercula conic. Annuli well differentiated, composed of 2 rows of large cells. Exostome teeth ca. 400 μm, yellowish-brown, cross-striolate proximally, strongly serrate, papillose distally; endostome of the same length or a little longer, basal membrane ca. 180 μm, papillose on the inner surface; segments carinate, perforated along a median line; cilia 2–3, filiform. Spores 13–20 μm, papillose.

**Variation.** Most specimens which represent *P. sibiricum* resemble each other although they vary in the density of branching, shape and areolation of alar cells –typically they are thin- to moderately thick-walled and in latter case brownish. However, two specimens, one from Transbaikalia and one from Badzhal Mountains (Khabarovsk Territory) have longer leaves which are more strongly falcate; alar groups in these specimens are strongly differentiated, inflated, brown, composed of thick-walled cells (that resemble *P. subeugyrium* s.str., *P. purpurascens* and "*P. neglectum* × *P. subarcticum*") that do not reach base of costae; however, sequences, obtained from both specimens are identical to "typical *P. sibiricum*" therefore, we consider these specimens as representing an unusual morphotype of *P. sibiricum*, likely ecologically induced. Most Siberian specimens and plants from Bashkortostan lack purplish coloration and have weakly falcate leaves, while several specimens from Khamar-Daban Range and Primorsky Territory are variegate to purple, and their leaves are more falcate; at the same time, the sequences from these specimens are identical to Siberian ones and no sufficient evidence of gene flow between them were revealed excepting a single recombinant specimen from the Khamar-Daban Mountains. In one specimen of *P. sibiricum* the upper branch leaves are tubulose, forming an acute branch apex which resembles that of *Calliergonella cuspidata.*

**Differentiation.** Typically, *P. sibiricum* can be recognized and differentiated from other species of *P. subeugyrium* affinity due to its complanate, yellowish- or brownish-green shoots and rather short and wide, weakly falcate leaves; it rather resembles such riparian mosses as *Hygrohypnum luridum, Stereodon pratensis* or complanate morphotypes of *Calliergonella lindbergii*. Unlike the former species, *P. sibiricum* grows on acidic rocks, while *H. luridum* usually settles on basic ones; forked costae typically reaching midleaf differentiate *H. luridum* from *P. sibiricum*. Both *Stereodon pratensis* and *Calliergonella lindbergii* differ from *P. sibiricum* in having a hyalodermis well developed throughout the stem and dioicous sexual condition. Long-leaved morphotypes of *P. sibiricum* may resemble *P. neglectum*, but they differ in having alar groups composed of thick-walled cells with brown walls. Such alar cells never occur in *P. neglectum*, but are characteristic of "recombinant *P. neglectum* × *P. subarcticum*" specimens, which, however lack any trace of central strand, are somewhat smaller and have strongly porose basal leaf cells. Differentiation from *P. orientale* is considered under that species.

**Ecology.** *Pseudohygrohypnum sibiricum* occurs on wet boulders composed of acidic siliceous rocks along streams or on wet cliffs, at middle elevations (all specimens are restricted to the forest belt). Most specimens comprise pure mats, usually encrusted by alluvium, which rarely occurs in other species of the genus. Compared with the other subaquatic *Pseudohygrohypnum, P. sibiricum* occupies lower parts of watercourses where the stream is slower, and this apparently results in more deposition of sandy alluvium.

**Distribution.** So far, *P. sibiricum* is known as an endemic of Russia, although its occurrence in Mongolia and Manchuria is quite likely. Its localities are scattered in montane areas with continental climatic conditions, predominantly along the southern border of Asiatic Russia eastward to Primorsky Territory with a single locality in European Russia (Bashkortostan) and two isolated localities in subarctic Anabar Plateau and Magadan Province. This species is locally abundant in the areas where rather xeric/continental climatic conditions with high temperature and precipitation seasonality are combined with very acidic rocks (granites or quartzites). Moreover, among all considered subaquatic species, only *P. sibiricum* demonstrates an affinity for continental areas with a rather xeric climate (including remarkably xeric Transbaikalia, where this species is rather frequent). It is hard to imagine hyperoceanic *P. subeugyrium* s.str. being able to survive in these areas. In more humid areas *P. sibiricum* becomes rarer, being partially replaced by *P. neglectum* or *P. neglectum* × *P. subarcticum* (*i.e.,* in the Altai and Badzhal Mts.), *P. purpurascens* and *P. orientale* (Primorsky Territory) or ''*P. sibiricum* × *P. purpurascens*'' in Khamar-Daban. The locality in Bashkortostan is somewhat outstanding climatically and modeling does not predict the presence of *P. sibiricum* there.

**Other specimens examined (Paratypes):** Russia. Bashkortostan Republic, Beloretzkiy Distr., South Uralian State Reserve, 54.183 N, 57.617 E, montane creek, on quartzite; 27.VII.1996, E.Z. Baisheva s.n. (MW9060449, 9060450*, LE). Altai Republic, Altaisky State Reserve, Teletskoe Lake shore, 51.667N, 87.667 E, ca. 440 m alt., on wet cliffs near waterfall; 11.VII.1989, N. Zolotukhin (MHA9046105); Ulagan Distr., Chulcha River (Chulyshman River tributary) 51.117N, 88.100E, ca. 800 m. alt., boulder near creek; 14.VIII.2012. M.S. Ignatov & Ignatova 12-854 (MW9060471*, MHA9046107); Kurkure Range, Kayakkatuyarykski, Creek basin, on wet cliffs near waterfall, 2.VI.1991, M.S. Ignatov (LE). Krasnoyarsk Territory, Taimyr Autonomous District, Anabar Plateau, Kotuykan River valley, 70.721N, 105.584E, on wet quartzite sandstone rock; 27.VII.2007, V.E. Fedosov 07-380 (MW9060459). Buryatia Republic, Kabansky Distr., Baikalsky State Reserve, Pereemnaya River valley, 51.425N, 105.294E, ca. 1,000 m alt., rock outcrop, near water, 17.VIII.1989, S.G. Kazanovsky (LE); the same area, valley of Osinovka Creek, 51.504N, 105.133E, ca. 600 m alt., on rocks near waterfall, 29.VII.2016, O.M. Afonina 0116/1 (LE); the same area, Mishikha River valley, 51.623N, 105.535E, ca. 480 m alt., on boulder near stream, 08.VIII.2016, O.M. Afonina 40716/4a (LE). Zabaikalsky Territory, Kalarsky Distr., Kodar Mts, Srednij Sakukan River, 56.906N, 117.809E, ca. 1060 m alt., 3.VII.2013 I.V. Czernyadjeva (MHA9046093*, LE); the same area, 56.907N, 117.809E, ca. 1160 m. alt., 7.VII.2013, O.M. Afonina (LE*); the same area, Levyi Syulban River basin, 56.862N, 117.288E, ca.1460 m. alt.; 11.VI.2015, O.M. Afonina (LE*); Kyrinsky Distr., Sokhondinsky State Reserve, Agutsa River valley, Zolotoy Klyuch Spring, 49.753 N, 111.253 E, ca. 1330 m alt., boulders in stream; 13.VII.2013, I.V. Czernyadjeva (LE*); the same area, near field station on Enda River, 49.450N, 110.850E, ca. 1070 m alt., 09.VII.2010 & 12.VII.2010, Czernyadjeva (LE*); the same area, ca. 30 km northward Kyra settl., 49.883 N, 112.050 E, ca. 1310 m. alt., 16.VIII.2006 O.M. Afonina (LE11606, 11707, MHA9046095). Magadan Province, Yagodinsky Distr., vicinity of Sybit-Tyellakh abandoned settlement, Olen' Creek, ca. 61.94N, 149,57E, 23.VII.1976, L.S. Blagodatskikh

(LE). Khabarovsk Territory, Khabarovsk Distr., Mountain ridge between Yarap River and its tributary, 50.285 N 134.710 E ca. 600 m. alt., stream bank; 30.VII.2016 V.E. Fedosov s.n. (MW\*). Primorsky Territory, Sikhote-Alin Mts, Dalnegorsk Distr., vicinity of Vysokogorsk –Dalnegorsk Pass, 44.494 N, 135.4 E, ca. 700 m alt., rocks in creek; 17.IX.2019, Fedosov, s.n. (MW9116123); Olsky Distr., vicinity of Milogradovo village, irrigated creek bed, 12.VIII.1977, L.V. Bardunov & V.Ya. Cherdantseva (LE); Skotovsky Distr., Livadijskaya Mt., 43.108 N, 132.693 E, ca. 400 m alt., on boulders near creek; 11.IX.2019, Fedosov, s.n. (MW\*); the same area, 43.09 N, 132.693 E, ca. 570 m alt., on boulders near creek, 13.IX.2019, Fedosov, s.n. (MW9130407); the same place and date, Kučera. (CBFS21701\*, 21737\*); Lazovsky Distr., Elomovskyj Klyuch Creek on the path to Benevskie Waterfalls, 43.237N, 133.725E, 400 m. alt., on boulders near stream; 9.IX.2019, Kučera (CBFS21434\*, 21478\*).

**Note:** One specimen tested using molecular markers combines an ITS sequence characteristic of the northern continental lineage of *P. purpurascens* with plastid markers identical to those of *P. sibiricum*. Morphologically it largely corresponds to *P. sibiricum* and has purplish coloration, which is not characteristic of this species. The northern slope of the Khamar-Daban Range, where it was collected, although situated in the central part of southern Siberia, has rather mild and humid climatic conditions due to the influence of Lake Baikal. Irkutsk Province, Slyudyanka District, Khamar-Daban Range, valley of Snezhnaya River 5.31 km SSE of the road bridge over Snezhnaya at Vydrino, 51.394N, 104.654E, ca. 500 m alt., inundated siliceous boulder at the riverbank, 31.VIII.2018, J. Kučera 20629 (CBFS) Interestingly, the northern slope of the Khamar-Daban Range is the only area in Siberia where the presence of *P. purpurascens* was predicted by SDM, well isolated from the East Asian Range of this species. Its penetration there may be dated from nearly the same time as "relict" populations of nemoral vascular plants concentrated in Khamar-Daban (*Chepinoga, Protopopova & Pavlichenko, 2017*), *i.e.,* in the late Pliocene. Since that time, isolated due to further climate cooling, populations of *P. purpurascens* in Khamar-Daban may have been assimilated by *P. sibiricum*, which currently occurs sympatrically, but the continental *P. purpurascens* ITS haplotype still persists there.

***Pseudohygrohypnum subarcticum*** **Fedosov & Ignatova, sp. nov.** (Figs. 4–7, 12, 18).
**Type:** Krasnoyarsk Territory, Taimyrsky Autonomous District, Anabar Plateau, Kotuykan River upper course, slope of Merkyu River valley, 70.446N, 106.464E, ca. 250 m. alt., dolerite rock outcrops, on fine soil; 19.VII.2011, V.E. Fedosov 11-1237 (Holotype MW9060458\*).
**Diagnosis.** This species is similar to *P. subeugyrium* in its lack of purplish coloration, absence of stem central strand and hyalodermis and falcate leaves, but differs from it by its smaller size, weakly differentiated alar groups and rostrate opercula.
**Etymology.** The species name is given due to its distribution in northern Siberia, mostly restricted to Putorana and Anabarskoe Plateau, which are situated near the northern limit of larch forests in the subarctic climatic zone.
**Description.** Plants small, golden-brownish, more rarely light- or yellowish-green to brownish, usually with very remarkable golden sheen. Stems prostrate, yellow, brownish or reddish, to 1.5 cm, with undifferentiated epidermal cells, thick 2–4 layered sclerodermis

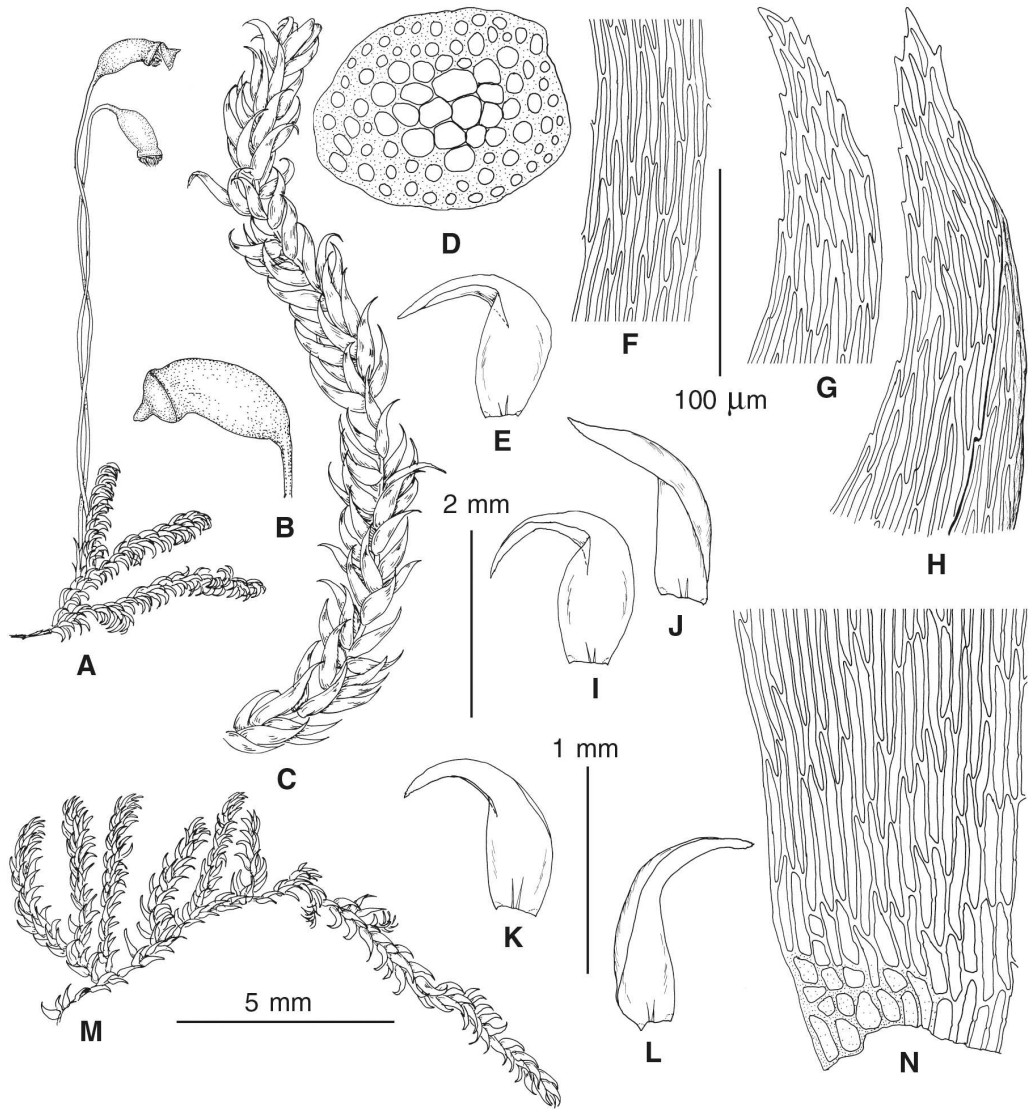

**Figure 18** *Pseudohygrohypnum subarcticum* **(from holotype).** (A, C, M) Habit. (B) Capsule. (D) Stem transverse section. (E, I–L) Leaves. (F) Mid-leaf cells. (H) Upper leaf cells. (N) Basal leaf cells. Scale bars: 5 mm for A, M; 2 mm for B, C; 1 mm for E, I–L; 100 μ m for D, F–H, N.

and no central strand, densely pinnately or, more rarely, irregularly branched. Leaves (0.65–)0.7–0.9(−1.0) × 0.2−0.35 mm, strongly falcate-secund, lanceolate, rarely ovate-lanceolate, acuminate, with blunt tips, rarely short apiculate, concave to slightly canaliculate, rarely nearly plain, not narrowed or slightly and gently or abruptly narrowed toward insertion, widest at ca. 1/9–1/5 of leaf length; costa double, very weak, typically reaching 1/10–1/7 of leaf length; margins plane, denticulate in upper half or throughout; upper laminal cells 50–80 × 3−4.5 μm, linear, not flexuose, thin- to moderately thick-walled, not porose, at leaf tip shorter, elongate to rhomboid, with thicker walls; basal cells usually thick-walled, concolorous, often with porose cell walls; alar groups indistinctly delimited,

not inflated, composed of few large, brown, thick-walled, short rectangular or elliptic cells, among which the marginal cell is usually thinner-walled, not reaching the costa; an excavated row of thick-walled brownish cells along insertion usually present; few small subquadrate supra-alar cells usually present. Autoicous. Perichaetial leaves lanceolate, acuminate, plicate, up to 1.6 mm long. Setae red, 1–1.5 cm. Capsules 1.5–2 mm long, inclined to horizontal, curved, contracted below mouth, yellowish-brown, smooth. Exothecial cells moderately thick-walled, elongate, below mouth rounded in few rows. Opercula conic-rostrate. Annuli well differentiated, composed of large, inflated cells. Exostome teeth up to 450 μm, yellowish-brown, cross-striolate proximally, papillose distally; endostome of the same length or a little longer, basal membrane ca. 200 μm, weakly papillose on the inner surface; segments carinate, perforated along a median line, papillose on the inner surface; cilia 2, filiform. Spores 15–19 μm, weakly papillose. This species was also illustrated and described by *Czernyadjeva (2003)* as *P. subeugyrium*.

**Variation.** Although all studied specimens of *P. subarcticum* and two "recombinant *P. subarcticum* × *P. neglectum* specimens" have identical ITS haplotypes, sequences from the cp markers in *P. subarcticum* are different even though all the *P. subarcticum* s.str. specimens sampled for molecular study came from a rather small area (Taimyrsky District). Studied specimens from Anabar Plateau, SE Taimyr bear apomorphic characters that result in their grouping, while specimens from Byrranga and Putorana occupy a basal position within the *P. subarcticum* clade. Morphological variation is rather insignificant within this species. Plants from shaded environments typically have greener coloration, thinner and less porose cell walls. Of the variability that is apparently not induced environmentally, the leaf apices are worth mentioning. Typically, *P. subarcticum* has blunt leaf tips; however, a few specimens have fairly acuminate leaf tips, which hardly agree with the traditional morphological concept of the genus *Pseudohygrohypnum*.

**Differentiation.** In having rather long, typically strongly falcate leaves, pinnate branching, golden sheen, thick-walled brown alar cells, and growth on moist rocks often not associated with watercourses, *P. subarcticum* resembles *Campylium bambergeri* rather than any other *Pseudohygrohypnum* species, and a collection of this species made by L.V. Bardunov indeed was referred by him to *C. bambergeri*. From this species *P. subarcticum* differs in its smaller size (leaves shorter than one mm *vs.* 1.2–2 mm), strongly denticulate leaf apices and autoicous sexual condition. The combination of small plants, weakly differentiated alar cells, well-developed annulus, lack of purplish pigmentation, and no stem central strand or hyalodermis differentiate *P. subarcticum* from all other *Pseudohygrohypnum* species.

**Ecology.** Unlike the other species of *P. subeugyrium* affinity, *P. subarcticum* often occurs on moist bases of cliffs and boulders, in shady niches in rock fields and montane rocky tundra throughout its altitudinal range. Moreover, it is not rare in plateau summit areas and near snowfields where vegetation is nearly absent due to severe environments. When growing along streams, it inhabits acidic rocks, like other related species, but in other ecotopes it prefers basic siliceous rocks, *i.e.,* basalts, dolerites, ijolites, *etc.*, often growing with species of the genera *Schistidium, Grimmia, Andreaea, Hymenoloma, Isopterygiopsis, Bryoerythrophyllum, Encalypta, Distichium*, and saxicolous hepatics.

**Distribution.** The distribution of *P. subarcticum* looks quite intriguing. This species mostly occurs in continental subarctic and Arctic mountains in Northern Siberia; it is very frequent on Putorana and Anabarskoe Plateaus, as well as in the central part of the Byrranga Range, the northernmost continental mountain system. Just a few localities were found in the Altai mountains and in Yakutia; moreover, although plants from these localities fit well the morphological concept of *P. subarcticum*, they may represent "recombinant haplotypes" which combine the ITS of *P. subarcticum* with cp markers characteristic of *P. neglectum*. Despite the first author's special search, this species has been not been found either in the Dixon area or in Polar Ural, despite environments seemingly quite similar to those where it occurs in central Byrranga, Putorana and Anabar Plateau. It is also hard to imagine that this remarkable species could be overlooked in well studied subarctic areas such as the Kola Peninsula, Yakutia and Alaska; therefore we assume that this species does not occur there despite the model predicting its presence. Somewhat similar overall distributions have been found for several recently discovered species such as *Orthotrichum hyperboreum* Fedosov & Ignatova*, Orthothecium remotifolium* Ignatov & Ignatova, *etc.*, but they still differ in details.

**Other specimens examined (Paratypes):** Russia. Altai Republic: Shebalino Distr., Karakolskie Lakes, 51.483N, 86.433E, ca. 1900 m. alt., subalpine belt, moist rock outcrops; 3.VIII.1991, M.S. Ignatov 28/74 (MHA9046106). Krasnoyarsk Territory, Taimyrsky Autonomous District: Byrranga Range on the northern shore of Taimyr Lake, Ledyanaya Bay, 74.481N, 99.770E, ca. 150 m alt., moist niche of dolerite cliff; 30.VII.2004, V.E. Fedosov (MW9060465*), the same area, ca. 180 m. alt., moist niche of dolerite cliff near the top of rocky ridge; 12.VIII.2004, V.E. Fedosov (MW9060469); Putorana Plateau, northern shore of Ajan Lake, 69.267N, 93.483E, ca. 400 m. alt.; forest belt, in water of stream, 4.VIII.1983, I.V. Czernyadjeva (LE, MHA9046094, MW9060468, 9060451); the same area, 19.VII.1983, I.V. Czernyadjeva 65 (LE); the same area, 8.VII.1983, I.V. Czernyadjeva 15 (LE); Ajan Lake southern shore, I.V. Czernyadjeva, 28.VII.1984-15.VIII.1984, ##46, 50, 58, 61, 82, 85, 89, 102, 103 (LE); vicinity of Kapchuk Lake, subalpine belt, on moist rock, VIII.1978, R. Vilde (LE); Glubokoe Lake, northern shore, 69.307N, 90.101E, ca. 200 m. alt, on fine soil at cliff base; 30.VII.20015, V.E. Fedosov 15-0763 (MW9060452*); Kotuyskoe Plateau, Kogotok Creek lower course, 70.810N, 100.961E, ca. 100 m alt., shaded crevice of dry andesite cliff; 2.VII.009, V.E. Fedosov 09-281 (MW9060462), same area, Maymecha River valley 11 km upstream Chopko River Mouth, 70.723N, 101.234E, ca. 150 m. alt., on boulder near creek; 23.VI.2009 09-381 (MW9060463); middle course of Kotuy River near Medvezhja River mouth, Plateau with altitudinal mark 347 m, 71.119N, 102.657E, ca. 300 m. alt., moist cliff crevice; 13.VII.2005, V.E. Fedosov 05-485 (MW9060467), same area, Odikhincha Mt., 70.930N, 103.039E, ca. 620 m alt., rocky montane tundra, on soil; 13.VIII.2011, V.E. Fedosov 11-983 (MW9060453); same area, vicinity of Kotuykan River mouth, 70.570N, 103.563E, ca. 120 m alt., onboulder near stream, 9.VIII.2011, V.E. Fedosov 11-1437 (MW9060454*); lower course of Kotuy River near Kayak settl., 71.522N, 103.007E, ca. 30 m. alt, moist cliff niche; 23.VIII.2007, V.E. Fedosov, 07-813 (MW9074994); Khara-Tas Range in NW periphery of Anabar Plateau, Longdoko Mt. 71.688N, 104.887E, ca. 500 m. alt., rockfield on the top surface of plateau, on finesoil;13.VIII.2006, V.E. Fedosov, 06-654, 06-673, 06-699, 06-722, 06-720 (MHA9046089, 9046087, 9046088, MW9060470*,

9075168), the same area, vicinity of Fomich River mouth, plateau with altitudinal mark 386 m, 72.062N, 110.19E, ca. 350 m alt., rockfield on the top of Plateau, on moist shaded surface of boulder, 13.VII.2008, V.E. Fedosov 08-8, 08-469 (MW9060464*, MHA9046086); Anabar Plateau, Kotuykan River valley upstream Merkyu River mouth, 70.527N, 105.909E, ca. 250 m alt., on boulders near water; 14.VII.2007 V.E. Fedosov 07-383 (MW9060460), the same place, 20.VII.2007, V.E. Fedosov 07-362 (MW9060461), the same area, Burdur Creek lower course, 70.541N, 105.854E, ca. 240 m alt., shaded base of sandstone boulder; 17.VII.2007, V.E. Fedosov 07-405 (MW9060455); Kotuykan River upper course, 70.471, 106.383, ca. 250 m alt, 16.VII.2011, V.E. Fedosov 11-690, 11-741 (MW9060456, 9060457), 19.VII.2011, V.E. Fedosov 11-1237 (MW9060458*). Evenkiya Distr: Southern periphery of Putorana Plateau, vicinity of Beldunchana Lake, ca. 67.89N, 95.74E, ca. 330 m alt., on boulders near lake; 27.VII.1971, L.V. Bardunov s.n. (LE). Yakutia Republic: Tomponsky Distr., Suntar-Khayata Range, At-Moole Creek, 63.11N, 138.62E, ca. 770 m. alt., wet cliffs; 22.VII.2003, E.I. Ivanova & V.I. Zolotov s.n. (MHA9020464).

*Pseudohygrohypnum neglectum* Fedosov & Ignatova, sp. nov. (Figs. 4–7, 19).

**Type:** Khabarovsk Territory, southern spurs of Badzhal'sky Range, Yarap River in its upper course; 50.285 N 134.713 E, 580 m alt., on wet and submerged boulders along a stream. 2.VIII.2016 V.E. Fedosov & O.Yu. Pisarenko (Mosses Of The Russian Far East Exsiccatae Fasc. II. No. 62), MW9077401 (holotype), MHA, LE, NSK?, VGBI?, KRAM?, MO?, CAS? (see Differentiation) (isotypes).

**Diagnosis.** This species is similar to *P. purpurascens* in its weakly developed to absent stem central strand and epidermis, falcate leaves and variegate, pink to purplish coloration, but differs by (1) larger size of plants; (2) slightly concave, not tubulose distally, narrowed toward insertion leaves; (3) alar groups composed of inflated, hyaline, thin-walled cells; (4) not excavated basalmost cells between alar groups; and (5) somewhat larger spores.

**Etymology.** This epithet is chosen since this species actually occurs in eastern North America but has not yet been separated from *P. subeugyrium* s.str. despite rather clear morphological differentiation. It no doubt would have remained neglected had the molecular data presented here not suggested it be recognized.

**Description.** Plants medium-sized to large, green yellowish-green or often variegate, purplish-green with light green leaf bases and pink to dark purplish distal leaf portions. Stems prostrate, green, brownish or reddish, to four cm, weakly complanate, with weakly differentiated epidermal cells (*i.e.,* with thinner walls than the inner sclerodermis cells), moderately thick 2–3(–4)-layered sclerodermis and a poorly or undifferentiated central strand, weakly and irregularly branched. Leaves (1.3–)1.4–1.8(–2.0) × 0.45−0.7mm, falcate-secund, ovate-lanceolate, acuminate, moderately concave to nearly plain, notably narrowed toward insertion widest at ca. 1/7–1/3 of leaf length; costa double, weak, 1/7–1/5 of leaf length, rarely reaching 1/3 of leaf length; margins plane, denticulate in upper half; upper laminal cells 40–75× 4–5 µm, vermicular, moderately to strongly thick-walled, subporose, at leaf tips shorter, elongate; basal cells usually strongly porose, concolorous, along margin indistinctly bordered by a row of a few shorter and wider, cells; alar groups well delimited, rounded, more or less inflated, composed of 2–3 rows of inflated, more or less thin-walled,

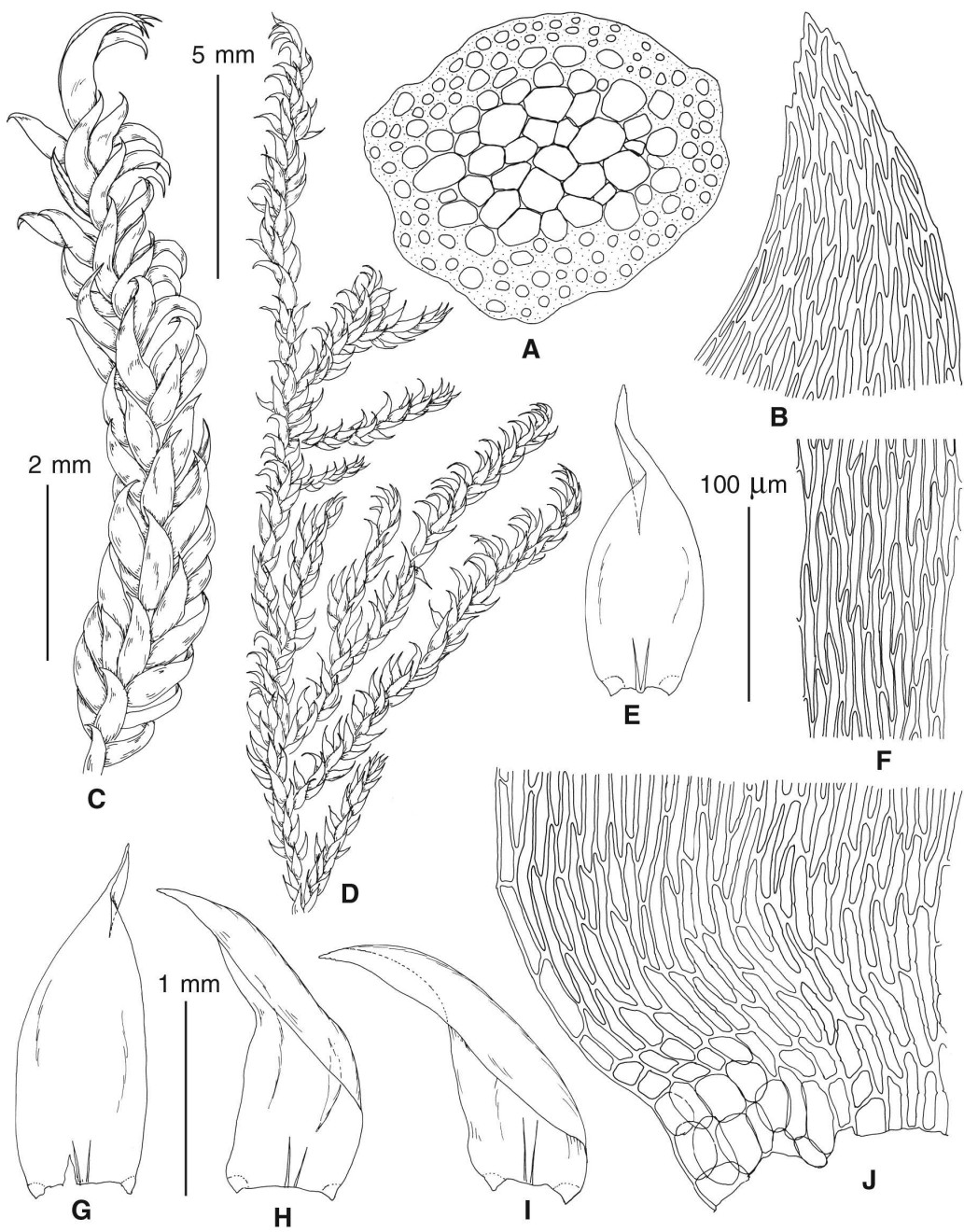

**Figure 19** *Pseudohygrohypnum neglectum* **(from holotype).** (A) Stem transverse section. (B) Upper leaf cells. (C, D) Habit. (E, G–I) Leaves. (F) Mid-leaf cells. (J) Basal leaf cells. Scale bars: 5 mm for D; 2 mm for C; 1 mm for E, G–I; 100 μ m for A, B, F, J.

hyaline to concolorous cells, not reaching costa; subquadrate supra-alar cells usually present. Autoicous. Perichaetial leaves lanceolate, acuminate, plicate. Setae reddish-brown, 1–1.5 cm. Capsules 1.5–2 mm long, inclined to horizontal, curved, contracted below mouth, brownish, usually longitudinally plicate. Exothecial cells moderately thick-walled, elongate,

below mouth rounded in a few rows. Opercula conic-apiculate. Annuli differentiated, composed of large, inflated cells. Exostome teeth up to 450 μm, yellowish-brown, cross-striolate proximally, papillose distally; endostome of the same length or a little longer, basal membrane ca. 150 μm, papillose on the inner surface; segments carinate, perforated along a median line; cilia 2, filiform, appendiculate. Spores 12–18 μm, papillose.

**Variation.** Asian plants of *P. neglectum* look quite uniform, varying only in coloration: a specimen from the Altai Mountains and one of the specimens from Khabarovsk Territory have bronze rather than pink or purplish coloration and somewhat longer leaves; a single North American specimen also has a rather poorly expressed pinkish aspect, although in other respects it fits well the morphological concept of *P. neglectum*. The alar region varies even on a single stem, in a few leaves a purplish to light brownish alar group may occur; in a few specimens it is more poorly delimited, concolorous, composed of cells with thicker walls. The stem central strand in the type specimen is absent, while the specimens from Dusse-Alyn Range (Khabarovsk Territory), Maine and the Altai Mountains (South Siberia) have an indistinctly differentiated central strand.

**Differentiation.** *P. neglectum* typically differs from all other species of *Pseudohygrohypnum* in the combination of pink to purplish coloration, weakly concave leaves and inflated, round alar group, which does not reach the costa, composed of thin-walled hyaline cells. This combination, together with the rather large size of the plants, indistinct stem central strand, falcate leaves with denticulate margins, and thick-walled, porose basal laminal cells makes *P. neglectum* remarkably similar to *Calliergonellopsis dieckii,* which differs from *P. neglectum* in having a well differentiated stem hyalodermis and dioicous sexual condition. According to the phylogenetic tree inferred from the cp data, *P. neglectum* is close to *P. subarcticum*. However, these two species are quite easy to distinguish since the plant sizes are very different and their leaf length ranges do not overlap; alar groups in *P. neglectum* are well differentiated, composed of thin-walled cells, while in *P. subarcticum* they are weakly differentiated, composed of few thick-walled cells; in addition, *P. subarcticum* has a quite specific golden sheen, while plants of *P. neglectum* are not glossy, but usually have pink to purplish coloration. Its rounded alar group composed of thin-walled hyaline cells, not reaching costa and well delimited from the cells of the medial portion of the insertion, lack of canaliculate aspect of the upper portion of the leaf as well as its larger plant size and less incrassate sclerodermis differentiate *P. neglectum* from *P. purpurascens* and *P. subeugyrium*, which may occur in the same areas. *Pseudohygrohypnum neglectum* may be hard to differentiate from *P. sibiricum*, which also may have an inflated alar group composed of hyaline cells but usually has shorter, less falcate leaves with shorter upper leaf cells and only seldomly has a purplish tint. In addition, the central strand in *P. sibiricum* is typically well differentiated, while in *P. neglectum* it is absent or weak, and most Siberian specimens of *P. sibiricum* possess a more or less distinctly complanate habit that is not characteristic of *P. neglectum*. It may also resemble *Calliergonella lindbergii*, although it usually differs from it by having a pink tint. In the field, the somewhat larger plants of *P. neglectum* should help separate it from other species of the *P. subeugyrium* group. Together with *P. neglectum* recombinant plants which bear ITS sequences characteristic of *P. subarcticum* and cp markers of *P. neglectum* occur, and several duplicates of "Mosses

of the Russian Far East Exsiccatae" No. 62 actually comprise such plants instead of *P. neglectum*. One such specimen kept in CBFS was demonstrated using molecular data. These recombinant plants are much smaller than *P. neglectum*, with leaves up to 1.2 mm long; they also differ from *P. neglectum* by having a more weakly delimited alar group, composed of quadrate cells with brownish cell walls that are not inflated.

**Ecology.** *Pseudohygrohypnum neglectum* occurs on wet boulders composed of acidic siliceous rocks along streams at middle elevations reaching the subalpine birch krummholz zone. Typically, it forms pure mats, or grows admixtured with *Hygrohypnella polaris* (Lindb.) Ignatov & Ignatova, *Sciuro-hypnum plumosum* (Hedw.) Ignatov & Huttunen and *Codriophorus* spp., often encrusted by sandy alluvium and diatoms.

**Distribution.** *P. neglectum* has a disjunctive "east-eastern" cool temperate distribution, which is still insufficiently known since quite a limited amount of material is available for study. At the same time, with ca. 10 specimens collected in NE part of Khabarovsk Territory in two rather well studied areas, we may assume that this species is rather common there; two isolated localities are known from the vicinity of Teletskoe Lake in the Altai Mts, and in Maine, USA. Limited distribution in North Asia with remote occurrence of *P. neglectum* in eastern North America and its plesiomophic position in the tree inferred from ITS suggest that *P. neglectum* is rather close to the ancestor of this clade and may be considered relictual. The east Asian early diversification of the genus *Pseudohygrohypnum* and most likely of the *P. subeugyrium* clade, supports this view. In addition, the areas where *P. neglectum* occurs in Khabarovsk Territory are home to several remarkable relict species like *Apotreubia hortonae* R.M. Schust. & Konstant., *Cryphaea amurensis* Ignatov, *Actinothuidium hookeri* (Mitt.) Broth., etc (*Ignatov et al., 2000*; Bakalin, pers. comm.). Although, quite a limited number of specimens was available for study during the MS preparation, further records of *P. neglectum* in North America are expected, especially in the western part of the North American range of *P. subeugyrium* s.l.

**Other species examined (Paratypes):** Russia. Altai Republic: vicinity of Teletskoe Lake, Bolshoe Istyube Creek, ca. 51.783N, 87.450E, ca. 570 m. alt., wet cliffs; 22.VII.1991. M.S. Ignatov (MHA9046107). Khabarovsk Territory, Khabarovsk Distr., Mountain ridge between Yarap River and its tributary, 50.338 N 134.656 E ca. 1300 m. alt., stream bank; 9.VIII.2016 V.E. Fedosov s.n. (MW9112895); the same area, Yarap River valley, 50.285N 134.710 E ca. 600 m. alt., on wet boulder near stream, 30.VII.2016, V.E. Fedosov 16-27 (MW9112894), the same place and date, O.Yu Pisarenko (NSK2006566); Verknebureinsky Distr., Bureinsky State Reserve, Dusse-Alin Range, Medvezh'e Lake, 52.083N 135.016 E ca. 1,700 m alt., on wet W-facing cliff; 10. VIII.1997 M.S. Ignatov 97-571 (MHA9046090, 9046102, 9046103), Ignatov 97-575 (MHA9046098); the same area, Lednikovyj Klyuch Creek basin, 52.117N, 134.416E, ca. 1420 m alt., on moist boulder on the bottom of temporary stream; 24.VIII.1987, Petelin D.A. (MHA9046104). USA. Maine. Franklin County, Town of Eustis; along Tim Brook near Tim Pond; 5.VII.2008, B. Allen 29078 (MO).

***P. neglectum*** × ***P. subarcticum*** (Fig. 4, 6, 20).

Although only three specimens which combine ITS sequences characteristic of *P. subarcticum* and plastid sequences of *P. neglectum* were detected using molecular data, they

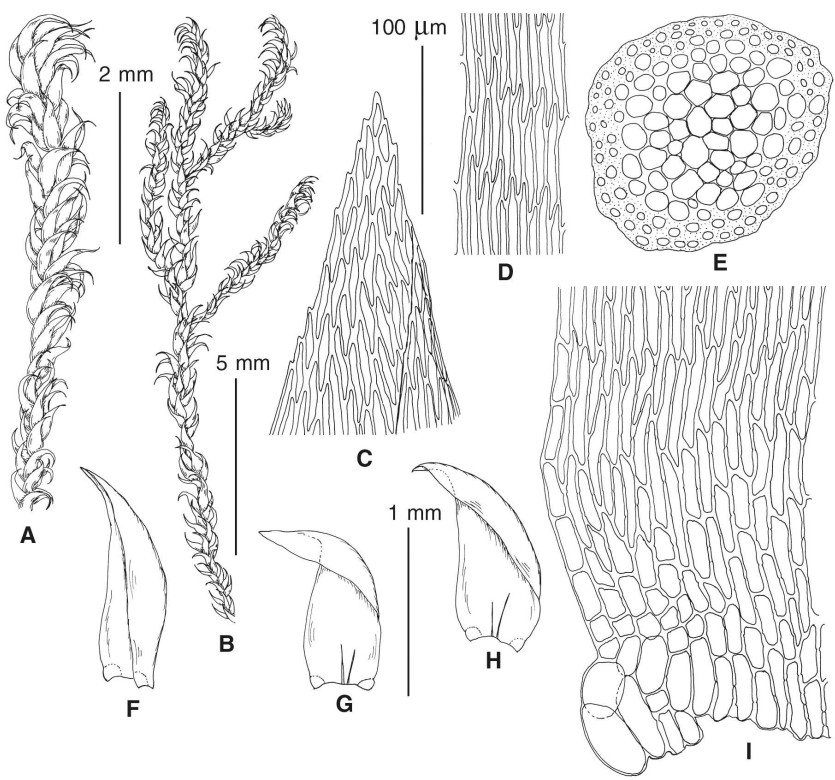

**Figure 20** *Pseudohygrohypnum subarcticum* × *P. neglectum* **(from HyF5 Yakutia, Kyurbelyakh, Ignatov & Ignatova 11-2242 MW, HyF5).** (A, B) Habit. (C) Upper leaf cells. (D) Mid-leaf cells. (E) Stem transverse section. (F–H) Leaves. (I) Basal leaf cells. Scale bars: 5 mm for B; 2 mm for A; 1 mm for F–H; 100 μ m for C–E, I.

are quite distinct morphologically due to a combination of (1) small plant size (although somewhat larger than in *P. subarcticum*); (2) brownish, pinkish to purplish coloration; (3) strongly falcate leaves; (4) distinct alar groups composed of quadrate cells with thick, brownish walls which ascend along leaf margins, and thus have a triangular shape; (5) linear cells with thick and typically strongly porose walls between alar groups; (6) often pinnate branching resembling that of *P. subarcticum*. The revision of herbarium collections revealed at least five additional specimens.

**Specimens examined:** Russia. Zabaikalsky Territory, Kalarsky Distr., Kyrinsky Distr., Sokhondinsky State Reserve, upper course of the Larionov Klyuch Creek, 49.784N, 111.242E, ca. 1440 m alt., rocky outcrops; 14.VII.2013 O.M. Afonina 4713 (LE). Yakutia, Tomponsky Distr., Kyurbelyakh Creek valley, 63.117N, 139.033E, ca. 900 m alt., in creek, 9.VII.2011, M.S. Ignatov & E.A. Ignatova 11-2242 (MW9060472*, MHA9046085); Khabarovsk Territory, Khabarovsk Distr., Yarap River valley, 50.303N 134.739 E ca. 600 m. alt., stream bank; 12.VIII.2016, O.Yu. Pisarenko s.n. (NSK2006568); the same area, 50.296N 134.718 E ca. 580 m. alt., stream bank; 2.VIII.2016, O.Yu. Pisarenko s.n. (NSK2006567); Verknebureinsky Distr., Bureinsky State Reserve, Dusse-Alin Range, Kuraigagna Creek, 52.067N, 134.867E, 1540 m. alt., rocky slope; 4.VIII.1992, B. Borisov (MW9060476*);

the same place, 1050 m. alt., rocks near waterfall, 14.VIII.1997 M.S. Ignatov 97-576 (MHA9046099); the same area, Medvezh'e Lake, 52.083N 135.016 E ca. 1700 m alt., wet cliff crevice; 9.VIII.1997 M.S. Ignatov s.n. (MHA9046096).

Thus, these "recombinant" specimens occur in NE Asia in the same areas as *P. neglectum* or between the ranges of *P. neglectum* and *P. subarcticum*. Most likely, these plants originated as a result of hybridization/introgression until climate oscillation, and further speciation divided the ranges of these species by a large area of extremely harsh climate, apparently not populated by *Pseudohygrohypnum* at all.

## ACKNOWLEDGEMENTS

We are grateful to Tom Blockeel, Olga Pisarenko and Blanka Aguero (herbarium DUKE) for sending specimens for molecular phylogenetic studies and to three anonymous reviewers for very useful comments which helped to improve the text of the manuscript. The work on SEM was performed at the User Facilities Center of M.V. Lomonosov Moscow State University. The research was carried out using the equipment of MSU Shared Research Equipment Center "Technologies for obtaining new nanostructured materials and their complex study" and purchased by MSU in the framework of the Equipment Renovation Program (National Project "Science").

### Funding

The work of Vladimir E. Fedosov, Anna V. Shkurko, Alina V. Fedorova, Elena A. Ignatova and Michael S. Ignatov was supported by RSF 18-14-00121. Molecular analyses performed by Jan Kučera at University of South Bohemia were funded by the institutional sources of the Faculty of Science. Computational resources ('Metacentrum VO') were supplied by the Ministry of Education, Youth and Sports of the Czech Republic under the Projects CESNET (Project No. LM2015042). The work of Vladimir E. Fedosov was also supported by contract # -20-120031990012-4 of the Botanical Garden-Institute FEB RAS. The work of Anna V. Shkurko and Alina V. Fedorova was also supported by Tsitsin Main Botanical Garden state assignment no. 118021490111–5 and 19-119012390082-6. We also thank the Ministry of Higher Education and Science of the Russian Federation for support and the Center of Collective Use "HerbariumMBG RAS" (grant 075-15-2021-678). The work on SEM was performed at the User Facilities Center of M.V. Lomonosov Moscow State University with financial support from the Ministry of Education and Science of Russian Federation. The funders had no role in study design, data collection and analysis, decision to publish, or preparation of the manuscript.

### Grant Disclosures

The following grant information was disclosed by the authors:
RSF 18-14-00121.
The institutional sources of the Faculty of Science.

Ministry of Education, Youth and Sports of the Czech Republic under the Projects CESNET (Project No. LM2015042).

The Botanical Garden-Institute FEB RAS: # -20-120031990012-4.

Tsitsin Main Botanical Garden state assignment no. 118021490111–5 and 19-119012390082-6.

The Ministry of Higher Education and Science of the Russian Federation: 075-15-2021-678.

The Ministry of Education and Science of Russian Federation.

## Competing Interests

The authors declare there are no competing interests.

## Author Contributions

- Vladimir E. Fedosov conceived and designed the experiments, performed the experiments, analyzed the data, prepared figures and/or tables, authored or reviewed drafts of the paper, and approved the final draft.
- Anna V. Shkurko and Evgeniya N. Solovyeva analyzed the data, prepared figures and/or tables, and approved the final draft.
- Alina V. Fedorova performed the experiments, authored or reviewed drafts of the paper, and approved the final draft.
- Elena A. Ignatova performed the experiments, prepared figures and/or tables, authored or reviewed drafts of the paper, and approved the final draft.
- John C. Brinda performed the experiments, authored or reviewed drafts of the paper, and approved the final draft.
- Michael S. Ignatov analyzed the data, authored or reviewed drafts of the paper, and approved the final draft.
- Jan Kučera performed the experiments, analyzed the data, authored or reviewed drafts of the paper, and approved the final draft.

## DNA Deposition

The following information was supplied regarding the deposition of DNA sequences:

The sequences are available at GenBank: OK847665–OK847720, OK666850–OK666869, MZ218005–MZ218051, MZ209424–MZ209593.

## Data Deposition

The data is available Supplemental Files.

## New Species Registration

The following information was supplied regarding the registration of a newly described species:

*Pseudohygrohypnum appalachianum, Pseudohygrohypnum orientale, Pseudohygrohypnum sibiricum, Pseudohygrohypnum subarcticum, Pseudohygrohypnum neglectum.*

## Supplemental Information

Supplemental information for this article can be found online at http://dx.doi.org/10.7717/peerj.13260#supplemental-information.

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

## FURTHER READING

**Karger DN, Conrad O, Böhner J, Kawohl T, Kreft H, Soria-Auza RW, Zimmermann NE, Linder P, Kessler M. 2017.** Climatologies at high resolution for the Earth land surface areas. *Scientific Data* **4**:170122 DOI 10.1038/sdata.2017.122.

**Zomer RJ, Trabucco A, Bossio DA, Verchot LV. 2008.** Climate change mitigation: a spatial analysis of global land suitability for clean development mechanism afforestation and reforestation. *Agriculture, Ecosystems & Environment* **126**(1–2):67–80 DOI 10.1016/j.agee.2008.01.014.