# Peer review of "Need for split: integrative taxonomy reveals unnoticed diversity in the subaquatic species of Pseudohygrohypnum (Pylaisiaceae, Bryophyta)"

_PeerJ, doi:10.7717/peerj.13260_

## Round 0.1 · original submission · Major Revisions

Dear Dr. Fedosov,

Overall, the reviewers find your manuscript interesting, and they think it may provide new knowledge on mosses species diversity. However, according to one of them the discussion needs to be more focused on your finds, more clearly structured and less speculative. He/she also detects some weakness in methods explanation. So, I encourage you to improve the manuscript according to reviewer suggestions. Please, respond point-to-point to the comments of reviewers to speed up the process of revision.

Once again, thank you for submitting your manuscript to PeerJ and we look forward to receiving your revision.

Sincerely,
Gabriele Casazza

Reviewer 1 ·

Basic reporting

This is a very solid study with a substantial amount of new phylogenetic and morphological information that illustrates how integrative taxonomy can help us to understand better bryophyte diversity. It also opens the door to new and exciting questions such as the intriguing difference in "behavior" of different organismal groups such as vascular plants and animals opposed to bryophytes regarding Rapoport´s rule (only one p), for example. The methods are explained with great detail and the outcomes include a distributional map and graphs that highlight ecological differences in the habitat of the revealed lineages. Further, a key and diagnosis, as well as high quality images, drawings and tables of morphology are provided that can be used by a wide range of bryologists.

Experimental design

The laboratory and analytical methods are generally sound and explained in great detail. The results of the study are convincing and sufficiently supported by the data.

Validity of the findings

The sampling intensity within individual populations, and proximate sites is sufficient to draw conclusions about the population structures.

Additional comments

Overall, this is a valuable and interesting contribution and I recommend its publication.

Reviewer 2 ·

Basic reporting

This manuscript presents a good example of the hidden diversity that is frequently found in organisms with a relatively simple morphology, and then especially in environments where variable habitat conditions are likely to influence the phenotype. In addition, it highlights our poor understanding of the diversity of Asia outside the tropics. The latter no doubt reflects difficulties in accessing material from these enormous regions of Asia, something that was even more problematic in earlier times.

The English should be checked by a native English speaker. One example is the use or absence of the definite article (the).

Discussion: In general, I find the Discussion too speculative and difficult to follow. It consists of more than eight manuscript pages without a single break. I have a few suggestions to improve the Discussion in a way that will simplify for the reader to understand. First, I suggest that subheadings are used to introduce a clear structure. Second, I would suggest a clearer focus on Pseudohygrohypnum itself and less on general Holarctic diversity and species evolution. Third, I would take out speculation that does not relate directly to Pseudohygrohypnum, and make sure that the remaining speculation is backed up by relevant references. If these changes are made, my impression is that the length of the Discussion could easily be reduced by 20%.

In the text, Fig. 20 is cited before Fig. 12. On the other hand, I cannot really see the relevance of including Fig. 20 at all, and I suggest deleting this figure.

Figure 1: PP and BP values <0.95 and <0.67, respectively, indicate no support. This should be considered in the tree.

Figure 2: From what are the Bootstrap values inferred?

Figure 3: The colour of the samples does not seem to correspond with the descriptions. Are these old herbarium specimens?

Figure 8: Indicate which entities differ statistically significantly from each other using letters (e.g., a, b, ab, ac, c, d, and so on).

Figure 9: Why are the colours different from those for the same entities in Fig. 7?

Figure 10: The temperature curve is somewhat difficult to see.

Experimental design

The methods used to elucidate which species occur among aquatic Pseudohygrohypnum are in general highly suitable for the purpose. Besides a few unclear points (see 4. Additional comments), regarding the chosen methodology I only have a few comments regarding the MaxEnt modelling. First, I was happy to see that the data were adjusted for sampling effort. Secondly, it was a bit unclear if (lines 242 ff.) the authors first used the equivalency test. If this was the case, it should be noted that the equivalency test tends to excessively reject the null hypothesis of niche identity and should therefore be avoided (Broennimann, et al. 2012. Measuring ecological niche overlap from occurrence and spatial environmental data. Global Ecology and Biogeography 21: 481-497; Peterson. 2011. Ecological niche conservatism: a time-structured review of evidence. Journal of Biogeography 38: 817-827). If this test was used, this needs to be discussed and its use motivated.

Validity of the findings

As I explain below, I believe the Discussion needs to be shorter and better focussed so that the reader could understand which of the results are both significant and supported. At the moment, much of the Discussion seems to want to cover much more than the study itself justifies. One thing that strikes me as a contradiction that may be due to the authors' wanting to cover too much here is that the authors conclude that aridification is a driver for diversification. How can this be for organisms that are subaquatic and obviously stay subaquatic during the diversification?

This manuscript requires a clearer focus and structure, where the reader can easily follow the reasoning of the authors.

Additional comments

Several geographical names will be unfamiliar to most readers. Please consider if you can add some additional geographical information for those names that are potentially unfamiliar. (e.g., Bashkiria, Sikhote-Alin, Khamar-Daban, Badzal)

Lines 117 ff.: Please write out the complete names of the used molecular markers the first time they appear in the text. Also, note that in several places where the abbreviated molecular markers should be partly in italics, they are entirely in normal fonts.

Lines 123-124: There seems to be something wrong in this sentence.

Lines 163-164: I do not understand this sentence.

Line 172: What is meant by ‘considered in details’?

Lines 220 ff.: Many H. eugyrium specimens collected before 1970 were checked by Jamieson, who recognized H. subeugyrium. Thus, did you check older American H. eugyrium samples against the identifications cited in Jamieson’s thesis? If not, this could possibly expand the certain geographical distribution of H. subeugyrium in N. America.

Line 254: Is ‘(R-project)’ a reference? Normally, a reference to R looks different.

Line 261 (and elsewhere): Please explain acronyms, such as LSID, the first time they are used.

Line 320: What does ‘reasonably’ imply?

Line 322: Is the habitat used as an ‘informal’ character?

Line 341 (and elsewhere): Write out climate parameters, like BiO1 (BiO2, BiO7), the first time they are used in the text and if it helps to follow the text when they are brought up again much later. It is otherwise difficult to follow the text.

Line 367: Should Fig. 8 be cited in the second sentence?

Line 404: Should ‘valuable’ be changed to ‘significant’?

Lines 616 ff.: I believe it is relevant to concentrate on truly Amphiatlantic patterns (i.e., surrounding the Atlantic) and not W Eurasiatic-W North American patterns; the latter is what, for example, Huttunen et al. (2008) discuss. At least the difference between W and E North America needs to be seriously discussed here if such patterns should be considered. Homalothecium sericeum s.str. shows an Amphiatlantic distribution, like some Sphagnum species and Antitrichia curtipendula s. str. (with A. gigantea in W North America). At least some of these taxa most likely got this kind of distributions in connection with the Pleistocene oscillations rather than because of much older events.

Line 672: What is the evidence for bottlenecks and genetic drift in Pseudhygrohypnum?

Line 784: ‘shorter’ than what? (No length is provided under the first alternative).

Line 807 (and elsewhere): In what sense ‘peculiar’? This term means different things to different readers, so please describe how it looks instead.

Lines 819 ff.: This concerns all species descriptions. I found several discrepancies between the first two descriptions (which I checked carefully), regarding included characters, how characters are presented, and the order in which they are presented. Descriptions at the same hierarchical level need to be similarly arranged and to include the same characters to be fully useful to the users.

Line 860: What does ‘more developed’ mean?

Line 902: Why include Affinity here? This is a topic for the Discussion.

Line 1006: ‘red’ rather than ‘pinkish’?

Line 1103: ‘Tysslinge’

Line 1535: Either the central strand is absent or present. Thus, ‘clearly’ is redundant.

References: Authors are cited in at least three different ways. Capitalisation of titles also differ, and Latin names of species are mostly in normal fonts, but in italics in the last one. The references need to be formatted correctly for PeerJ.

Table 1: The shape of the leaf cells should be indicated for all species.

Table 2. In the legend, please explain what I and D mean (which tests ).

Reviewer 3 ·

Basic reporting

I believe that the paper entitled 'Need for split: integrative taxonomy reveals unnoticed diversity in subaquatic species of Pseudohygrohypnum' is of great interest for the scientific community. The authors have performed a detailed study on the morphology, genetical variation and ecological affinities (especially regarding climatic conditions) for a representation of specimens of Pseudohygrohypnum that covers all the distributional area of this genus. The authors conclude that there is a higher diversity within this genus than that currently recognized and accordingly propose several new species. This proposal is sustained by clear genetical and ecological evidences, and includes a good morphological description of the studied taxa, including the newly proposed species. In addition, the authors include complete and clear illustrations of the considered morphotaxa, as well as a useful identification key that will help the bryologist to adopt their proposal. In my opinion, this is a very clear and good example of use of the integrative taxonomical approach.

My main concerns regarding the manuscript include format and stile improvements, as well as English expression. The manuscript would profit from a revision of the English text, and the authors should carefully revise the guidelines for authors, especially regarding literature references and text subdivision.

Experimental design

This is an original research that perfectly covers the scope and interests of this journal. The research question is clearly exposed and addressed, and the experimental design is appropriate to answer the question. The investigation has been rigorously performed and meets the currently desired technical standards. Methods are in general clearly described.

Validity of the findings

As indicated above, I think that the results of this paper are of general interest for bryologist, since the authors clarify the real diversity of the studied genus. In addition, I think that this paper constitutes an example of good praxis in taxonomy that can help researchers from all disciplines to approach taxonomical studies in an integrative frame that combines morphological, genetical and biogeographical evidences. In this sense, I believe that the conclusions of this study are in general terms very robust.

Additional comments

As indicated above, I think that the authors should address some changes, especially regarding the English stile to improve and clarify the manuscript. In my opinion, the authors should consider to send the paper to an English native speaker.
I highlight here some of the points that could be addressed, although as I said I think that a general revision of English expression is needed all along the manuscript:
- Summary: the use of the words genus and species is confusing: one genus cannot be an example of species. Do they refer to ‘species complex’?
- Line 90: ‘the unusual distribution of P. subeugyrium s.l.’ Which is this distribution?
- Lines 105-117: the selection of specimens is not clearly described. I would suggest to change the order: The study is based on the analysis of Pseudohygrohypnum eugyrium and P. subeugyrium, for which XX accession are included that cover all their known distributional area plus the detected morphological lineages. This study is performed within a phylogenetical frame that includes all the species currently included in this genus (a total of xx accessions from xx species), plus an outgroup that includes xxxx.
- Line 147. How did the authors decide which model to use? Please, specify.
- Line 157: indicate reference for FigTree.
- Line 164: it is not clear to me what has happened with some of the markers that the authors had amplified, such as the mitochondrial. Please, especify.
- Line 226: indicate references for QGIS and ArcGIS
- Results section: I miss a description of the lenght and varibility of the matrices. This information would be needed to understand the validity of the results and the incongruences observed among the plastid and ITS data sets.
- Regarding the analysis of the environmental variables, I believe that there exists some correlation among the considered variables, such as Bio12 with Bio17, Bio19 and Bio19. Have the authors taken this into account?
- Line 365: the authors refer to the warmest quarter. Please, indicate that this refers to BiO18
- The explanation about BiO19 is missing
- Discussion, lines 463-490. I find all this part confusing. I think that it would help to divide the discussion of each new species in a different paragraph.
- Line 481: the authors indicate that type material has not been yet revised. Ideally, this should be done in advance to the present publication. If this is not possible, the authors should indicate why this has not been done.
- Line 724: diversity of “this group” of bryophytes
- Line 769: check italics for P. subeugyrium
- Lines: 967-968. The authors indicate that P. orientale was illustrated and described by Czernyadjeva as P. eugyrium, Do they mean that this species was included within the variability described for P. eugyrium? Please, clarify.
- Lines 1405-1406. The same cmment as above for P. subarcticum under P. subeugyrium.
- Figure 10: I think that the text would be sometimes easier to follow if the names of the species were given (eg P. orientale, etc.)
In addition, I have detected several mistakes in the references list that should be corrected:
- Lines 83 and 93: Czernyadjeva (2002) is 2003 in the list
- Line 94: Afonina, 2009 is 2019 in the list
- Line 142: Lanfear et al., 2012 is 2017 in the list
- Line 181: Tamur should be Tamura
- Line 628: Confirm whether is Grímsson or Grimsson
- Line 638: Does Denk, Grímssn & Zetter refer to Denk et al 2013? Please, confirm.
- Line 1066: check author (or editor) of Flora of North America and cite accordingly
- Line 1068: check Czernyadjeva et al 2003 (only Czernyadjeva in the list)
- Line 1905: Smith 1978, check if this reference is cited and in this case complete in the list
Finally, the authors should check the instructions for authors regarding the following points:
- Use of italics for the phylogenetic markers used (check and correct all along the manuscript).
- Use of ‘Appendix’ or ‘Supplementary material’
- Use of ‘et al’ or list all of three authors (e.g. line 551 Gama et al, Moroni et al; line 659 Liu et al; line 743 Pärtel et al)
- Separation in paragraphs. The format is very compact and this is sometimes not useful for the readers. For example, I would suggest to separate the identification key from the descriptions, and one description from another
- References list: the authors should check the format required by the journal. Please, pay especial attention to: the position of the year of publication (sometimes indicated twice, such as Afonina 2019), the use of italics for the scientific names (generally missing in all the list), the position of the accessory letters a and b in the case of El-Gabbas & Dormann 2018, the use of capital letters (see for eg. Hall 1999 or Katoh & Standley 2013), avoid the names of the authors (use only the initial, for eg. in Huttunen et al 2012 or Ignatov et al 2020), check the use of coma, dot or parenthesis for the year.

---

## Round 0.2 · Minor Revisions

Dear Dr. Fedosov,

The reviewer found your manuscript was strongly improved and he/she detected only minor concerns. So, I encourage you to improve the manuscript according to the tips of the reviewer. Please, respond point-to-point to the comments of reviewers to speed up the process of revision.

Once again, thank you for submitting your manuscript to PeerJ and we look forward to receiving your revision.

Sincerely,
Gabriele Casazza

Reviewer 2 ·

Basic reporting

I was very happy to see this manuscript again, and that the authors have followed my suggestions to make the Discussion text much clearer. It is now easy to follow the Discussion and to understand their reasoning. I found just a few minor issues that require attention.

Line 130: ‘scored using the’

Line 132: ‘homoplastic’

Line 148’ frequency of one’

Line 156: ‘on the extended’

Line 180: ‘contradict the’

Lines 286-306, and Fig. 1: PP < 0.95, and BP < 0.67 are insignificant. Even if was claimed in the rebuttal that the cutting levels were changed, this is not so. I do not understand why the authors put arbitrary cut levels instead of using the relevant ones. This will affect some of the branches in the trees, and may affect the text in the Results / Discussion concerning this. The network really clarifies the situation nicely (a tree is actually a special case of a network, so this is not surprising).

Line 333: Change ‘scheme’ to ‘hypothesis’

Line 379: Delete the last ‘one’

Line 479: ‘from the ASAP’

Line 522: ‘Presumably, the’

Lines 567ff.: Here, it may be worth pointing out that the found niche differentiation likely reflects a primary physiological differentiation (which is as significant as a morphological differentiation).

Line 590: Change ‘represent a’ to ‘are’

Line 653: Change ‘amphipacific’ to ‘amphiatlantic’

Lines 688, 701, 703: Avoid teleological expressions (managed to, succeeded in)

Line 730: ‘our climate model’

Line 736: Delete the comma after ‘both’

Figure 2: From what are the Bootstrap values inferred? I cannot see that any information on how they were generated has been added.

Experimental design

no comment

Validity of the findings

no comment

---

## Round 0.3 · accepted · Accept

Dear Dr. Fedosov,

I am very pleased to inform you that your paper "Need for split: integrative taxonomy reveals unnoticed diversity in the subaquatic species of Pseudohygrohypnum (Pylaisiaceae, Bryophyta)" is accepted for publication in the PeerJ. Congratulations!

Thank you for submitting your work to PeerJ.

Sincerely,
Gabriele Casazza